# Crossing Symmetric Spinning S-matrix Bootstrap: EFT bounds

Subham Dutta Chowdhury[a*], Kausik Ghosh[b†], Parthiv Haldar[c‡],
Prashanth Raman[c§] and Aninda Sinha[c¶]

[a] *Enrico Fermi Institute & Kadanoff Center for Theoretical Physics,*
*University of Chicago, Chicago, IL 60637, USA*

[b] *Laboratoire de Physique Théorique,*
*de l'École Normale Supérieure, PSL University,*
*CNRS, Sorbonne Universités, UPMC Univ. Paris 06*
*24 rue Lhomond, 75231 Paris Cedex 05, France*

[c] *Centre for High Energy Physics, Indian Institute of Science,*
*C.V. Raman Avenue, Bangalore 560012, India.*

## Abstract

We develop crossing symmetric dispersion relations for describing 2-2 scattering of identical external particles carrying spin. This enables us to import techniques from Geometric Function Theory and study two sided bounds on low energy Wilson coefficients. We consider scattering of photons, gravitons in weakly coupled effective field theories. We provide general expressions for the locality/null constraints. Consideration of the positivity of the absorptive part leads to an interesting connection with the recently conjectured weak low spin dominance. We also construct the crossing symmetric amplitudes and locality constraints for the massive neutral Majorana fermions and parity violating photon and graviton theories. The techniques developed in this paper will be useful for considering numerical S-matrix bootstrap in the future.

---

[*]subham@uchicago.edu

[†]kau.rock91@gmail.com

[‡]parthivh@iisc.ac.in

[§]prashanth.raman108@gmail.com

[¶]asinha@iisc.ac.in

# 1  Introduction

In the context of EFTs, it is important to put bounds on the theory space [1–3]. In recent times, there has been an increase in interest in establishing two-sided bounds on ratios of parameters in front of higher-dimensional operators in the EFT lagrangians. Recent work in this direction include [4–20]. Starting from the original attempts to constrain scalar EFTs, research has been extended to external particles carrying spin [7, 21, 22]. It is thus of interest and importance to know which ratios can be bounded and what the mathematical reasons for the existence of such bounds are.

The standard attempts to put bounds on EFT coefficients begin with a fixed-$t$ dispersion relation. Then imposing crossing symmetry leads to constraints, dubbed null constraints [5, 23]. These null constraints lead to two-sided bounds on Wilson coefficients. In [24, 25] a different consideration was put forth, which makes use of powerful techniques and theorems in an area of mathematics called Geometric Function Theory (GFT) [26, 27]. The starting point in this approach makes use of a crossing symmetric dispersion relation (CSDR) [28–30]. In this approach, crossing symmetry is manifest at the outset. However, the penalty paid is the loss of locality, leading to the "locality constraints" [29, 30]. These locality constraints are essentially a linear combination of the null constraints in the fixed-$t$ approach [30].

The main advantage of using the CSDR is that instead of the usual Mandelstam variables $(s, t, u)$ it is more natural to use a different dispersion variable $z$ and a parameter $a$, which is held fixed. The amplitude for identical scalars then has unobvious and interesting properties in terms of the function in the complex $z$ plane. As shown in [25], for a suitable range of the parameter $a$, for pion scattering, the amplitude is, in the parlance used in Geometric Function Theory, typically real. In other words, in this range of $a$, it satisfies the condition

$$Im f(z) Im z > 0 \,, \tag{1.1}$$

inside the unit disk $|z| < 1$, which in turn imposes the Bieberbach-Rogosinski (BR) two-sided bounds on the Taylor expansion coefficients of $f(z)$. In terms of Wilson coefficients, an argument based on the Markov brothers' inequality, as shown in [25], leads to two-sided bounds on the ratios of Wilson coefficients. It is known that there is a connection between typically real functions in geometric function theory (GFT) and quantum field theory (QFT) dating back to [31–35]. However, it has not been used extensively in the study of scattering amplitudes.

In this paper, we will extend the CSDR for identical, neutral external particles carrying spin. Our formalism is general, although for concreteness, we will focus on the 2-2 scattering of photons and gravitons, as well as neutral Majorana fermions. In the photon and graviton cases, we will be able to identify combinations of helicity amplitudes whose Taylor expansion coefficients are two-sided bounded using GFT arguments. We will be able to write down a general expression for the locality constraints. Our formalism paves the way for a future systematic study of the S-matrix bootstrap for the 2-2 scattering of identical particles with spin. The crossing symmetric dispersion relation of a scalar amplitude $M_0(s_1, s_2)$ takes the following form,

$$M_0(s_1, s_2) = \alpha_0 + \frac{1}{\pi} \int_{M^2}^{\infty} \frac{ds_1'}{s_1'} \mathcal{A}(s_1', s_2^+(s_1', a)) H(s_1'; s_1, s_2, s_3) \tag{1.2}$$

where $\mathcal{A}(s_1', s_2^+(s_1', a))$, called the absorptive part, is the $s$- channel discontinuity and $H(s_1'; s_1, s_2, s_3)$ is a manifestly crossing symmetric kernel. The parameter $a = (s_1 s_2 s_3)/(s_1 s_2 + s_2 s_3 + s_3 s_1) \equiv y/x$ is kept fixed writing this dispersion relation and $s_2^+$ is one of the two roots obtained from this equation on using $s_1 + s_2 + s_3 = 0$. For a massive theory with a gap, as for pion scattering, the dispersive integral starts at

$8m^2/3$, where $m$ is the mass of the pion. In this case $s_1 = s - 4m^2/3, s_2 = t - 4m^2/3, s_3 = u - 4m^2/3$ with $s, t, u$ being the usual Mandelstam variables. For EFTs, the lower limit starts at some cut-off $M^2$ and all external particles are considered massless. The absorptive part can be expanded in partial waves involving Gegenbauer polynomials. Then Taylor expanding around $a = 0$ leads to the conclusion that for each partial wave, there are in principle any arbitrary power of $a$, and hence of $x = s_1s_2 + s_2s_3 + s_3s_1$ which are absent for a local theory. On demanding that such powers responsible for non-local terms cancel, leads to what we call the "locality" constraints.

The range of the parameter $a$ is crucial in this story. For theories with a gap, as for pion scattering, axiomatic arguments can be used to finding this range of $a$ [24]. However, when describing EFTs [4–6, 23], these axiomatic arguments do not work. In this paper, taking a leaf out of [5], we will make use of the locality constraints and linear programming, and establish the range of $a$ where the absorptive part of the amplitude is positive. This is crucial since, in our approach, it is vital that this range of $a$ satisfies $-a_{min} < a < a_{max}$ with both ends non-zero to have two-sided bounds. One surprising conclusion that will emerge from our analysis is that this range is related to the weak Low Spin Dominance (wLSD) conjecture made in [36]. Our findings lead to the conclusion that a few low-lying spins control the sign of the absorptive part. For non-unitary theories, generically, the imaginary part of the partial wave coefficient (often referred to as the spectral function) is not of a definite sign. However, if it is known that the spectral functions for some low-lying spins are positive, it is possible then to have two-sided bounds in a local but non-unitary theory using wLSD.

We will focus on light-by-light scattering and graviton scattering in weakly coupled EFTs [6, 36, 37] and derive two-sided bounds. We consider the linearly independent helicity amplitudes, $T^{\lambda_3\lambda_4}_{\lambda_1\lambda_2}(s_1, s_2, s_3)$, for 2-2 scattering ($\lambda_1\lambda_2 \rightarrow \lambda_3\lambda_4$) of graviton, photon and massive Majorana fermions in four spacetime dimensions. Here $\lambda_i$ are helicity labels and these take values $-j$ and $+j$ for massless particle with spin $j$ while there are $2j + 1$ independent helicities for a massive particle with spin $j$. Generically these helicity amplitudes mix among themselves under crossing,

$$T^{\lambda_3\lambda_4}_{\lambda_1\lambda_2}(s_1, s_2, s_3) \rightarrow \sum_{ijkl} C_{ijkl} T^{\lambda_k\lambda_l}_{\lambda_i\lambda_j}(P_{s_1}, P_{s_2}, P_{s_3}), \tag{1.3}$$

where $(P_{s_1}, P_{s_2}, P_{s_3})$ represents some permutation of $(s_1, s_2, s_3)$. Using representation theory of $S_3$ (the permutation group relevant for Mandelstam invariants), we construct a basis of crossing symmetric amplitudes, $F(s_1, s_2, s_3)^1$, using the helicity amplitudes $T^{\lambda_3\lambda_4}_{\lambda_1\lambda_2}(s_1, s_2, s_3)$. These crossing symmetric helicity amplitudes transform as a singlet under $S_3$,

$$F(s_1, s_2, s_3) = F(s_2, s_1, s_3) = F(s_3, s_2, s_1). \tag{1.4}$$

Our construction can be generalised to other spacetime dimensions in a straightforward manner, although for this paper, we will focus on $d = 4$. We then write down locality constraints associated with the crossing symmetric amplitudes $F^{\lambda_3\lambda_4}_{\lambda_1\lambda_2}(s_1, s_2, s_3)$. Explicit formulae for a subclass of the amplitudes in closed form can be found. Our method allows us to write the locality constraints for all the crossing symmetric amplitudes. To be precise, we consider the following crossing symmetric amplitudes for the photon case[2],

$$F^\gamma_1(s_1, s_2, s_3) = T_2(s_1, s_2, s_3), \quad F^\gamma_2(s_1, s_2, s_3) = T_1(s_1, s_2, s_3) + T_3(s_1, s_2, s_3) + T_4(s_1, s_2, s_3) \tag{1.5}$$

where the helicity amplitudes $T_i$ are defined in (2.9). These amplitudes have the low energy EFT expansion

$$F^{\gamma,i}(s_1, s_2) = \sum_{p,q} \mathcal{W}^i_{p,q} x^p y^q. \tag{1.6}$$

---

[1]We don't put any helicity labels for crossing symmetric amplitudes since they are often combinations of amplitudes with different helicity labels (2.20).

[2]The relevant amplitudes for graviton are a bit subtle and will be dealt with in section 6, and we will just quote the results here.

The locality constraints for the amplitude $F_2^\gamma(s_1, s_2, s_3) + x_1 F_1^\gamma(s_1, s_2, s_3) = \sum_{p,q} \mathcal{W}_{p,q}^{(x_1)} x^p y^q$ for $x_1 \in [-1, 1]$, are

$$\mathcal{W}_{p,q}^{(x_1)} = 0, \quad \forall \ p < 0.$$

We present the explicit expressions for $\mathcal{W}_{p,q}^{(x_1)}$ using CSDR in the main text (eqn (5.11)) and from those we obtain positivity conditions called $PB_C^\gamma$ (eqn (5.14)). We also show that the dispersive part of the amplitude can be written as a Typically Real function leading to bounds on the range of the variable $a$. In general, for massless theories the lower bound on $a = a_{min}$ is zero [25], which only leads to one-sided bounds. We observe that the Wigner-$d$ functions, $d_{m,n}^\ell(\sqrt{\xi(s_1, a)})$, are positive for all spins when its argument $\xi(s_1, a)$ is greater than 1. Adding a suitable linear combination of the locality constraints, we can show the positivity of the absorptive part arises even when $\xi(s_1, a) < 1$. This translates to $-a_{min} < a < a_{max}$. This is indicative of the dominance of low spin partial waves in EFTs and is called Low spin dominance (LSD). This behaviour was observed for gravitons in [2, 7]. In this paper, we will show how this naturally emerges out of our analysis using the locality constraints. We will show that the lower range of $a$ tells us about which spins dominate in the determination of the positivity of the absorptive part for $-a_{min} < a < a_{max}$. We demonstrate the same for the case of type-II string amplitude in appendix G.

After showing that the amplitude is typically-real for a range of $a \in [-a_{min}, a_{max}]$, we can directly find two sided bounds on the ratio of Wilson coefficients $w_{p,q} = \frac{\mathcal{W}_{p,q}}{\mathcal{W}_{1,0}}$ from GFT. Below we show examples of bounds found for scattering of scalars, photons and gravitons in Table 1. The detailed list of bounds for photon and graviton scattering are summarised in Table 3 and 4.

| Theory | EFT amplitudes | Range of $a$ and LSD | $w_{01}$ bound |
|---|---|---|---|
| Scalar | $F(s_1, s_2, s_3) = \mathcal{W}_{1,0} x + \mathcal{W}_{01} y + \cdots$ | $-0.1933 M^2 < a^{scalar} < \frac{2M^2}{3}$ (Spin-2 dominance) | $\frac{-3}{2M^2} < w_{0,1} < \frac{5.1733}{M^2}$ |
| Photon | $F_2(s_1, s_2, s_3) = 2g_2 x - 3g_3 y + \cdots$ $F_1(s_1, s_2, s_3) = 2f_2 x - f_3 y + \cdots$ | $-0.1355 M^2 < a^\gamma < \frac{2M^2}{3}$ (Spin-3 dominance) | $\frac{-4.902}{M^2} < \frac{g_3 + x_1 \frac{f_3}{3}}{g_2 + x_1 f_2} < \frac{1}{M^2}$ where $x_1 \in [-1, 1]$ |
| Graviton | $\tilde{F}_2^h = 2x f_{0,0} + 3y f_{1,0} + \cdots$ | $-0.1933 M^2 < a^h < \frac{2M^2}{3}$ ( Spin-2 dominance) | $-\frac{1}{M^2} < \frac{f_{1,0}}{f_{0,0}} < \frac{3.44}{M^2}$ |

**Table 1:** Example of two sided bounds we have found for scalars, photons and gravitons using GFT.

Apart from the conceptual clarity that the GFT techniques enable us with, are there any technical advantages using our approach? We wish to point out a couple of obvious ones. First, unlike the fixed-$t$ methods where one uses SDPB techniques and hence needs to worry about convergence in the spin, the dispersive variable as well as the number of null constraints, in our approach, once the range of $a$ has been determined, one only needs to check for convergence in the number of $BR$ inequalities we use. Second, we can write simple codes directly in Mathematica to study bounds. However, there are also some disadvantages. The main one is that while we do obtain two-sided bounds quite easily, these are not necessarily the sharpest ones possible since we do not make use of all the locality constraints. It is not clear to us if there is a way to get optimum bounds[3] using purely GFT techniques.

The paper is organised as follows. In section 2, we describe the construction of fully crossing symmetric amplitude. Through multiple subsections of section 3, we describe the key formulas like CSDR, locality constraints, typical realness of the amplitude and then we introduce BR bounds as well. We also discuss in section 3 how low spin dominance emerges out of our analysis, taking into account the locality constraints. Through section 4,5, 6 we describe the bounds obtained for scalars, photons and gravitons respectively. We end our discussion with concluding remarks in section 7. Several technical details are relegated to multiple appendices at the end.

---

[3]A bound will be considered optimum if there is a consistent S-matrix saturating it.

# 2 Crossing symmetric amplitudes

In this section, we present a general construction for crossing symmetric amplitudes following [38]. Let us begin with a short review of scattering amplitudes of identical particles as irreducible representations (irreps) of $S_3$ [39, 40]. Consider the scattering of four identical particles (massive or massless, with or without spin) in $d = 4$. The momenta of the particles satisfy,

$$p_i^2 = -m^2, \qquad \sum_{i=1}^{4} p_i^\mu = 0, \tag{2.1}$$

where $m$ is mass of each particle. We use the mostly positive convention and define Mandelstam variables,

$$
\begin{aligned}
s &:= -(p_1 + p_2)^2 = -(p_3 + p_4)^2 = 2m^2 - 2p_1.p_2 = 2m^2 - 2p_3.p_4 \\
t &:= -(p_1 + p_3)^2 = -(p_2 + p_4)^2 = 2m^2 - 2p_1.p_3 = 2m^2 - 2p_2.p_4 \\
u &:= -(p_1 + p_4)^2 = -(p_2 + p_3)^2 = 2m^2 - 2p_1.p_4 = 2m^2 - 2p_2.p_3.
\end{aligned}
\tag{2.2}
$$

Due to momentum conservation we have $s + t + u = 4m^2$. For identical bosonic particles, the S-matrix is to be thought of as the function of Mandelstam invariants (and polarizations), which is $S_4$ invariant, the symmetry group of permutations of four particles. In the present context, $S_4$ acts on the momenta and the helicities of the particles. We usually impose the $S_4$ invariance in two steps. Recall that that $\mathbb{Z}_2 \times \mathbb{Z}_2$ is the normal subgroup of $S_4$ and the remnant symmetry is $\frac{S_4}{\mathbb{Z}_2 \times \mathbb{Z}_2} = S_3$. Action of $\mathbb{Z}_2 \times \mathbb{Z}_2$ on four objects $(1, 2, 3, 4)$ is the simultaneous exchange of two particles- $(12)(34)$, $(13)(24)$ and $(14)(23)$ while $S_3$ is the permutation of three objects $(1, 2, 3)$. Since the Mandelstam invariants $s, t$ and $u$ are invariant under the $\mathbb{Z}_2 \times \mathbb{Z}_2$, we first impose $\mathbb{Z}_2 \times \mathbb{Z}_2$ invariance, which leaves the Mandelstam invariants unchanged and we are left with the remnant $S_3$ symmetry which acts on $(s, t, u)$. Note that helicities (or equivalently tensor structures in higher dimensions) may not be $\mathbb{Z}_2 \times \mathbb{Z}_2$ invariant, and we might need to impose $\mathbb{Z}_2 \times \mathbb{Z}_2$ invariance. However, for most of the non-crossing symmetric helicity amplitudes that we consider in this work, the $\mathbb{Z}_2 \times \mathbb{Z}_2$ symmetry has already been taken care of [41]. The S-matrix, which is invariant under the $\mathbb{Z}_2 \times \mathbb{Z}_2$ invariance, is often referred to as "Quasi-invariant" S-matrix. The "Quasi-invariant" S-matrix can be decomposed into irreps of $S_3$ and the *crossing equations are relations between the orbits of* $S_3$.

To simplify the discussion, unless otherwise mentioned, we will work with the following shifted Mandelstam variables,

$$s_1 = s - \frac{4m^2}{3} \qquad s_2 = t - \frac{4m^2}{3} \quad , \quad s_3 = u - \frac{4m^2}{3}, \tag{2.3}$$

such that, $s_1 + s_2 + s_3 = 0$. With the aid of the representation theory of $S_3$, which we review in appendix A, one can write the most general Quasi-invariant S-matrix, therefore, takes the form [38]

$$
\begin{aligned}
F(s_1, s_2, s_3) = \; & f(s_1, s_2, s_3) + (2s_1 - s_2 - s_3)g_1(s_1, s_2, s_3) + (s_2 - s_3)g_2(s_1, s_2, s_3) \\
& + (2s_1^2 - s_2^2 - s_3^2)h_1(s_1, s_2, s_3) + (s_2^2 - s_3^2)h_2(s_1, s_2, s_3) \\
& + (s_1 - s_2)(s_2 - s_3)(s_3 - s_1)j(s_1, s_2, s_3),
\end{aligned}
\tag{2.4}
$$

where $f(s_1, s_2, s_3), j(s_1, s_2, s_3), g_i(s_1, s_2, s_3)$ and $h_i(s_1, s_2, s_3)$ are crossing symmetric amplitudes. We can decompose $F(s_1, s_2, s_3)$ into irreps of $S_3$,

$$F(s_1, s_2, s_3) = f_{\text{Sym}}(s_1, s_2, s_3) + f_{\text{Anti−sym}}(s_1, s_2, s_3) + f_{\text{Mixed+}}(s_1, s_2, s_3) + f_{\text{Mixed−}}(s_1, s_2, s_3). \tag{2.5}$$

From eqn (A.6) and (2.5), we have the following set of equations

$$
\begin{aligned}
f_{\text{Sym}}(s_1, s_2, s_3) &= f(s_1, s_2, s_3), \\
f_{\text{Anti−sym}}(s_1, s_2, s_3) &= (s_1 - s_2)(s_2 - s_3)(s_3 - s_1)j(s_1, s_2, s_3), \\
f_{\text{Mixed+}}(s_1, s_2, s_3) &= (2s_1 - s_2 - s_3)g_1(s_1, s_2, s_3) + (2s_1^2 - s_2^2 - s_3^2)h_1(s_1, s_2, s_3), \\
f_{\text{Mixed−}}(s_1, s_2, s_3) &= (s_2 - s_3)g_2(s_1, s_2, s_3) + (s_2^2 - s_3^2)h_2(s_1, s_2, s_3).
\end{aligned}
\tag{2.6}
$$

This can be inverted to give the required crossing symmetric basis [38].

$$
\begin{aligned}
f(s_1, s_2, s_3) &= f_{\text{Sym}}(s_1, s_2, s_3)\,, \\
j(s_1, s_2, s_3) &= \frac{f_{\text{Anti-sym}}(s_1, s_2, s_3)}{(s_1 - s_2)(s_2 - s_3)(s_3 - s_1)}\,, \\
g_1(s_1, s_2, s_3) &= \frac{f_{\text{Mixed}+}(s_1, s_2, s_3)(s_1^2 + s_2^2 - 2s_3^2) - f_{\text{Mixed}+}(s_3, s_1, s_2)(s_2^2 + s_3^2 - 2s_1^2)}{3(s_1 - s_2)(s_2 - s_3)(s_3 - s_1)}\,, \\
h_1(s_1, s_2, s_3) &= \frac{f_{\text{Mixed}+}(s_3, s_1, s_2)(s_3 + s_2 - 2s_1) - f_{\text{Mixed}+}(s_1, s_2, s_3)(s_1 + s_2 - 2s_3)}{3(s_1 - s_2)(s_2 - s_3)(s_3 - s_1)}\,, \\
g_2(s_1, s_2, s_3) &= \frac{f_{\text{Mixed}-}(s_3, s_1, s_2)(s_2^2 - s_3^2) - f_{\text{Mixed}-}(s_1, s_2, s_3)(s_1^2 - s_2^2)}{(s_1 - s_2)(s_2 - s_3)(s_3 - s_1)}\,, \\
h_2(s_1, s_2, s_3) &= \frac{f_{\text{Mixed}-}(s_1, s_2, s_3)(s_1 - s_2) - f_{\text{Mixed}-}(s_3, s_1, s_2)(s_2 - s_3)}{(s_1 - s_2)(s_2 - s_3)(s_3 - s_1)}\,.
\end{aligned}
\tag{2.7}
$$

The following additional comments are in order:

- Given a Quasi-invariant S-matrix the algorithm to construct the crossing symmetric basis, therefore, is straightforward. We construct the irreps $\{f_{\text{Sym}}, f_{\text{Anti-sym}}, f_{\text{Mixed}\pm}\}$ following (A.6) and use (2.7) to construct the crossing symmetric basis[4].

- The basis elements *do not* have any spurious poles at $s_i = s_j$ and are analytic functions of $s_1, s_2, s_3$, which can be easily checked by plugging (A.6) into (2.7).

- The basis in eq.(2.7) is not unique since the last two equations of (2.6) are two 2 equations for 4 unknowns $\{g_i(s_1, s_2, s_3), h_i(s_1, s_2, s_3)\}_{i=1,2}$. One can use any permutation of the arguments of $f_{\text{Mixed}\pm}$ to get a system of full rank. We have used the permutations $s_i \to s_{i+1\text{mod}(3)}$ on $f_{\text{Mixed}\pm}$ as these are best suited for our purposes[5].

- If $F(s_1, s_2, s_3)$ is symmetric or anti-symmetric then only $f(s_1, s_2, s_3)$ and $j(s_1, s_2, s_3)$ are non zero respectively. Furthermore $F(s_1, s_2, s_3)$ is $t - u$ symmetric then only $\{f(s_1, s_2, s_3), g_1(s_1, s_2, s_3), h_1(s_1, s_2, s_3)\}$ are nonzero.

## 2.1 Photons and Gravitons

In this sub-section, we apply the formalism developed in the previous section to the case of parity even photon and graviton amplitudes. We will work with helicity amplitudes and show that they transform in irreps of $S_3$. Subsequently, we construct the crossing symmetric amplitudes from them using (2.7). As a consequence of the $CPT$ theorem and the fact that we will be considering particles on which charge conjugation acts trivially, our helicity amplitudes are $PT$ invariant. We will consider the sub-cases whether parity is preserved or not. We will follow the notations and conventions of [41].

### 2.1.1 $\mathcal{P}$ invariant theories

Massless photon and graviton theories in $d = 4$ are characterised by their helicities which can take values $(\pm 1)$ for photons and $(\pm 2)$ for gravitons respectively. This tells us that there are possibly 16 helicity amplitudes. Since the particles are identical, the helicity amplitudes enjoy a $\mathbb{Z}_2 \times \mathbb{Z}_2$ symmetry.

$$
\begin{aligned}
T^{\lambda_3, \lambda_4}_{\lambda_1, \lambda_2}(s_1, s_2, s_3) &= T^{\lambda_4, \lambda_3}_{\lambda_2, \lambda_1}(s_1, s_2, s_3), \qquad T^{\lambda_3, \lambda_4}_{\lambda_1, \lambda_2}(s_1, s_2, s_3) = T^{-\lambda_2, -\lambda_1}_{-\lambda_4, -\lambda_3}(s_1, s_2, s_3), \\
T^{\lambda_3, \lambda_4}_{\lambda_1, \lambda_2}(s_1, s_2, s_3) &= T^{-\lambda_3, -\lambda_4}_{-\lambda_1, -\lambda_2}(s_1, s_2, s_3)
\end{aligned}
\tag{2.8}
$$

---

[4]See also [42, 43] for similar considerations for the massive pion case.

[5]In [38] the permutation $s_1 \to s_3$ was used. We warn the reader referring to [38] that there is a minor typo in the analog of (2.4) where the sign of the $j(s_1, s_2, s_3)$ term is wrong.

Additionally, since we are looking at parity invariant theories, we have the following constraints from parity, time-reversal respectively,

$$T^{\lambda_3,\lambda_4}_{\lambda_1,\lambda_2}(s_1,s_2,s_3) = \eta_1^*\eta_2^*\eta_3\eta_4(-1)^{j_1+j_2+j_3+j_4}(-1)^{\lambda_1-\lambda_2-\lambda_3+\lambda_4}T^{\lambda_1,\lambda_2}_{\lambda_3,\lambda_4}(s_1,s_2,s_3)$$

$$T^{\lambda_3,\lambda_4}_{\lambda_1,\lambda_2}(s_1,s_2,s_3) = \varepsilon_1^*\varepsilon_2^*\varepsilon_3\varepsilon_4 T^{\lambda_1,\lambda_2}_{\lambda_3,\lambda_4}(s_1,s_2,s_3). \tag{2.9}$$

where $|\eta_i|^2 = |\epsilon_i|^2 = 1$. Note that for scattering of four identical photons and gravitons, we have $\eta_1^*\eta_2^*\eta_3\eta_4 = (|\eta|^2)^2 = 1$ trivially. These conditions reduce the number of independent parity preserving helicity amplitudes which are given by [41],

$$T_1(s_1,s_2,s_3) = T^{++}_{++}(s_1,s_2,s_3), \qquad T_2(s_1,s_2,s_3) = T^{--}_{++}(s_1,s_2,s_3), \qquad T_3(s_1,s_2,s_3) = T^{+-}_{+-}(s_1,s_2,s_3)$$
$$T_4(s_1,s_2,s_3) = T^{-+}_{+-}(s_1,s_2,s_3), \qquad T_5(s_1,s_2,s_3) = T^{+-}_{++}(s_1,s_2,s_3). \tag{2.10}$$

[6] These linearly independent set of five amplitudes are the basis of Quasi-invariant S-matrices defined in the previous section. They transform in irreps of $S_3$ which we determine from the following crossing equation [37, 41].

$$T^{\lambda_3,\lambda_4}_{\lambda_1,\lambda_2}(s_1,s_2,s_3) = \epsilon'_{23}T^{-\lambda_2,\lambda_4}_{\lambda_1,-\lambda_3}(s_2,s_1,s_3),$$
$$T^{\lambda_3,\lambda_4}_{\lambda_1,\lambda_2}(s_1,s_2,s_3) = \epsilon'_{24}T^{\lambda_3,-\lambda_2}_{\lambda_1,-\lambda_4}(s_3,s_2,s_1) \tag{2.11}$$

where $\epsilon'_{23}$ and $\epsilon'_{24}$ are arbitrary phases which were left unfixed from the general considerations of crossing symmetry using which (2.11) were derived. We will fix them in this section using constraints from consistency of crossing equations and comparing against explicit helicity amplitudes in literature. Note that (2.11) differs from the equivalent equation of [41] (eqns 2.81 and 2.82). This is due to the fact that we assume the following assignment for the Wigner-$d$ angles

$$\alpha_1 = 0, \quad \alpha_2 = \pi, \quad \alpha_3 = 0, \quad \alpha_4 = \pi$$
$$\beta_1 = 0, \quad \beta_2 = \pi, \quad \beta_3 = 0, \quad \beta_4 = \pi \tag{2.12}$$

in contrast with eqns 2.78 and 2.79 of [41]. Using (2.11), we can determine the crossing matrices to be,

$$C^p_{st} = \epsilon'_{23}\begin{pmatrix} 0 & 0 & 0 & 1 & 0 \\ 0 & 1 & 0 & 0 & 0 \\ 0 & 0 & 1 & 0 & 0 \\ 1 & 0 & 0 & 0 & 0 \\ 0 & 0 & 0 & 0 & 1 \end{pmatrix}, \qquad C^p_{su} = \epsilon'_{24}\begin{pmatrix} 0 & 0 & 1 & 0 & 0 \\ 0 & 1 & 0 & 0 & 0 \\ 1 & 0 & 0 & 0 & 0 \\ 0 & 0 & 0 & 1 & 0 \\ 0 & 0 & 0 & 0 & 1 \end{pmatrix}. \tag{2.13}$$

At this stage we have two undetermined phases $\epsilon'_{23}$ and $\epsilon'_{24}$. In order to determine the $t-u$ crossing relation we use the following relation for identical scattering particles [41],

$$T^{\lambda_3,\lambda_4}_{\lambda_1,\lambda_2}(s_1,s_2,s_3) = (-1)^{\lambda_2-\lambda_1+\lambda_4-\lambda_3}T^{\lambda_3,\lambda_4}_{\lambda_2,\lambda_1}(s_1,s_3,s_2)$$
$$T^{\lambda_3,\lambda_4}_{\lambda_1,\lambda_2}(s_1,s_2,s_3) = (-1)^{-\lambda_2+\lambda_1+\lambda_4-\lambda_3}T^{\lambda_4,\lambda_3}_{\lambda_1,\lambda_2}(s_1,s_3,s_2), \tag{2.14}$$

---

[6]Note that due to (2.9), the amplitudes $T_2$ and $T_5$ enjoy the additional symmetry

$$T^{--}_{++}(s_1,s_2,s_3) = T^{++}_{--}(s_1,s_2,s_3), \qquad T^{+-}_{++}(s_1,s_2,s_3) = T^{++}_{+-}(s_1,s_2,s_3)$$

We can independently try to derive the $C^p_{tu}$ crossing matrix by using the following composition for the generators of $S_3$.

$$C^p_{tu} = C^p_{st} C^p_{su} C^p_{st}, \qquad C^p_{tu} = \epsilon'^2_{23} \epsilon'_{24} \begin{pmatrix} 1 & 0 & 0 & 0 & 0 \\ 0 & 1 & 0 & 0 & 0 \\ 0 & 0 & 0 & 1 & 0 \\ 0 & 0 & 1 & 0 & 0 \\ 0 & 0 & 0 & 0 & 1 \end{pmatrix}. \tag{2.15}$$

to conclude $\epsilon_{24'} = 1$ while the phase $\epsilon'_{23}$ is undetermined. We can try to fix $\epsilon'_{23}$ in the following way. We can compare against a known amplitude to check the phase. To be precise let us compare against the explicit helicity amplitudes computed in the Euler-Heisenberg EFT, from the last equality in eqn 2.9 of [44] and tree level graviton amplitude from eqn 17 of [36], we see $\epsilon'_{23} = 1$ for both photons and graviton amplitudes[7]. For convenience we write down the crossing matrices finally

$$C^p_{st} = \begin{pmatrix} 0 & 0 & 0 & 1 & 0 \\ 0 & 1 & 0 & 0 & 0 \\ 0 & 0 & 1 & 0 & 0 \\ 1 & 0 & 0 & 0 & 0 \\ 0 & 0 & 0 & 0 & 1 \end{pmatrix}, \qquad C^p_{su} = \begin{pmatrix} 0 & 0 & 1 & 0 & 0 \\ 0 & 1 & 0 & 0 & 0 \\ 1 & 0 & 0 & 0 & 0 \\ 0 & 0 & 0 & 1 & 0 \\ 0 & 0 & 0 & 0 & 1 \end{pmatrix}. \tag{2.16}$$

One can immediately see from these crossing matrices that $T_2(s_1, s_2, s_3)$ and $T_5(s_1, s_2, s_3)$ are crossing symmetric by themselves while it takes a little bit more effort to see that $(T_1(s_1, s_2, s_3), T_3(s_1, s_2, s_3), T_4(s_1, s_2, s_3))$ transforms in a $\mathbf{3_S} = \mathbf{1_S} + \mathbf{2_M}$ (a reducible representation of dimension 3). To see that $T_2(s_1, s_2, s_3)$ and $T_5(s_1, s_2, s_3)$ are crossing symmetric, note that under $(s_1, s_2)$ and $(s_1, s_3)$ they map to themselves and since the other orbits of $S_3$ are generated by products of this transposition, all the orbits will map to themselves. However, to systematise the procedure we explain in detail the case of photons. Using the projector (A.2) we see that

$$\begin{aligned}
P_{\mathbf{1_S}}(T_2(s_1, s_2, s_3)) &= T_2(s_1, s_2, s_3), \\
P_{\mathbf{1_S}}(T_5(s_1, s_2, s_3)) &= T_5(s_1, s_2, s_3), \\
P_{\mathbf{1_S}}(T_1(s_1, s_2, s_3)) &= P_{\mathbf{1_S}}(T_3(s_1, s_2, s_3)) = P_{\mathbf{1_S}}(T_4(s_1, s_2, s_3)), \\
&= \frac{(T_1(s_1, s_2, s_3) + T_3(s_1, s_2, s_3) + T_4(s_1, s_2, s_3))}{3}, \\
P_{\mathbf{1_A}}(T_i(s_1, s_2, s_3)) &= 0.
\end{aligned} \tag{2.17}$$

This tells us that the triplet $(T_1(s_1, s_2, s_3), T_3(s_1, s_2, s_3), T_4(s_1, s_2, s_3))$ has a $\mathbf{1_S}$ part while $T_2(s_1, s_2, s_3)$ and $T_5(s_1, s_2, s_3)$ are crossing symmetric by themselves. We now want to check whether there is a $\mathbf{2_M}$ also in $(T_1(s_1, s_2, s_3), T_3(s_1, s_2, s_3), T_4(s_1, s_2, s_3))$. From (A.4), we get,

$$\begin{aligned}
P^{(1)}_{\mathbf{2_{M+}}}(T_1(s_1, s_2, s_3)) &= -2P^{(1)}_{\mathbf{2_{M+}}}(T_3(s_1, s_2, s_3)) = -2P^{(1)}_{\mathbf{2_{M+}}}(T_4(s_1, s_2, s_3)), \\
&= \frac{(2T_1(s_1, s_2, s_3) - T_3(s_1, s_2, s_3) - T_4(s_1, s_2, s_3))}{3}. \\
P^{(2)}_{\mathbf{2_{M+}}}(T_1(s_1, s_2, s_3)) &= -2P^{(2)}_{\mathbf{2_{M+}}}(T_3(s_1, s_2, s_3)) = -2P^{(2)}_{\mathbf{2_{M+}}}(T_4(s_1, s_2, s_3)), \\
&= \frac{(2T_3(s_1, s_2, s_3) - T_1(s_1, s_2, s_3) - T_4(s_1, s_2, s_3))}{3}.
\end{aligned} \tag{2.18}$$

---

[7]Note the difference in convention in defining helicity amplitudes, $A^{\text{us},\lambda_3,\lambda_4}_{\lambda_1,\lambda_2} = A^{\text{them},\lambda_3,\lambda_4,-\lambda_1,-\lambda_2}$

Therefore, we identify our crossing symmetric matrix by substituting the following sets of solutions in (2.7),

$$
\begin{aligned}
f_{\text{Sym}}^{\alpha,1}(s_1, s_2, s_3) &= T_2(s_1, s_2, s_3)\,, \\
f_{\text{Sym}}^{\alpha,2}(s_1, s_2, s_3) &= T_5(s_1, s_2, s_3)\,, \\
f_{\text{Sym}}^{\alpha,3}(s_1, s_2, s_3) &= \frac{T_1(s_1, s_2, s_3) + T_3(s_1, s_2, s_3) + T_4(s_1, s_2, s_3)}{3}\,, \\
f_{\text{Mixed+}}^{\alpha}(s_1, s_2, s_3) &= \frac{(2T_1(s_1, s_2, s_3) - T_3(s_1, s_2, s_3) - T_4(s_1, s_2, s_3))}{3}\,.
\end{aligned}
\tag{2.19}
$$

where, $\alpha \equiv \gamma, h$ for photons and gravitons respectively. Explicitly written out, the crossing symmetric photon and graviton $S$-matrices are,

$$
F_1^{\alpha}(s_1, s_2, s_3) = T_2(s_1, s_2, s_3)\,,
\tag{2.20}
$$

$$
F_2^{\alpha}(s_1, s_2, s_3) = T_1(s_1, s_2, s_3) + T_3(s_1, s_2, s_3) + T_4(s_1, s_2, s_3)\,,
\tag{2.21}
$$

$$
F_3^{\alpha}(s_1, s_2, s_3) = T_5(s_1, s_2, s_3)\,,
\tag{2.22}
$$

$$
F_4^{\alpha}(s_1, s_2, s_3) = \frac{f_{\text{Mixed+}}^{\alpha}(s_3, s_1, s_2)(s_3 + s_2 - 2s_1) - f_{\text{Mixed+}}^{\alpha}(s_1, s_2, s_3)(s_1 + s_2 - 2s_3)}{3(s_1 - s_2)(s_2 - s_3)(s_3 - s_1)}\,,
\tag{2.23}
$$

$$
F_5^{\alpha}(s_1, s_2, s_3) = \frac{f_{\text{Mixed+}}^{\alpha}(s_1, s_2, s_3)(s_1^2 + s_2^2 - 2s_3^2) - f_{\text{Mixed+}}^{\alpha}(s_3, s_1, s_2)(s_2^2 + s_3^2 - 2s_1^2)}{3(s_1 - s_2)(s_2 - s_3)(s_3 - s_1)}\,.
\tag{2.24}
$$

### 2.1.2 $\mathcal{P}$ violating theories

In this subsubsection we consider Parity violating (and hence Time-reversal violating theories) theories where we do not impose the condition (2.9). As a result, the independent helicity amplitudes are,

$$
\begin{aligned}
&T_1(s_1, s_2, s_3) = T_{++}^{++}(s_1, s_2, s_3), \quad T_2(s_1, s_2, s_3) = T_{++}^{--}(s_1, s_2, s_3), \quad T_3(s_1, s_2, s_3) = T_{+-}^{+-}(s_1, s_2, s_3) \\
&T_4(s_1, s_2, s_3) = T_{+-}^{-+}(s_1, s_2, s_3), \quad T_5(s_1, s_2, s_3) = T_{++}^{+-}(s_1, s_2, s_3), \quad T_2'(s_1, s_2, s_3) = T_{--}^{++}(s_1, s_2, s_3) \\
&T_5'(s_1, s_2, s_3) = T_{+-}^{++}(s_1, s_2, s_3)\,.
\end{aligned}
\tag{2.25}
$$

The crossing matrices are modified to

$$
C_{st}^{pv} = \begin{pmatrix} 0 & 0 & 0 & 1 & 0 & 0 & 0 \\ 0 & 1 & 0 & 0 & 0 & 0 & 0 \\ 0 & 0 & 1 & 0 & 0 & 0 & 0 \\ 1 & 0 & 0 & 0 & 0 & 0 & 0 \\ 0 & 0 & 0 & 0 & 1 & 0 & 0 \\ 0 & 0 & 0 & 0 & 0 & 1 & 0 \\ 0 & 0 & 0 & 0 & 0 & 0 & 1 \end{pmatrix}, \quad C_{su}^{pv} = \begin{pmatrix} 0 & 0 & 1 & 0 & 0 & 0 & 0 \\ 0 & 1 & 0 & 0 & 0 & 0 & 0 \\ 1 & 0 & 0 & 0 & 0 & 0 & 0 \\ 0 & 0 & 0 & 1 & 0 & 0 & 0 \\ 0 & 0 & 0 & 0 & 1 & 0 & 0 \\ 0 & 0 & 0 & 0 & 0 & 1 & 0 \\ 0 & 0 & 0 & 0 & 0 & 0 & 1 \end{pmatrix}.
\tag{2.26}
$$

The new objects we need to consider are $T_2'$ and $T_5'$, which are crossing symmetric by themsleves.

$$
\begin{aligned}
P_{\mathbf{1_S}}(T_2'(s_1, s_2, s_3)) &= T_2'(s_1, s_2, s_3)\,, \\
P_{\mathbf{1_S}}(T_5'(s_1, s_2, s_3)) &= T_5'(s_1, s_2, s_3)\,.
\end{aligned}
\tag{2.27}
$$

Therefore we identify our crossing symmetric matrix by substituting the following sets of solutions in (2.7),

$$
\begin{aligned}
\tilde{f}_{\text{Sym}}^{\alpha,1}(s_1, s_2, s_3) &= T_2(s_1, s_2, s_3), \\
\tilde{f}_{\text{Sym}}^{\alpha,2}(s_1, s_2, s_3) &= T_5(s_1, s_2, s_3), \\
\tilde{f}_{\text{Sym}}^{\alpha,3}(s_1, s_2, s_3) &= \frac{T_1(s_1, s_2, s_3) + T_3(s_1, s_2, s_3) + T_4(s_1, s_2, s_3)}{3}, \\
\tilde{f}_{\text{Sym}}^{\alpha,4}(s_1, s_2, s_3) &= T_2'(s_1, s_2, s_3), \\
\tilde{f}_{\text{Sym}}^{\alpha,5}(s_1, s_2, s_3) &= T_5'(s_1, s_2, s_3), \\
\tilde{f}_{\text{Mixed+}}^{\alpha}(s_1, s_2, s_3) &= \frac{(2T_1(s_1, s_2, s_3) - T_3(s_1, s_2, s_3) - T_4(s_1, s_2, s_3))}{3}.
\end{aligned}
\tag{2.28}
$$

where, $\alpha \equiv \gamma, g$ for photons and gravitons respectively. Explicitly written out, the crossing symmetric photon and graviton s-matrices are,

$$
\tilde{F}_1^{\alpha}(s_1, s_2, s_3) = T_2(s_1, s_2, s_3),
\tag{2.29}
$$

$$
\tilde{F}_2^{\alpha}(s_1, s_2, s_3) = T_1(s_1, s_2, s_3) + T_3(s_1, s_2, s_3) + T_4(s_1, s_2, s_3),
\tag{2.30}
$$

$$
\tilde{F}_3^{\alpha}(s_1, s_2, s_3) = T_5(s_1, s_2, s_3),
\tag{2.31}
$$

$$
\tilde{F}_4^{\alpha}(s_1, s_2, s_3) = \frac{f_{\text{Mixed+}}^{\alpha}(s_3, s_1, s_2)(s_3 + s_2 - 2s_1) - f_{\text{Mixed+}}^{\alpha}(s_1, s_2, s_3)(s_1 + s_2 - 2s_3)}{3(s_1 - s_2)(s_2 - s_3)(s_3 - s_1)},
\tag{2.32}
$$

$$
\tilde{F}_5^{\alpha}(s_1, s_2, s_3) = \frac{f_{\text{Mixed+}}^{\alpha}(s_1, s_2, s_3)(s_1^2 + s_2^2 - 2s_3^2) - f_{\text{Mixed+}}^{\alpha}(s_3, s_1, s_2)(s_2^2 + s_3^2 - 2s_1^2)}{3(s_1 - s_2)(s_2 - s_3)(s_3 - s_1)},
\tag{2.33}
$$

$$
\tilde{F}_6^{\alpha}(s_1, s_2, s_3) = T_2'(s_1, s_2, s_3),
\tag{2.34}
$$

$$
\tilde{F}_7^{\alpha}(s_1, s_2, s_3) = T_5'(s_1, s_2, s_3).
\tag{2.35}
$$

We note that the crossing equations are consistent with the photon module classification done in [45]. In [45], it was found that there is one parity even module transforming in a **3**, two parity even **1$_{\mathbf{S}}$** module and two parity odd **1$_{\mathbf{S}}$** module. It is satisfying to see that the degrees of freedom encoded in crossing in the two different approaches nicely match.

## 2.2 Massive Majorana fermions

Let us now consider the scattering amplitude of four massive Majorana fermions in parity conserving theory. The five independent helicity structures are the following

$$
\begin{aligned}
\Phi_1(s_1, s_2, s_3) &= T_{++}^{++}(s_1, s_2, s_3), &\quad \Phi_2(s_1, s_2, s_3) &= T_{++}^{--}(s_1, s_2, s_3), &\quad \Phi_3(s_1, s_2, s_3) &= T_{+-}^{+-}(s_1, s_2, s_3) \\
\Phi_4(s_1, s_2, s_3) &= T_{+-}^{-+}(s_1, s_2, s_3), &\quad \Phi_5(s_1, s_2, s_3) &= T_{++}^{+-}(s_1, s_2, s_3).
\end{aligned}
\tag{2.36}
$$

Further one can separate out the kinematical singularities and branch cuts to define the improved amplitudes $H_I(s_1, s_2, s_3)$ such that [41],

$$
\phi_I(s_1, s_2, s_3) = \sum_{J=1}^{5} M_{IJ}^{-1} H_J(s_1, s_2, s_3),
\tag{2.37}
$$

where $M$ matrix is defined as follows,

$$M = \begin{pmatrix} \frac{4}{s_1 - \frac{8m^2}{3}} & \frac{-4}{s_1 - \frac{8m^2}{3}} & \frac{2(1 - \frac{s_2 + 4m^2/3}{s_3 + 4m^2/3})}{s_1 - 8m^2/3} & \frac{2(1 - \frac{s_3 + 4m^2/3}{s_2 + 4m^2/3})}{s_1 - 8m^2/3} & \frac{2(s_1 + 16m^2/3)(s_2 - s_3)}{m(s_1 - 8m^2/3)\sqrt{(s_1 + 4m^2/3)(s_2 + 4m^2/3)(s_3 + 4m^2/3)}} \\ 0 & 0 & \frac{2}{s_3 + 4m^2/3} & \frac{-2}{s_2 + 4m^2/3} & -\frac{8m}{\sqrt{(s_1 + 4m^2/3)(s_2 + 4m^2/3)(s_3 + 4m^2/3)}} \\ 0 & 0 & \frac{2}{s_3 + 4m^2/3} & \frac{-2}{s_2 + 4m^2/3} & -\frac{2s}{m\sqrt{(s_1 + 4m^2/3)(s_2 + 4m^2/3)(s_3 + 4m^2/3)}} \\ 0 & 0 & \frac{2}{s_3 + 4m^2/3} & \frac{2}{s_2 + 4m^2/3} & 0 \\ -\frac{4}{s_1 + 4m^2/3} & -\frac{4}{s_1 + 4m^2/3} & \frac{2}{s_3 + 4m^2/3} + \frac{4}{s_1 + 4m^2/3} & \frac{2}{s_2 + 4m^2/3} + \frac{4}{s_1 + 4m^2/3} & \frac{2(s_2 - s_3)}{m\sqrt{(s_1 + 4m^2/3)(s_2 + 4m^2/3)(s_3 + 4m^2/3)}} \end{pmatrix}$$

$$(2.38)$$

Crossing symmetry is imposed by the following two crossing matrices,

$$\tilde{C}^f_{st} = \begin{pmatrix} -\frac{1}{4} & -1 & \frac{3}{2} & 1 & -\frac{1}{4} \\ -\frac{1}{4} & \frac{1}{2} & 0 & \frac{1}{2} & \frac{1}{4} \\ \frac{1}{4} & 0 & \frac{1}{2} & 0 & \frac{1}{4} \\ \frac{1}{4} & \frac{1}{2} & 0 & \frac{1}{2} & -\frac{1}{4} \\ -\frac{1}{4} & 1 & \frac{3}{2} & -1 & -\frac{1}{4} \end{pmatrix}, \qquad \tilde{C}^f_{su} = \begin{pmatrix} -\frac{1}{4} & 1 & -\frac{3}{2} & 1 & -\frac{1}{4} \\ \frac{1}{4} & \frac{1}{2} & 0 & -\frac{1}{2} & -\frac{1}{4} \\ -\frac{1}{4} & 0 & \frac{1}{2} & 0 & -\frac{1}{4} \\ \frac{1}{4} & -\frac{1}{2} & 0 & \frac{1}{2} & -\frac{1}{4} \\ -\frac{1}{4} & -1 & -\frac{3}{2} & -1 & -\frac{1}{4} \end{pmatrix} \qquad (2.39)$$

The analysis for massive fermions is a bit more involved. Using the projectors defined in (A.2), we find

$$P_{\mathbf{1_S}}(H_1(s_1, s_2, s_3)) = P_{\mathbf{1_S}}(H_4(s_1, s_2, s_3)) = P_{\mathbf{1_S}}(H_5(s_1, s_2, s_3))$$
$$= \frac{(H_1(s_1, s_2, s_3) + 4H_4(s_1, s_2, s_3) - H_5(s_1, s_2, s_3))}{6},$$
$$P_{\mathbf{1_S}}(H_2(s_1, s_2, s_3)) = P_{\mathbf{1_S}}(H_3(s_1, s_2, s_3)) = 0,$$
$$P_{\mathbf{1_A}}(H_i(s_1, s_2, s_3)) = 0.$$
$$(2.40)$$

This implies that the we have an irrep that transforms in an $\mathbf{1_S}$ and none in $\mathbf{1_A}$. We now use the projector for the mixed symmetry to evaluate

$$P^{(1)}_{\mathbf{2_{M+}}}(H_1(s_1, s_2, s_3)) = \frac{1}{6}(5H_1(s_1, s_2, s_3) - 4H_4(s_1, s_2, s_3) + H_5(s_1, s_2, s_3)),$$
$$P^{(2)}_{\mathbf{2_{M+}}}(H_1(s_1, s_2, s_3)) = \frac{1}{12}(-5H_1(s_1, s_2, s_3) + 12H_2(s_1, s_2, s_3) - 18H_3(s_1, s_2, s_3) + 4H_4(s_1, s_2, s_3)$$
$$- H_5(s_1, s_2, s_3)),$$
$$P^{(1)}_{\mathbf{2_{M+}}}(H_4(s_1, s_2, s_3)) = \frac{1}{6}(-H_1(s_1, s_2, s_3) + 2H_4(s_1, s_2, s_3) + H_5(s_1, s_2, s_3)),$$
$$P^{(2)}_{\mathbf{2_{M+}}}(H_4(s_1, s_2, s_3)) = \frac{1}{12}(H_1(s_1, s_2, s_3) - 6H_2(s_1, s_2, s_3) - 2H_4(s_1, s_2, s_3) - H_5(s_1, s_2, s_3)).$$
$$(2.41)$$

Rest of the $\mathbf{2_{M+/-}}$ projections are either zero or a linear combinations of these. Hence the independent data that can be used in (2.7) are,

$$\begin{aligned} f^\psi_{\text{Sym}}(s_1, s_2, s_3) &= \frac{(H_1(s_1, s_2, s_3) + 4H_4(s_1, s_2, s_3) - H_5(s_1, s_2, s_3))}{6} \\ f^{\psi,1}_{\text{Mixed+}}(s_1, s_2, s_3) &= \frac{1}{6}(5H_1(s_1, s_2, s_3) - 4H_4(s_1, s_2, s_3) + H_5(s_1, s_2, s_3)) \\ f^{\psi,2}_{\text{Mixed+}}(s_1, s_2, s_3) &= \frac{1}{6}(-H_1(s_1, s_2, s_3) + 2H_4(s_1, s_2, s_3) + H_5(s_1, s_2, s_3)). \end{aligned}$$
$$(2.42)$$

Explicitly written out, the crossing symmetric fermion S-matrices are,:

$$\Psi_1(s_1, s_2, s_3) = (H_1(s_1, s_2, s_3) + 4H_4(s_1, s_2, s_3) - H_5(s_1, s_2, s_3)), \qquad (2.43)$$

$$\Psi_2(s_1, s_2, s_3) = \frac{f_{\text{Mixed}+}^{\psi,1}(s_3, s_1, s_2)(s_3 + s_2 - 2s_1) - f_{\text{Mixed}+}^{\psi,1}(s_1, s_2, s_3)(s_1 + s_2 - 2s_3)}{3(s_1 - s_2)(s_2 - s_3)(s_3 - s_1)}, \qquad (2.44)$$

$$\Psi_3(s_1, s_2, s_3) = \frac{f_{\text{Mixed}+}^{\psi,1}(s_1, s_2, s_3)(s_1^2 + s_2^2 - 2s_3^2) - f_{\text{Mixed}+}^{\psi,1}(s_3, s_1, s_2)(s_2^2 + s_3^2 - 2s_1^2)}{3(s_1 - s_2)(s_2 - s_3)(s_3 - s_1)}. \qquad (2.45)$$

$$\Psi_4(s_1, s_2, s_3) = \frac{f_{\text{Mixed}+}^{\psi,2}(s_3, s_1, s_2)(s_3 + s_2 - 2s_1) - f_{\text{Mixed}+}^{\psi,2}(s_1, s_2, s_3)(s_1 + s_2 - 2s_3)}{3(s_1 - s_2)(s_2 - s_3)(s_3 - s_1)}, \qquad (2.46)$$

$$\Psi_5(s_1, s_2, s_3) = \frac{f_{\text{Mixed}+}^{\psi,2}(s_1, s_2, s_3)(s_1^2 + s_2^2 - 2s_3^2) - f_{\text{Mixed}+}^{\psi,2}(s_3, s_1, s_2)(s_2^2 + s_3^2 - 2s_1^2)}{3(s_1 - s_2)(s_2 - s_3)(s_3 - s_1)}. \qquad (2.47)$$

# 3 Crossing symmetric dispersion relation: Overview

In the previous section, we constructed various fully crossing symmetric amplitudes, i.e., amplitudes invariant under $S_3$, the group of permutations of $(s_1, s_2, s_3)$. Now, we will discuss a manifestly crossing symmetric dispersive representation for such an amplitude. Such a representation was first derived in [28]. Recently, this representation was explored in [29] in the context of EFT bootstrap. In this section, we will review this dispersion relation and its multi-faceted consequences, which were explored recently in [25, 29, 43][8] for scalar amplitudes. We will present the discussion in such a fashion which generalizes naturally to helicity amplitudes that we will be considering in the present work for dealing with spinning particles.

Let us consider a $S_3$-invariant 'amplitude' associated[9] with scattering of identical particles $\mathcal{M}(s, t, u)$, with $s + t + u = 4m^2 = \mu$, $m$ being the mass of the scattering particles. The amplitude is known/assumed to satisfy the following two crucial properties.

I. We assume that the amplitude is analytic in some domain[10] $\mathcal{D} \subset \mathbb{C}^2$, which includes the physical domains of all the three channels. For massive theories, such domains (e.g. enlarged Martin domain [47]) have been established rigorously from axiomatic field theory considerations. For massless theories, even though such domains are not established within the rigorous framework of axiomatic field theory, they can be argued physically in general. Thus we will assume the existence of such domains, to begin with.

II. The amplitude is Regge bounded in all the three channels. While for massive theories this is established rigorously from axiomatic field theory, for massless theories this is a working assumption which we will make. Thus, for example, fixed $t$ Regge-boundedness reads

$$\mathcal{M}(s, t) = o(s^2) \quad \text{for } |s| \to \infty, \quad t \text{ fixed}, \quad (s, t) \in \mathcal{D}. \qquad (3.1)$$

This is equivalent to the amplitude admitting a twice subtracted fixed-transfer (for example, fixed-$t$) dispersive representation.

## 3.1 Massive amplitudes

In order to write a manifestly crossing symmetric dispersion relation, one first introduces a certain parametrisation [28, 29] for the Mandelstam variables $\{s_1 = s - \mu/3, \ s_2 = t - \mu/3, \ s_3 = u - \mu/3\}$:

$$s_k(a, z) = a \left[ 1 - \frac{(z - z_k)^3}{z^3 - 1} \right], \quad a \in \mathbb{R}, \quad k = 1, 2, 3. \qquad (3.2)$$

---

[8]See also [30, 46] for crossing symmetric/ Anti-Symmetric kernels in context of CFT bootstrap.

[9]There can be multiple such amplitudes associated with a given scattering when the particles have extra quantum numbers such as spin, isospin e.t.c

[10]We are considering both $s$ and $t$ as complex variables.

Here $\{z_k\}$ are the cube-roots of unity. $\tilde{z} := z^3, a$ are crossing symmetric variables. We also introduce the following crossing symmetric combinations, $x := -(s_1 s_2 + s_2 s_3 + s_3 s_1) = \frac{-27 a^2 z^3}{(z^3-1)^2}$, $y := -(s_1 s_2 s_3) = \frac{-27 a^3 \tilde{z}^3}{(z^3-1)^2}$ such that $a = y/x$. With these parametrizations, as shown by [28], one can write the following dispersive representation of the amplitude $\mathcal{M}$ with a *manifestly crossing-symmetric kernel*:

$$\mathcal{M}(s_1, s_2) = \alpha_0 + \frac{1}{\pi} \int_{M^2}^{\infty} \frac{ds_1'}{s_1'} \mathcal{A}\left(s_1'; s_2^{(+)}\left(s_1', a\right)\right) H\left(s_1'; s_1, s_2, s_3\right) , \qquad (3.3)$$

where $\mathcal{A}(s_1; s_2)$ is the s-channel discontinuity, $s_2^{(+)}(s_1', a) = -\frac{s_1'}{2}\left[1 - \left(\frac{s_1' + 3a}{s_1' - a}\right)^{1/2}\right]$ and the kernel is given by

$$
\begin{aligned}
H\left(s_1'; s_1, s_2, s_3\right) &= \left[\frac{s_1}{(s_1' - s_1)} + \frac{s_2}{(s_1' - s_2)} + \frac{s_3}{(s_1' - s_3)}\right] \\
&= \frac{27 a^2 (3a - 2s_1')}{\left(-27 a^3 + 27 a^2 s_1' + s_1'^3 \left(\frac{-27 a^2}{x}\right)\right)} .
\end{aligned} \qquad (3.4)
$$

## 3.2   Massless theories: EFT amplitudes

For massless theories, we will consider crossing-symmetric dispersion relation for the amplitudes in the sense of effective field theories (EFT) as detailed below. One can write the following (twice subtracted) fixed $t$ dispersion relation for a massless amplitude [5]

$$\frac{\mathcal{M}(s, t)}{s(s+t)} = \int_{-\infty}^{\infty} \frac{ds'}{\pi (s' - s)} \mathrm{Im}\left[\frac{\mathcal{M}(s', t)}{s'(s' + t)}\right], \qquad (t < 0, \ s \notin \mathbb{R}), \qquad (3.5)$$

where the subtraction points are chosen to be $s = 0$ and $s = -t, t < 0$. Now, the amplitude can be divided into two parts, the high energy amplitude $\mathcal{M}_{\text{high}}$ and the low energy amplitude $\mathcal{M}_{\text{low}}$. The high energy amplitude $\mathcal{M}_{\text{high}}$ admits an 'effective (fixed-transfer) dispersion relation' with two subtractions. For example, the fixed $t$ effective dispersion relation is of the form

$$\frac{\mathcal{M}_{\text{high}}(s, t)}{s(s+t)} = \int_{M^2}^{\infty} \frac{ds'}{\pi}\left(\frac{1}{s' - s} + \frac{1}{s' + s + t}\right) \mathrm{Im}\left[\frac{\mathcal{M}_{\text{high}}(s', t)}{s'(s' + t)}\right]. \qquad (3.6)$$

Here $M^2$ is some UV cut-off such that the physics beyond this scale is unknown a-priori to us. The low energy amplitude $\mathcal{M}_{\text{low}}$ is to be understood in the sense of effective field theory amplitude. The dispersion relation for the full amplitude $\mathcal{M}$, (3.5), relates $\mathcal{M}_{\text{high}}$ to this EFT amplitude.

Now, the amplitudes are all crossing symmetric. Therefore, we can write a crossing-symmetric dispersion relation for $\mathcal{M}_{high}$ [29] similar to (3.8)

$$\mathcal{M}_{\text{high}}(s_1, s_2) = \alpha_0 + \frac{1}{\pi} \int_{M^2}^{\infty} \frac{ds_1'}{s_1'} \mathcal{A}_{\text{high}}\left(s_1'; s_2^{(+)}\left(s_1', a\right)\right) H\left(s_1'; s_1, s_2, s_3\right), \qquad (3.7)$$

with the kernel as in (3.4) and $\mathcal{A}_{\text{high}}$ being the absorptive part of the high energy amplitude $\mathcal{M}_{\text{high}}$.

In summary, we can write the crossing-symmetric dispersive representation for the amplitude ( in the sense of EFT when required ) as

$$\mathcal{M}(s_1, s_2) = \alpha_0 + \frac{1}{\pi} \int_{\Lambda_0}^{\infty} \frac{ds_1'}{s_1'} \mathcal{A}\left(s_1'; s_2^{(+)}\left(s_1', a\right)\right) H\left(s_1'; s_1, s_2, s_3\right), \qquad (3.8)$$

where $\Lambda_0 = 2\mu/3$ for massive theories and $\Lambda_0 = M^2$ (the UV cut-off) for massless amplitudes and the partial wave decomposition reads

$$\mathcal{A}\left(s_1'; s_2^{(+)}\left(s_1', a\right)\right) = \Phi(s_1) \sum_{J=0}^{\infty} (2J + 2\alpha) a_J(s_1) C_J^{\left(\frac{d-3}{2}\right)}\left(\sqrt{\xi(s_1, a)}\right) \qquad (3.9)$$

where $\xi(s_1, a) = \xi_0 + 4\xi_0 \left(\frac{a}{s_1 - a}\right)$ and $\xi_0 = \frac{s_1^2}{(s_1 - \Lambda_0)^2}$ for massive theories while $\xi_0 = 1$ for massless theories and $a_\ell(s_1)$ is the spectral density which is defined as the imaginary part of the partial wave amplitude.

## 3.3  Wilson coefficients and locality constraints

In this section we outline two central ingredients needed for our bounds. Let us first review the case of massive scalar EFTs. The crossing symmetric amplitude, pole subtracted if required, admits a crossing symmetric double power series can be expanded in terms of crossing-symmetric variables $x := -(s_1 s_2 + s_2 s_3 + s_3 s_1)$, $y := -(s_1 s_2 s_3)$:

$$\mathcal{M}(s_1, s_2) = \sum_{p,q=0}^\infty \mathcal{W}_{p,q}\, x^p y^q \qquad (3.10)$$

This is equivalent to a low-energy (EFT) expansion for the amplitude. The coefficients $\{\mathcal{W}_{p,q}\}$ are themselves or related to the Wilson coefficients appearing in the effective Lagrangian of a theory. Thus, these coefficients parametrize the space of EFTs. These coefficients can be obtained from the amplitude via the inversion formula [29]

$$\mathcal{W}_{n-m,m} = \int_{\Lambda_0}^\infty \frac{ds_1}{2\pi s_1^{2n+m+1}} \Phi(s_1) \sum_{J=0}^\infty (2J + 2\alpha)\, a_J(s_1)\, \mathcal{B}_{n,m}^{(J)}(s_1), \qquad (3.11)$$

with

$$\mathcal{B}_{n,m}^J(s_1) = 2 \sum_{j=0}^m \frac{(-1)^{1-j+m} p_J^{(j)}(\xi_0)\,(4\xi_0)^j\,(3j - m - 2n)\Gamma(n-j)}{j!(m-j)!\Gamma(n-m+1)} \qquad \xi_0 := \frac{s_1^2}{(s_1 - 2\mu/3)^2} \qquad (3.12)$$

Here $\{a_J\}$ are the *spectral functions* which appear as coefficients in partial wave expansion of the absorptive parts $\mathcal{A}$. The functions $\{p_J^{(j)}(\xi_0)\}$ are derivatives of Gegenbauer polynomials $C_J^{(\alpha)}$ for scalars

$$p_J^{(j)}(\xi_0) := \left. \frac{\partial^j C_J^{(\alpha)}(\sqrt{z})}{\partial z^j} \right|_{z=\xi_0}. \qquad (3.13)$$

The inversion formula leads to two kinds of constraints that play central role in our subsequent analysis. Let us briefly review these. For more details, the readers are encouraged to look into [29, 30].

**Locality constraints**

The first type of constraint that we will consider is related to the locality. In any local theory, a crossing symmetric amplitude admits a low energy expansion of the form (3.10). In particular, there should not be any negative power of $x$. However, this is not manifest at the level of the crossing symmetric dispersion relation since (3.11) is valid for both $n \leq m$ and $n > m$, the latter leading to the negative power of $x$. As explained in [29, 30] this is the price one has to pay for making crossing symmetry manifest. Fixed-transfer dispersion relations are manifestly local, but crossing symmetry has to be imposed as an additional constraint, whereas in the crossing symmetric dispersion relation, by making crossing symmetry manifest, the locality is lost and has to be imposed. The equivalence of these two approaches was argued in [30].

Thus one needs to *impose locality by demanding*

$$\mathcal{W}_{n-m,m} = 0, \quad \text{for} \quad n < m. \qquad (3.14)$$

This gives rise to an infinite number of constraints on the partial wave coefficients $\{a_J\}$ known as *locality constraints*, which are, in general, linear combinations of the null constraints [5]. In principle,

solving these infinite number of constraints can drastically restrict the space of allowed theories. However, even solving a finite number of such constraints give valuable information that we will see later.

**Massless spinning particles**

For massless spinning particles, the decomposition of the amplitude into partial waves remains more or less unchanged with the technical difference because instead of Gegenbauer polynomials, we have Wigner-$d$ functions in the expansion. From section 2, we know how to construct the crossing symmetric amplitudes given the linearly independent basis of helicity amplitudes for various massless and massive spinning particles. The crossing symmetric decomposition (3.8) is therefore modified to be,

$$\mathcal{M}^{\text{spin}}(s_1, s_2) = \alpha_0 + \frac{1}{\pi} \int_{\Lambda_0}^{\infty} \frac{ds_1'}{s_1'} \mathcal{A}^{\text{spin}}\left(s_1'; s_2^{(+)}\left(s_1', a\right)\right) H\left(s_1'; s_1, s_2, s_3\right). \tag{3.15}$$

If the crossing symmetric amplitude is given by $\sum_i \beta_i T_i$, where $T_i$ denote linearly independent helicity amplitudes and $\beta_i$ are some numbers,

$$\mathcal{A}^{\text{spin}}\left(s_1; s_2^{(+)}\left(s_1, a\right)\right) = \sum_i \beta_i \, \Phi(s_1) \sum_{J=0}^{\infty} (2J + 2\alpha) \, a_J^i(s_1) \, f_i(d_{m,n}^J(\sqrt{\xi(s_1, a)})) \tag{3.16}$$

where $f_i(d_{m,n}^J(\sqrt{\xi(s_1, a)}))$ denote the particular linear combinations of Wigner-$d$ functions that appear for $i$th helicity amplitude and $a_J^i(s_1)$ now denotes the imaginary part of the partial waves of the particular helicity amplitude $T_i = T_{\lambda_1, \lambda_2}^{\lambda_3, \lambda_4}\left(s_1; s_2^{(+)}\left(s_1, a\right)\right)$. In our conventions, a helicity amplitude $T_i$ admits the following Wigner-$d$ matrix decomposition,

$$f_i\left(d_{m,n}^J(\sqrt{\xi(s_1, a)})\right) = d_{\lambda_1 - \lambda_2, \lambda_3 - \lambda_4}^J(\sqrt{\xi(s_1, a)}), \qquad J \geq |\lambda_1 - \lambda_2|. \tag{3.17}$$

The restriction over the spin has been explained in Appendix C. Consequently, the inversion formula gets modified to,

$$\mathcal{W}_{n-m,m}^{\text{spin}} = \int_{\Lambda_0}^{\infty} \frac{ds_1}{2\pi s_1^{2n+m+1}} \Phi(s_1) \sum_i \beta_i \sum_{J=0}^{\infty} (2J + 2\alpha) \, a_J^i(s_1) \, \hat{\mathcal{B}}_{n,m}^{(J),i}(s_1), \tag{3.18}$$

with

$$\hat{\mathcal{B}}_{n,m}^{J,i}(s_1) = 2 \sum_{j=0}^{m} \frac{(-1)^{1-j+m} p_J^{(j),i}(1)(4)^j (3j - m - 2n) \Gamma(n - j)}{j!(m-j)!\Gamma(n-m+1)} \qquad p_J^{(j),i}(1) := \frac{\partial^j f_i\left(d_{m,n}^J(\sqrt{z})\right)}{\partial z^j}\bigg|_{z=1}, \tag{3.19}$$

The locality constraints are now modified to be,

$$\mathcal{W}_{n-m,m}^{\text{spin}} = 0, \quad \text{for} \quad n < m. \tag{3.20}$$

## 3.4 $PB_C$ constraints

In order to get bounds, we would like to show that the expression on the RHS of (3.11) (and also (3.18)) or a linear combination of them must be of definite sign. We identify the main characters of this analysis.

- Unitarity translates to positivity conditions on the spectral functions $\{a_J\}$ as reviewed in appendix C.

- We have $\xi_0 \geq 1$ for the entire range of $s_1$ integration in the inversion formula while $\xi_0 = 1$ identically for massless scalars ((3.12)). The functions $p_J^{(j)}(\xi_0)$ are positive due to the fact that the Gegenbauer polynomials, and its derivatives, are positive for arguments larger than unity.

From these conditions, explicitly it follows[11]

$$\mathcal{W}_{n,0} \geq 0. \tag{3.21}$$

More generally, this positivity property does not hold because $\mathcal{B}_{n,m}^{(J)}(s_1)$ (see (3.11)) control the sign of any term in $J$-expansion of the inversion formula. In that case one can ask whether taking suitable linear combinations of $\mathcal{B}_{n,m}^{(J)}(s_1)$s can restore the positivity. The answer turns out to be yes and one obtains [29]

$$
\begin{aligned}
\sum_{r=0}^{m} \chi_n^{(r,m)}(\Lambda_0^2)\mathcal{W}_{n-r,r} &\geq 0 \\
0 \leq \mathcal{W}_{n,0} &\leq \frac{\mathcal{W}_{n-1,0}}{(\Lambda_0)^2}\,, \qquad\qquad n \geq 2\,.
\end{aligned} \tag{3.22}
$$

The coefficient functions $\{\chi_n^{(r,m)}(\Lambda_0^2)\}$ satisfy the recursion relation:

$$\chi_n^{(m,m)}(\Lambda_0^2) = 1, \qquad \chi_n^{(r,m)}(\Lambda_0^2) = \sum_{j=r+1}^{m} (-1)^{j+r+1} \chi_n^{(j,m)}(\Lambda_0^2) \frac{\mathfrak{U}_{n,j,r}^{(\alpha)}(\Lambda_0^2)}{\mathfrak{U}_{n,r,r}^{(\alpha)}(\Lambda_0^2)} \tag{3.23}$$

with

$$\mathfrak{U}_{n,m,k}^{(\alpha)}(s_1) = \sum_{k=0}^{m} \frac{\sqrt{16\xi_0}^{-k}(\alpha)_k(m+2n-3j)\Gamma(n-j)\Gamma(2j-k)}{s_1^{m+2n}\Gamma(k)j!(m-j)!(j-k)!(n-m)!}. \tag{3.24}$$

The conditions (3.22) are the so-called *Positivity conditions*. In short we will call them $PB_C$.

### Massless spinning particles

For massless spinning particles, the conditions for positivity are modified as follows.

- Recall that based on the construction outlined in 2, our crossing symmetric amplitudes are linear combination of helicity amplitudes. The spectral functions, therefore, need to appear in the right combinations amenable to positivity constraints. Unlike the scalar case, spectral functions of all the helicity amplitudes do not obey positivity conditions but instead are constrained by unitarity considerations in a way that certain specific linear combinations are positive (See appendix C for details). In subsequent sections, we will see how it is achieved in our crossing symmetric amplitudes.

- For helicity amplitudes, we obtain Wigner-$d$ functions in the partial wave decomposition with the precise form given by (3.16) and (3.17). One can check that for the relevant helicity amplitude basis we have, and the linear combination in which they appear for our crossing symmetric amplitudes, relevant linear combinations of Wigner-$d$ function and its derivatives are positive for $\xi_0 = 1$.

Considering these points, just like the scalar case, we would like to construct a linear combination of $W_{n-m,m}^{\mathrm{spin}}$, which is positive. It will turn out that the scalar ansatz suffices for the cases we consider. The structural reason which allows us to do this will also become clear as we work out the relevant examples in subsections 5.1 and 6.1.

---

[11]These conditions can also be shown to arise from the $TR_U$ inequalities discussed next, on Taylor expanding those conditions around $a = 0$ [29].

## 3.5 Typically-realness and Low Spin Dominance: $TR_U$

The crossing-symmetric dispersive representation (3.8) uncovers an interesting connection between scattering amplitudes and the mathematical discipline of geometric function theory (GFT) [25]. In particular, the crossing symmetric dispersive representation (3.8) can be cast into what is known as *Robertson integral* which enables one to establish typical realness properties of the amplitude in the variable $\tilde{z} \equiv z^3$ for a certain range of $a$. This further enables the application of GFT techniques to bound the coefficients $\{\mathcal{W}_{n,m}\}$.

### 3.5.1 Typically real functions:

A function $f : \mathbb{C} \to \mathbb{C}$ is defined to be *typically real* on a domain $\mathcal{D} \in \mathbb{C}$ containing segments of real axis, if it is real on these segments and satisfies

$$\mathrm{Im}\,[f(z)]\,\mathrm{Im}[z] \geq 0, \quad \mathrm{Im}[z] \neq 0, \ z \in \mathcal{D}. \tag{3.25}$$

For our analysis, we will be interested in a particular class of typically real functions known as $TR$. A function $f(z) \in TM$ is analytic and typically real in the unit disc $\Delta := \{z \in \mathbb{C} : |z| < 1\}$ and admits Taylor series of the form

$$f(z) = z + \sum_{n=2}^{\infty} c_n z^n. \tag{3.26}$$

The coefficients $\{c_n\}$ are real which follows from the definition. Such functions can be represented by Stieltjes integrals known as *Robertson integrals*. A function $f(z)$ regular in $\Delta$ belongs to the class $TR$ *if and only if* it can be represented as the Robertson integral

$$f(z) = \int_{-1}^{1} d\mu(\xi)\,\frac{z}{1 - 2\xi z + z^2}, \tag{3.27}$$

where $\mu(\xi)$ is a non-decreasing function on $\xi \in [-1, 1]$ and satisfying $\mu(1) - \mu(-1) = 1$.

### 3.5.2 Robertson form of dispersion integral

Let us now chalk out how to establish typical realness property of the amplitude [25]. Defining

$$\xi := 1 + \frac{27a^2}{2(s_1')^3}(a - s_1'), \qquad d\mu(\xi) := \frac{\mathcal{A}(\xi, s_2(\xi, a))\,d\xi}{\int_{-1}^{1} d\xi \mathcal{A}(\xi, s_2(\xi, a))}, \tag{3.28}$$

one can formally cast the dispersion integral into

$$\widetilde{\mathcal{M}}(\tilde{z}, a) := \frac{\mathcal{M}(\tilde{z}, a) - \alpha_0}{\frac{2}{\pi}\int_{-1}^{1} d\xi \mathcal{A}(\xi, s_2(\xi, a))} = \int_{-1}^{1} d\mu(\xi)\,\frac{\tilde{z}}{1 - 2\xi\tilde{z} + \tilde{z}^2} \tag{3.29}$$

which is of the Robertson form (3.27). But this is not enough. We need to establish analyticity property of $\widetilde{\mathcal{M}}(\tilde{z}, a)$ in $\tilde{z}$ and the desired non-decreasing property of $\mu(\xi)$ over $\xi \in [-1, 1]$.

1. The non-decreasing property of $\mu(\xi)$ over $\xi \in [-1, 1]$ follows so long $\mathcal{A}$ is non-negative. To see this, consider

$$\mu(\xi_1) - \mu(\xi_2) = \int_{\xi_2}^{\xi_1} d\mu(\xi). \tag{3.30}$$

It readily follows from (3.28) that so long $\mathcal{A}$ is non-negative, $\mu(\xi_1) \geq \mu(\xi_2)$ for $\xi_1 \geq \xi_2$. Non-negativity of the absorptive part $\mathcal{A}$ depends on the values of the free parameter $a$. Usually there exists a real interval $a \in I_p$ where $\mathcal{A}$ is non-negative.

2. One further needs to investigate the analyticity of the amplitude inside the unit disc $|\tilde{z}| < 1$. The $\tilde{z}$ analyticity properties are controlled by that of the Robertson kernel

$$\frac{\tilde{z}}{1 - 2\xi\tilde{z} + \tilde{z}^2}. \tag{3.31}$$

It turns out that kernel is analytic inside the unit disc $|\tilde{z}| < 1$ for a particular range of $a$ which gives another interval $I_a$.

Finally, collecting everything, we have that the amplitude $\widetilde{\mathcal{M}}(\tilde{z}, a)$ admits the Roberston representation for

$$a \in I_p \cap I_a, \tag{3.32}$$

and therefore,

$$\widetilde{\mathcal{M}}(\tilde{z}, a) \in TR, \qquad \forall \ a \in I_p \cap I_a. \tag{3.33}$$

Let us illustrate the analysis for the case of EFTs of scalar massless particles, in which case one finds [25] that

$$I_a = \left[-\frac{M^2}{3}, 0\right) \cup \left(0, \frac{2M^2}{3}\right]. \tag{3.34}$$

Next, we need to find the interval $I_p$. We can demand a very strong constraint that each term in the partial wave decomposition (3.9) be positive. The Gegenbauer polynomial functions are positive for $\cos\theta = \sqrt{\frac{s_1 + 3a}{s_1 - a}} \geq 1$. This leads to the constraint $a \in (0, M^2]$ with the upper limit coming from the fact that $a < s_1$ for real $\cos\theta$. Therefore we find that the amplitude is typically real for

$$0 \leq a^{scalar} \leq \frac{2M^2}{3}. \tag{3.35}$$

Note that here we have imposed a kinematic condition- all the Gegenbauer polynomials are positive. We are not considering the fact that depending on the relative ratio of $a_J(s)$, we can have $\cos\theta < 1$, but the overall sum still remain positive. This would need us to consider the dynamical implications of the locality constraints on $a_J(s)$, which we do in the next section and will lower the bound on $a$ from 0.

### 3.5.3 Positivity and Low-Spin Dominance(LSD): Massless scalar EFT

The analysis we have presented so far only requires positivity of the absorptive part as a whole i.e $\mathcal{A}(s_1', s_2^+(s_1', a)) \geq 0$. We imposed the positivity of each term in (3.9), spin by spin, which of course, guarantees the positivity of the total absorptive part. In particular, in this way of imposing positivity we demanded the positivity of Gegenbauer polynomials $C_J^{(\alpha)}(\sqrt{\xi(s_1', a)})$, which led to the constraint $\sqrt{\xi(s_1', a)} > 1$ in the previous subsection, and the positivity of the partial wave amplitudes $a_J$ following from the unitarity. However, this is a rather weak condition on the partial wave amplitudes. The dynamical consequence of locality captured by the locality constraints (3.14) has not been considered. It is quite natural to expect that locality constraints result in relative magnitudes of the partial amplitudes such that the positivity of $\mathcal{A}$ can still be satisfied for $\sqrt{\xi(s_1', a)} < 1$.

Let us illustrate this with the case of massless scalar EFT. We begin with the dispersion relation (3.8) and add

$$N_c := -\sum_{\substack{n < m \\ m \geq 2}} c_{n,m} \mathcal{W}_{n-m,m} a^{2n+m-3} y \tag{3.36}$$

Here $\{c_{n,m}\}$ are arbitrary weights. Using (3.11), we obtain

$$\mathcal{M}(s_i, a) + N_c$$

$$= \alpha_0 + \int_{M^2}^{\infty} \frac{ds_1'}{\pi s_1'} \sum_{\substack{J \geq 0 \\ J \text{ even}}} (2J+1) a_J(s_1') \left[ C_J^{\left(\frac{d-3}{2}\right)}\left(\sqrt{\xi(s_1', a)}\right) - \sum_{\substack{n < m \\ m \geq 2}} c_{n,m} \hat{\mathcal{B}}_{n,m}^J \frac{a^{2n+m} H(a; s_i)}{2(s_1')^{2n+m} H(s_1', s_i)} \right] H(s_1', s_i)$$

$$(3.37)$$

Where $\sqrt{\xi} = 1 + \frac{2s_2^+}{s_1'} = \sqrt{\frac{s_1' + 3a}{s_1' - a}}$ and we have used the fact that $H(a; s_i) = -\frac{y}{a^3}$. We also have the following crucial difference from the massive scalar EFT expression in (3.12)

$$\hat{\mathcal{B}}_{n,m}^J = 2 \sum_{J=0}^{m} \frac{(-1)^{1-J+m} p_J^{(J)}(1)(4)^J (3J - m - 2n)\Gamma(n - J)}{J!(m-J)!\Gamma(n-m+1)} \tag{3.38}$$

i.e $\xi_0 = 1$. The locality constraints (3.14) ensure that $N_c = 0$. Thus adding this to $\mathcal{M}(s_i, a)$ does not change the amplitude. But now we can analyze the consequence of the locality constraints inside the dispersive representation. In fact, we now have the equivalent dispersive representation

$$\mathcal{M}(s_i, a) = \alpha_0 + \frac{1}{\pi} \int_{M^2}^{\infty} \frac{ds_1'}{s_1'} \mathcal{A}_L(s_1', a) H(s_1', s_i), \tag{3.39}$$

with

$$\mathcal{A}_L(s_1', a) := \sum_{\substack{J \geq 0 \\ J \text{ even}}} (2J+1) a_J(s_1') \left[ C_J^{\left(\frac{d-3}{2}\right)}\left(\sqrt{\xi(s_1', a)}\right) - \sum_{\substack{n < m \\ m \geq 2}} c_{n,m} \hat{\mathcal{B}}_{n,m}^J \frac{a^{2n+m} H(a; s_i)}{2(s_1')^{2n+m} H(s_1', s_i)} \right]. \tag{3.40}$$

Let us call $\mathcal{A}_L$ *local absorptive part*. For the purpose of our analysis, we now impose the *local positivity* condition as

$$\mathcal{A}_L(s_1', a) \geq 0, \qquad \forall \ s_1' \geq M^2. \tag{3.41}$$

This condition will result into a new range of validity for $a$. Let us analyze below how this range can be found out. It is worth emphasizing that this condition is *equivalent* to the usual positivity condition $\mathcal{A}(s_1', a) \geq 0$ when restricted to the to the subspace $N_c = 0$ in the space of partial wave amplitudes $\{a_J\}$.

Since we will eventually study the Wilson coefficient expansion as a Laurent series about $x, y = 0$, we analyze $\mathcal{A}_L(s_1', a)$ in a low energy expansion about $x = 0$. In order to do so, we can replace $s_1' \to a\frac{\xi^2 + 3}{\xi^2 - 1}$ in $\mathcal{A}_L$, (3.40), and write it as an expansion about $x = 0$. Using

$$\frac{H(a; s_i)}{H(s_1'; s_i)} = \frac{(\xi^2 + 3)^3}{(\xi^2 - 9)(\xi^2 - 1)^2} + O(x) \tag{3.42}$$

into (3.40), to leading order in $x$, the local positivity requirement becomes

$$\mathcal{A}_L(s_1', a) := \sum_{\substack{J \geq 0 \\ J \text{ even}}} (2J+1) a_J(s_1') \left[ C_J^{\left(\frac{d-3}{2}\right)}\left(\sqrt{\xi(s_1', a)}\right) - \sum_{\substack{n < m \\ m \geq 2}} c_{n,m} \hat{\mathcal{B}}_{n,m}^J \frac{(\xi^2 - 1)^{2n+m-2}}{2(\xi^2 - 9)(\xi^2 + 3)^{2n+m-3}} \right] \geq 0.$$

$$(3.43)$$

Observe that this leading contribution does not depend *explicitly* on $a$ and is purely a function[12] of $\xi$. We can then sum over $n, m$ to $2n + m \leq k$, and find the smallest solution $\xi_{min}$ such that

---

[12] We note that it seems like the denominator has a pole at $\xi = 3$, but this value is never attained since, from the analyticity requirement of $\mathcal{M}$, $a < \frac{2M^2}{3}$, we have $\xi^2 < 3$.

$0 < \xi_{min} < \xi < 1$ for which we can find a set of $c_{n,m}$'s such that (3.41) is satisfied. We can in turn use this value to determine an the new range of $a$ corresponding to the local positivity condition,

$$\sqrt{\frac{s_1' + 3a}{s_1' - a}} > \xi_{min} \implies \frac{(\xi_{min})^2 - 1}{(\xi_{min})^2 + 3}M^2 \le a. \tag{3.44}$$

Since, $0 < \xi_{min} < 1$ the lower bound is stronger than $0 < a$ but also weaker than $-\frac{M^2}{3} < a$. We can now combine this with (3.34) to have

$$\frac{(\xi_{min})^2 - 1}{(\xi_{min})^2 + 3}M^2 \le a \le \frac{2M^2}{3}. \tag{3.45}$$

The above exercise leads us to $\xi_{min} = 0.593$ for the scalar EFT when we consider all locality constraints up to $k = 21$[13] This gives the following:

$$\mathbf{Scalar} : -0.1933M^2 < a^{scalar} < \frac{2M^2}{3}, \tag{3.46}$$

Note that the lower bound of $a$ has been modified from the case where we demanded the positivity of each partial wave without considering the locality constraints. From a physical perspective, the dynamical constraints of UV consistency on scalar IR EFT are responsible for lowering the bound on $a$ from the previous subsection.

Our findings are also indicative of the well-known phenomenon of *low spin dominance*(LSD), i.e. the higher spin partial wave amplitudes are suppressed. To be precise, we can calculate the $\xi_{min}$ in an alternative way.

- Consider (3.37) without the locality constraints, and instead, we truncate the sum over spin to some $J = J_c$ try to find $\xi_{min}$ demanding this finite sum be positive. This means we assume that the sign of the absorptive part in (3.37) does not change beyond a certain critical spin $J = J_c$ because of the smallness of $a_{J>J_c}(s_1)$. Therefore truncating the partial wave sum and doing the positivity analysis is justified. Formally, the positivity condition for the truncated expression reads

$$\sum_{\substack{J=0 \\ J\,\text{even}}}^{J_c} (2J + 1)a_J(s_1') \, C_J^{(\frac{d-3}{2})}\left(\sqrt{\xi(s_1', a)}\right) > 0, \qquad \forall \ \xi > \xi_{min}, \ s_1' > M^2. \tag{3.47}$$

We are also assuming that for $a_J \ne 0$ at least for one $J \in \{0, 2, 4, \cdots, J_c\}$.

- Therefore, we look for the smallest simultaneous root $\xi_{J_c}$ of the set

$$\left\{ C_J^{(\frac{d-3}{2})}\left(\sqrt{\xi(s_1', a)}\right) \,\Big|\, J \in \{0, 2, 4, \cdots, J_c\} \right\}$$

such that

$$C_J^{(\frac{d-3}{2})}\left(\sqrt{\xi(s_1', a)}\right) > 0, \quad \forall \ \xi > \xi_{J_c}, \, J \in \{0, 2, 4, \cdots, J_c\}. \tag{3.48}$$

For a given $J_c$, this $\xi_{J_c}$ is the $\xi_{min}$ that we considered before. In particular, we have that $\xi_{J_c} \to 1$ as $J_c \to \infty$, which is expected. Therefore, for the truncated set we consider, $1 \ge \xi > \xi_{J_c}$ ensures that the LHS is positive since we have assumed that the sign of the absorptive part doesn't change after $J_c$.

---

[13]In order to obtain this numerical coefficient, we have used linear programming in Mathematica with 1700 digits of precision to find solutions to the system of inequalities (3.43) for $k \le 21$ while varying $\xi$ in steps of 0.048 from $\xi_{min} = 0.59329$ to $\xi_{max} = 2.99$ and spin $J$ from $J = 0$ to $J_{max} = 56$. The value of $\xi_{min}$ is determined by the lowest value of $\xi$ for which the system of inequalities (3.43) have a solution such that not all $c_{n,m} = 0$ & $c_{n,m} > -\infty$.

- Combining this with $PB_C$, this constrains the range of $a$ to

$$\frac{(\xi^{(J_c)})^2 - 1}{(\xi^{(J_c)})^2 + 3} M^2 \leq a \leq \frac{2M^2}{3}.$$  (3.49)

The first few values after rationalising to agree with 2 significant digits are:

| $J_c$ | Scalar |
|---|---|
| 2 | $-0.2M^2 < a < \frac{2M^2}{3}$ |
| 3 | $-0.69M^2 < a < \frac{2M^2}{3}$ |
| 4 | $-0.034M^2 < a < \frac{2M^2}{3}$ |

- We can see that the argument with locality constraints combined with the above analysis clearly indicates Spin-2 dominance for the scalar case. More precisely, we input locality constraints in estimating the range of $a$ in first part of this subsection (i.e in the analysis leading up to (3.46)). Locality constraints can also be interpreted as constraints on allowed $a_J(s)$ for scalar EFTs. Thus the range of $a$, after including the locality constraints (i.e., considering the allowed space of scalar theories), approximately coincides with the range that we get from a completely different analysis without using the null constraints *and assuming* that higher scalar partial waves do not change the sign of the absorptive part after spin 2 (see 1st entry of the table above). This implies that UV consistency of scalar EFTs leads to spin 2 dominance.

### 3.5.4 Massive scalars

For a massive theory we can repeat the analysis of the previous subsection. We shall consider the case of the massive scalar with mass $m$ and $\mu = 4m^2$. This was already considered in [25] where it was argued that the range of $a$ was $\frac{-M^2}{3} < a < \frac{2M^2}{3}$ and bounds were obtained for various Wilson coefficients. We revisit this using our new method using the locality constraints. The key changes are in the relation between $\xi$ and $s_1', a$ which is given by $\xi = \xi_0 \sqrt{\frac{s_1' + 3a}{s_1' - a}}$ with $\xi_0 = \frac{s_1'}{s_1' - \mu} > 1$ and the locality constraints (3.12):

$$\mathcal{B}_{n,m}^{J,i}(s_1) = 2 \sum_{j=0}^{m} \frac{(-1)^{1-j+m} p_J^{(j,i)}(\xi_0) (4\xi_0)^j (3j - m - 2n)\Gamma(n - j)}{j!(m - j)!\Gamma(n - m + 1)}.$$  (3.50)

It can be easily checked that this gives

$$\frac{a^{2n+m} H(a; s_i)}{(s_1')^{2n+m} H(s_1'; s_i)} = \frac{(\xi^2 - \xi_0^2)}{(\xi^2 - 9\xi_0^2)(\xi^2 + 3\xi_0^2)} + o(x).$$

Proceeding with the analysis it turns out that there are no solutions for any $\xi < 1$. However since $\xi = 1$ was used to obtain the previous range of $a$ namely $\frac{-M^2}{3} < a < \frac{2M^2}{3}$ these do not give us a stronger range of $a$. Thus, we conclude that

*there is no low spin dominance for the massive case.*

This justifies the results in [25] and highlights a key difference between the massive and massless cases. We have carried out explicit checks using the pion S-matrices from the S-matrix bootstrap [48–50] which verifies this claim.

## 3.6 Bieberbach-Rogosinski bounds

We can expand $\widetilde{M}(\tilde{z}, a)$ about $\tilde{z} = 0$ by expanding the kernel $H(s_1, \tilde{z})$

$$H(s_1, \tilde{z}) = \frac{27 a^2 \tilde{z} (2 s_1 - 3a)}{27 a^3 \tilde{z} - 27 a^2 \tilde{z} s_1 - (\tilde{z} - 1)^2 s_1^3} = \sum_{n=0}^{\infty} \beta_n(a, s_1) \tilde{z}^n \,, \qquad (3.51)$$

Comparing this with the low energy expansion of the amplitude

$$\widetilde{M}_0(\tilde{z}, a) = \sum_{p,q=0}^{\infty} \mathcal{W}_{p,q} x^p y^q = \sum_{n=0}^{\infty} a^{2n} \alpha_n(a) \tilde{z}^n$$

after rewriting in-terms of $\tilde{z}$ using $x = -\frac{-27 a^3 \tilde{z}}{(1-\tilde{z})^2}$ and $y = -\frac{-27 a^2 \tilde{z}}{(1-\tilde{z})^2}$ gives:

$$a^{2n} \alpha_n(a) = \frac{1}{\pi} \int_{M^2}^{\infty} \frac{ds_1'}{s_1'} \mathcal{A}(s_1'; s_2^+(s_1', a)) \beta_n(a, s_1') \,,$$

$$\text{with } \alpha_p(a) = \sum_{n=0}^{p} \sum_{m=0}^{n} \mathcal{W}_{n-m,m} a^{2n+m-2p} (-27)^n \frac{\Gamma(n+p)}{\Gamma(2n)(p-n)!} \,, \quad p \geq 1 \,. \qquad (3.52)$$

In particular we have $\mathcal{W}_{0,0} = \alpha_0$ and $a^2 \alpha_1(a) = \frac{1}{\pi} \int_{M^2}^{\infty} \frac{ds_1'}{s_1'} \mathcal{A}(s_1'; s_2^+(s_1', a)) \beta_1(a, s_1')$. Note that since $\beta_1(a, s_1) = \frac{27 a^2}{s_1^3}(3a - 2s_1)$ and $a < \frac{2M^2}{3} < \frac{2s_1}{3}$ we have $\beta_1 < 0$. Thus,

$$\boxed{\alpha_1(a) < 0} \qquad (3.53)$$

We can apply the Bieberbach-Rogosinski inequalities on the coefficients of any typically-real function $f(z) = z + a_2 z^2 + a_3 z^3 \cdots$ inside the unit disk following [25]:

$$-\kappa_n \leq \frac{\alpha_n(a) a^{2n}}{\alpha_1(a) a^2} \leq n \qquad (3.54)$$

with

$$\kappa_n = n \text{ for even } n, \qquad \kappa_n = \frac{\sin n \, \vartheta_n}{\sin \vartheta_n} \text{ for odd } n \,, \qquad (3.55)$$

where $\vartheta_n$ is the smallest solution of $\tan n\vartheta = n \tan \vartheta$ located in $(\frac{\pi}{n}, \frac{3\pi}{2n})$ for $n > 3$ and $\kappa_3 = 1$, to constrain the Wilson coefficients in a low-energy expansion of the amplitude. We call these conditions (3.54) collectively as $TR_U$.

## 3.7 Summary of algorithm

In this section, we summarise our algorithm. The central characters of the story are the Wilson Coefficients, the partial wave decomposition of the amplitude and the crossing symmetric kernel. Firstly, unitarity of the partial wave amplitude decomposition and positivity of the spherical harmonics and their derivatives for unphysical region of scattering, translate to positivity relations of the Wilson coefficients ( also known as the $PB_C$ conditions [25]). To be more precise, unitarity demands that the imaginary part of the partial wave coefficients is positive. The Gegenbauer polynomials (or the relevant linear combination of the Wigner-$d$ functions for the spinning case) or its derivatives which appear in the partial wave expansion of the amplitude are positive in the unphysical region of scattering $(\cos\theta > 1)$. The Wilson coefficients themselves however might contain positive or negative sum of both the manifestly positive quantities. The $PB_C$ conditions are then linear combination of the Wilson coefficient expressions such that it is manifestly positive. Secondly, the fact that the amplitude is typically real for a range of the parameter $a$ then allows us to systematically obtain two-sided bounds

on the Wilson coefficients (also known as the $TR_U$ conditions [25]). This is because the typically real amplitude, as an expansion in $\tilde{z}$, has Bieberbach-Rogosinski bounds on the expansion coefficients [25]. Thirdly, we use locality, which modifies the lower range of $a$ as obtained from $\cos\theta > 1$ and $TR_U$. In the following sections, we systematically implement this algorithm to first review bounds on the scalar and then obtain the same for graviton and photon EFTs. These steps are summarised in the flow chart below:

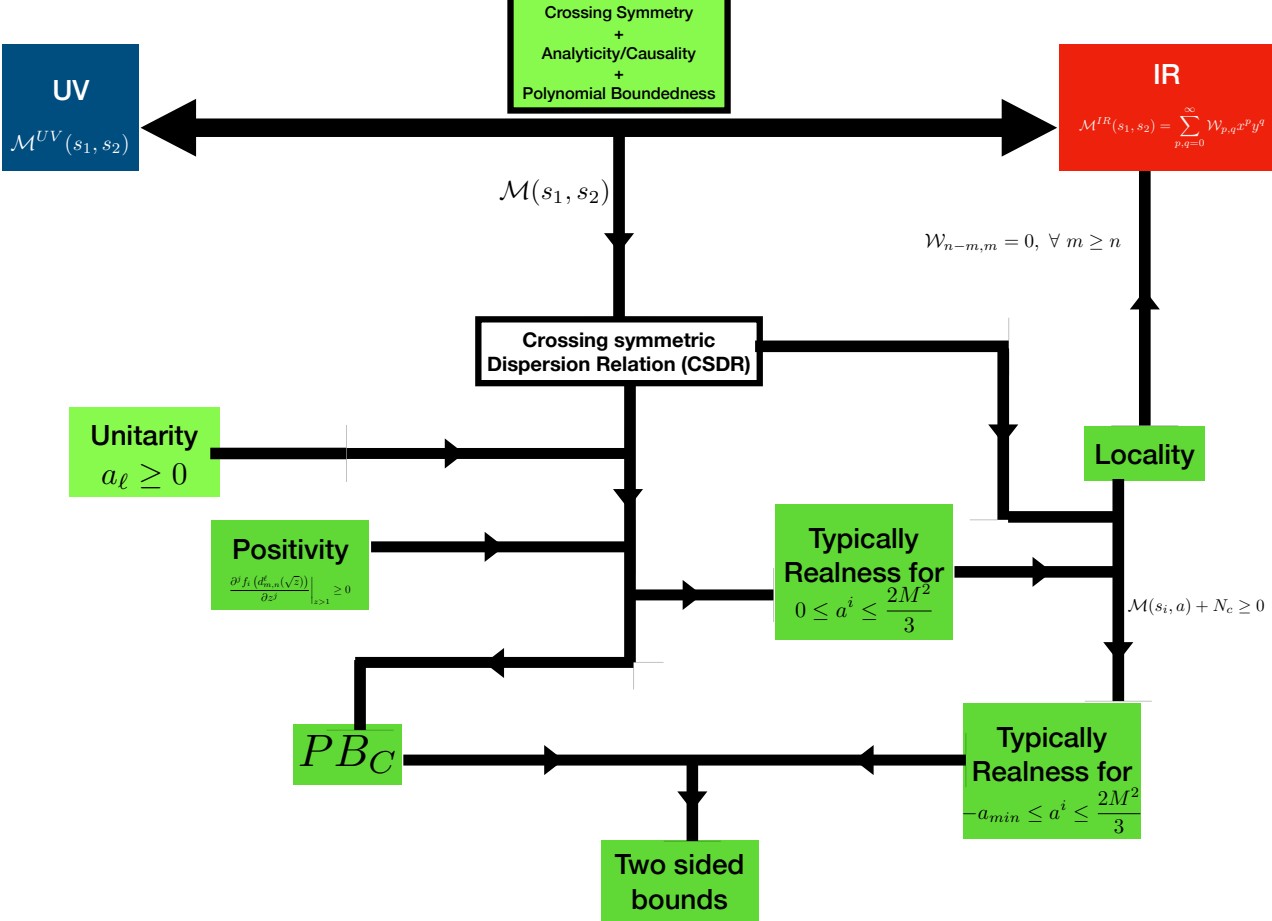

**Figure 1:** The above flowchart shows the steps involved in the GFT approach.

# 4    Scalar bounds

We now present the applications of formalism developed in the previous sections for various EFTs starting with the massless scalar case. The massive scalar was already addressed in [25]. Recall that the low energy EFT expansion of the amplitude takes the following form in terms of crossing-symmetric variables $x$, $y$

$$ F(s_1, s_2, s_3) \;=\; \sum_{p,q=0} \mathcal{W}_{p,q} x^p y^q \,, $$

Starting with the dispersion relation given by (3.8) and (3.9) we can systematically derive the positivity bounds. We linearise the steps in the following manner, which will serve as a guideline for us when evaluating EFTs with spinning particles.

- Unitarity implies that in the dispersion relation, the spectral functions $a_J$ are positive.

- From the positivity of the Gegenbauer polynomials in the dispersion relation and the typical realness of the amplitude, we have the following range for $a$ (3.46),

$$-0.1933M^2 < a^{scalar} < \frac{2M^2}{3}.$$

From (3.22), (3.23) and (3.24), we get positivity conditions on linear combinations [14] of $w_{p,q} = \frac{\mathcal{W}_{p,q}}{\mathcal{W}_{1,0}}$. These set of conditions have been referred to in literature [25] as $PB_C$ conditions. Since the amplitude is typically real, we also obtain Bieberbach Rogosinski bounds on $w_{p,q}$ from (3.54) (also known in literature as $TR_U$). The algorithm will generically follow [25] and we refer the interested reader to the details there. We begin by first noting that (3.53) implies

$$
\begin{aligned}
\alpha_1(a) \quad &= \quad -27a^2(1 + a\ w_{01}) \le 0 \qquad \forall -0.1933M^2 < a < \frac{2M^2}{3}, \\
&\Longrightarrow \quad \frac{-3}{2M^2} \le w_{01} \le \frac{5.1733}{M^2}.
\end{aligned}
\tag{4.1}
$$

We note that this precisely agrees with the result in [5] when we account for the difference in the definitions of $x$ due to the conventions. We have used $x = -s_1 s_2 - s_2 s_3 - s_3 s_1$ while [5] used $s_1^2 + s_2^2 + s_3^2$ which gives $w_{01} = \frac{-\tilde{g}_3}{2}$ (since $s_1 + s_2 + s_3 = 0$) which translates to $-10.34 < \tilde{g}_3 < 3$. In the second step, we solve for the set of inequalities derived from (3.22), (3.23), (3.24) and (3.54) upto a certain value of $n = n_{max}$. Note that the conditions derived from (3.54) are $a$ dependent. In order to efficiently solve the inequalities, we discretize the variable $a$ over the range specified in (3.46) in steps of $\delta a$ and then solve for the resulting larger set of inequalities. We present our results for $n_{max} = 5$ and $\delta a = \frac{1}{101}$ in the table below. We shall follow the convention of [25] namely $M^2 = \frac{8}{3}$ and re-write the results of [5] in this convention for ease of comparison.

| $w_{p,q} = \frac{\mathcal{W}_{p,q}}{\mathcal{W}_{1,0}}$ | $(TR_U + PB_C)^{min}$ | $SDPB^{min}$ | $(TR_U + PB_C)^{max}$ | $SDPB^{max}$ |
|---|---|---|---|---|
| $w_{01}$ | $-0.5625$ | $-0.5625$ | $1.939$ | $1.939$ |
| $w_{11}$ | $-0.1318$ | $-0.1318$ | $0.219$ | $0.216$ |
| $w_{02}$ | $-0.1533$ | $-0.1268$ | $0.063$ | $0.0296$ |
| $w_{20}$ | $0$ | $0$ | $0.140625$ | $0.140625$ |
| $w_{21}$ | $-0.02595$ | $-0.02595$ | $0.0513$ | $0.023$ |
| $w_{12}$ | $-0.061$ | $-0.02789$ | $0.0275$ | $0.0111$ |
| $w_{30}$ | $0$ | $0$ | $0.01977$ | $0.01977$ |
| $w_{03}$ | $-0.011$ | $-0.00156$ | $0.017$ | $0.0071$ |
| $w_{31}$ | $-0.0047$ | $-0.0047$ | $0.0022$ | $0.0022$ |
| $w_{40}$ | $0$ | $0$ | $0.00278$ | $0.00278$ |
| $w_{50}$ | $0$ | $0$ | $0.00039$ | $0.00039$ |

**Table 2:** A comparison of the values obtained using our results up to $n = 5$ and $SDPB$ in [5]. We used the locality constraints and the techniques of [5] adapted using linear programming to generate the SDPB values quoted above. These are identical to the ones quoted in [5]. The exact agreement between these values is a consequence of the fact that the locality and null constraints are equivalent as observed in [30].

We note that we get an excellent agreement with [5]. In [25] a comparison was done with the massive case and the results of [5], where it was noted that some of the results $TR_U$ were stronger. However, since [5] considered only the massless case, so the above is more appropriate comparison as the results show. We attribute the discrepancy in the $w_{02}, w_{21}, w_{12}, w_{03}$ values to the following: We have not completely solved the Locality constraints $N_c$ but we have implicitly assumed that they

---

[14]This is well defined as $\mathcal{W}_{1,0} > 0$ as argued in (5.13).

are zero when we consider a low energy expansion (4.1). However, each Wilson coefficient actually involves an infinite sum of locality constraints for instance

$$\alpha_1(a) = -27a^2 \mathcal{W}_{1,0} \left( (1 + aw_{01}) + \sum_{n=1}^{\infty} w_{-n,n+1}\, a^{n-1} \right). \tag{4.2}$$

So strictly speaking, what we have are bounds on these combinations and not the $w_{p,q}$'s themselves. In practice, however one would have expected that since we obtained the range of $a$ by using some of the locality constraints, this should have resolved the issue. However, we remind the reader that to get the bounds listed in the table we used $PB_c$ conditions in addition to the $TR_U$'s. The $PB_c$'s are linear conditions in the $w_{p,q}$'s which are *independent* of $a$.

# 5   Photon bounds

In this section, we will constrain parity preserving photon EFTs. To be precise, let us consider the following crossing symmetric helicity amplitudes from (2.20) and (2.21). This set up applies almost identically to the graviton case also so we present here the general amplitude that we will be considering for later use.

$$
\begin{aligned}
M^\alpha(s_1, s_2, s_3) &= F_2^\alpha(s_1, s_2, s_3) + x F_1^\alpha(s_1, s_2, s_3) \\
&= (T_1^\alpha(s_1, s_2, s_3) + T_3^\alpha(s_1, s_2, s_3) + T_4^\alpha(s_1, s_2, s_3)) + x_1 T_2^\alpha(s_1, s_2, s_3)
\end{aligned}
\tag{5.1}
$$

where $x_1 \in [-1, 1]$ and $\alpha = \gamma, h$ for photons and gravitons respectively. The partial wave expansion of this amplitude is given by,

$$
(F_2^\alpha(s_1, s_2) + x F_1^\alpha(s_1, s_2)) = \sum_{J=0,2,4,\cdots} 16\pi(2J+1)(\rho_J^{1;\alpha} + x\rho_J^{2;\alpha}) d_{0,0}^J(\theta) +
$$

$$
\sum_{J=2,4,\cdots} 16\pi(2J+1)\rho_J^{3;\alpha}(d_{2,2}^J(\theta) + d_{2,-2}^J(\theta)) + \sum_{J=3,5,\cdots} 16\pi(2J+1)\rho_J^{3;\alpha}(d_{2,2}^J(\theta) - d_{2,-2}^J(\theta))
$$

$$\tag{5.2}$$

$d_{m,m'}^J$ is the Wigner $d$-matrix defined in appendix (D). From the positivity of the spectral functions in these cases( see appendix (C.1)), the reader can understand that this combination is positive[15]- since $\rho_J^{1,\alpha} \pm \rho_J^{2,\alpha} \geq 0$ we have

$$
\rho_J^{1,\alpha} + x_1 \rho_J^{2,\alpha} = \underbrace{\frac{(1+x_1)}{2}(\rho_J^{1,\alpha} + \rho_J^{2,\alpha})}_{\geq 0} + \underbrace{\frac{(1-x_1)}{2}(\rho_J^{1,\alpha} - \rho_J^{2,\alpha})}_{\geq 0} \geq 0
\tag{5.3}
$$

while $\rho_J^{3,\alpha} \geq 0$ from the analysis in appendix (C.1). The crossing symmetric dispersion relation for the photon amplitude is given by,

$$
(F_2^\gamma(s_1, s_2) + x F_1^\gamma(s_1, s_2)) = \alpha_0^\gamma + \frac{1}{\pi} \int_{M^2}^{\infty} \frac{ds_1}{s_1'} \mathcal{A}^\gamma \left( s_1'; s_2^{(+)}\left(s_1', a\right) \right) H\left(s_1'; s_1, s_2, s_3\right),
$$

$$\tag{5.4}$$

---

[15] We leave the analysis of $F_4, F_5$ which have denominators involving $s_i$ for later analysis. For $F_3$ since unitarity does not fix the sign of $\rho_J^{\alpha,5}$ our methods are not applicable.

where $H(s'_1; s_1, s_2, s_3)$ is defined in (3.4) and the partial wave decomposition reads

$$
\begin{aligned}
\mathcal{A}^\gamma \left( s'_1; s_2^{(+)}(s'_1, a) \right) &= \sum_{J=0,2,4,\cdots} 16\pi(2J+1)(\rho_J^{1,\gamma} + x_1 \rho_J^{2,\gamma}) d_{0,0}^J(\theta) + \\
&\quad \sum_{J=2,4,\cdots} 16\pi(2J+1)\rho_J^{3,\gamma}(d_{2,2}^J(\theta) + d_{2,-2}^J(\theta)) + \\
&\quad + \sum_{J=3,5,\cdots} 16\pi(2J+1)\rho_J^{3,\gamma}(d_{2,2}^J(\theta) - d_{2,-2}^J(\theta)) ,
\end{aligned} \tag{5.5}
$$

where $\cos^2 \theta = \xi(s'_1, a) = 1 + 4\left(\frac{a}{s'_1 - a}\right)$. Note that due to the fact that we have written down crossing symmetric combination of helicity amplitudes, the crossing symmetric dispersion relation is essentially of the same structure as the scalar one. In writing the dispersion relations (3.8) and (5.2), we have used (3.16) and (3.17). The low energy EFT expansion of the amplitude reads,

$$
F_1^\gamma(s_1, s_2) = \sum_{p,q} \mathcal{W}_{p,q}^1 x^p y^q, \qquad F_2^\gamma(s_1, s_2) = \sum_{p,q} \mathcal{W}_{p,q}^2 x^p y^q . \tag{5.6}
$$

For our analysis, we will be considering the most general Euler-Heisenberg type EFT for the photon

$$
\mathcal{L} = -\frac{1}{4} F_{\mu\nu} F^{\mu\nu} + a_1 (F_{\mu\nu} F^{\mu\nu})^2 + a_2 (F_{\mu\nu} \tilde{F}^{\mu\nu})^2 + \cdots \tag{5.7}
$$

obtained starting with a UV complete theory such as QED and integrating out the other massive particles in the theory such as say the electron. To compare against the corresponding low energy EFT expansion coefficients of [22] we can rewrite our EFT expansion in the form (see (F.1)),

$$
F_2(s_1, s_2, s_3) = 2g_2 x - 3g_3 y + 2(g_{4,1} + 2g_{4,2})x^2 + \cdots \tag{5.8}
$$
$$
F_1(s_1, s_2, s_3) = 2f_2 x - f_3 y + 4f_4 x^2 + \cdots \tag{5.9}
$$

where the Wilson coefficients can be related to the EFT couplings such as $a_1 = \frac{f_2 + g_2}{16}$, $a_2 = \frac{f_2 - g_2}{16}$ etc.

## 5.1  Wilson coefficients and Locality constraints: $PB_C^\gamma$

The local low energy expansion of the amplitude (5.1) can be written as

$$
F_2^\gamma(s_1, s_2) + x_1 F_1^\gamma(s_1, s_2) = \sum_{p,q=0}^\infty \mathcal{W}_{p,q}^{(x)} x^p y^q = \sum_{p,q=0}^\infty \mathcal{W}_{p,q}^{(x)} x^{p+q} a^q \tag{5.10}
$$

where we have used $a = y/x$ and $\mathcal{W}_{p,q}^{(x)} = \mathcal{W}_{p,q}^2 + x_1 \mathcal{W}_{p,q}^1$. Just like in the scalar case, we would like to expand both the sides of the dispersion relation (5.4) to derive an expression for the locality constraints- recall that by incorporating crossing symmetry we have compromised on locality which serves as constraints in our formalism. In order to do so, we expand the kernel (3.4) and the partial wave Wigner-$d$ functions in (5.4) about $a = 0$ and compare powers on both sides. Note that for $a = 0$, the Wigner-$d$ functions are Taylor expanded about $\xi_0 = 1$ (since the argument of the Wigner-$d$ functions are $\xi(s_1, a) = 1 + 4\left(\frac{a}{s_1 - a}\right)$). We obtain,

$$
\begin{aligned}
\mathcal{W}_{n-m,m}^{(x_1)} &= \int_{M^2}^\infty \frac{ds_1}{2\pi s_1^{2n+m+1}} \sum_{J=0,2,4,\cdots} (2J+1) a_J^{(1)}(s_1) \mathcal{G}_{n,m}^{J,1} , \\
&+ \int_{M^2}^\infty \frac{ds_1}{2\pi s_1^{2n+m+1}} \sum_{J=2,4,\cdots} (2J+1) a_J^{(2)}(s_1) \hat{\mathcal{G}}_{n,m}^{J,2} , \\
&+ \int_{M^2}^\infty \frac{ds_1}{2\pi s_1^{2n+m+1}} \sum_{J=3,5,7,\cdots} (2J+1) a_J^{(3)}(s_1) \hat{\mathcal{G}}_{n,m}^{J,3} , \\
\hat{\mathcal{G}}_{n,m}^{J,i} &= 2 \sum_{j=0}^m \frac{(-1)^{1-j+m} q_J^{(j,i)}(1)(4)^j (3j - m - 2n) \Gamma(n-j)}{j!(m-j)! \Gamma(n-m+1)} .
\end{aligned} \tag{5.11}
$$

where $a_J^{(1)} = \rho_J^{1,\gamma} + x\rho_J^{2,\gamma}$, $a_J^{(2)} = a_J^{(3)} = \rho_J^{3,\gamma}$ and $q_J^{(j,i)}(1) = \left.\frac{\partial^j f^{(i)}(\sqrt{\xi})}{\partial \xi^j}\right|_{\xi=\xi_0=1}$ with $f^{(1)} = d_{0,0}^J$, $f^{(2)} = d_{2,2}^J + d_{2,-2}^J$, $f^{(3)} = d_{2,2}^J - d_{2,-2}^J$. For convenience, in order to compute the partial derivatives $q_J^{(j,i)}(1)$, we use the representation of the Wigner-$d$ functions in terms of Hypergeometric functions, given in (D.1). The locality constraints for this case are therefore given by

$$\mathcal{W}_{n-m,m}^{(x_1)} = 0 \quad \forall n < m\,. \tag{5.12}$$

We would also like to construct the spinning equivalent of $PB_C$ as done for scalars. To this end, we note that spectral functions $a_J^{(i)} \geq 0$ by unitarity and $\{d_{0,0}^J(\theta), d_{m,m}^J(\theta) \pm d_{m,-m}^J(\theta)\}$ are positive for all $J$ whenever cosine of the argument is bigger than or equal to 1 (see D) i.e., $q_J^{(j,i)}(1) > 0$ for all $J, j = 0, 1, 2, \cdots$ and $i = 1, 2, 3$. In particular we have

$$
\begin{aligned}
\mathcal{W}_{n,0}^{(x_1)} &= \int_{M^2}^{\infty} \frac{ds_1}{2\pi s_1^{2n+m+1}} \sum_{J=0,2,4,\cdots} (2J+1) a_J^{(1)}(s_1) q_J^{(0,1)}(1)\,, \\
&+ \int_{M^2}^{\infty} \frac{ds_1}{2\pi s_1^{2n+m+1}} \sum_{J=2,4,\cdots} (2J+1) a_J^{(2)}(s_1) q_J^{(0,2)}(1)\,, \\
&+ \int_{M^2}^{\infty} \frac{ds_1}{2\pi s_1^{2n+m+1}} \sum_{J=3,5,7,\cdots} (2J+1) a_J^{(3)}(s_1) q_J^{(0,3)}(1) \geq 0\,.
\end{aligned} \tag{5.13}
$$

More generally in (5.11) the sign of any term in $J$ expansion is controlled by $\mathcal{G}_{n,m}^{J,i}(s_1)$ alone. We can thus take linear combinations of various $\mathcal{W}_{p,q}^{(x)}$'s which is a positive sum of $\{d_{0,0}^J(\theta), d_{m,m}^J(\theta) \pm d_{m,-m}^J(\theta)\}$ and their derivatives and hence is manifestly positive. This gives us the *Positivity conditions*:

$$\sum_{r=0}^{m} \chi_n^{(r,m)}(M^2) \mathcal{W}_{n-r,r}^{(x)} \geq 0, \qquad 0 \leq \mathcal{W}_{n,0}^{(x)} \leq \frac{1}{(M^2)^2} \mathcal{W}_{n-1,0}^x\,, \qquad n \geq 2\,. \tag{5.14}$$

The $\chi_n^{(r,m)}(M^2)$ satisfy the recursion relation:

$$
\begin{aligned}
\chi_n^{(m,m)}(M^2) &= 1 \\
\chi_n^{(r,m)}(M^2) &= \sum_{j=r+1}^{m} (-1)^{j+r+1} \chi_n^{(j,m)} \frac{\mathscr{U}_{n,j,r}(M^2)}{\mathscr{U}_{n,r,r}(M^2)}
\end{aligned} \tag{5.15}
$$

with $\mathscr{U}_{n,m,k} = -\frac{4^k \Gamma\left(\frac{1}{2}(2k+1)\right)(3k-m-2n)\Gamma(n-k)s1^{-m-2n} {}_4F_3\left(\frac{k}{2}+\frac{1}{2},\frac{k}{2},k-m,k-\frac{m}{3}-\frac{2n}{3}+1;k+1,k-n+1,k-\frac{m}{3}-\frac{2n}{3};4\right)}{\sqrt{\pi}\Gamma(k+1)\Gamma(-k+m+1)\Gamma(-m+n+1)}$. We call the conditions (5.14) collectively as $PB_C^{\gamma}$[16]. We note here that the positivity conditions $PB_C^{\gamma}$ in this case are identical to the ones for massive scalar in [25, 29]. This is simply a consequence of the fact that (5.11) is the sum of three scalar like terms each of which has an identical structure except for the functions $q_J^{(j,i)}(1)$ in $\mathcal{G}_{n,m}^{J,i}(s_1)$. Since we do not use the explicit form of the function $q_J^{(j,i)}(1)$ anywhere in the argument above but just the fact that its positive, the result simply follows. Note that these positive combinations are certainly not unique and one can definitely find different linear combinations which may result in a stronger constraint however we will not pursue that here.

## 5.2 Typical Realness and Low Spin Dominance: $TR_U^{\gamma}$

In this section, we try to get $a$ range of $a$ using positivity of the amplitude coupled with locality constraints and typical realness of the amplitude. The analysis for typical realness is straightforward.

---

[16] We have used the closed form expression for $\mathscr{U}_{n,m,k}$ in [29]

From, section 3.5 and the discussion regarding the Robertson form of the integral (see the discussion around (3.34)), the limit of $a$ is given by,

$$\left(a \in \left[-\frac{M^2}{3}, 0\right) \cup \left(0, \frac{2M^2}{3}\right]\right) \ \cap \ \left(a \ \forall \ A(s_1,; s_2^{(+)}(s_1, a) \geq 0\right).$$

(5.16)

We can assume the positivity of the absorptive part as a whole i.e $\mathcal{A}(s_1', s_2^{+}(s_1', a)) \geq 0$ in (5.4) which gives us the range of $a$ as $a \in (0, \frac{2M^2}{3}]$ . This is obtained by considering the positivity of each term in the spin sum which of course guarantees the positivity of the full absorptive part though it maybe too strong (similar to the massless scalar case). A more careful analysis requires us to use the locality constraints $N_c^\gamma = - \sum_{\substack{n<m \\ m\geq 2}} c_{n,m} W_{n-m,m}^{x_1} a^{2n+m-3} y$ to it with arbitrary weights $c_{n,m}$'s (5.12).

$\mathcal{M}(s_i, a) + N_c =$

$$\int_{M^2}^\infty \frac{ds_1}{2\pi s_1} \sum_{J=0,2,4,\cdots} (2J+1) a_J^{(1)}(s_1) \left[f_J^{(1)}(\xi) - \sum_{\substack{n<m \\ m\geq 2}} c_{n,m} \hat{\mathcal{G}}_{n,m}^{J,1} \frac{a^{2n+m} H(a; s_i)}{(s_1')^{2n+m} H(s_1', s_i)}\right] H(s_1', s_i)$$

$$+ \int_{M^2}^\infty \frac{ds_1}{2\pi s_1} \sum_{J=2,4,\cdots} (2J+1) a_J^{(2)}(s_1) \left[f_J^{(2)}(\xi) - \sum_{\substack{n<m \\ m\geq 2}} c_{n,m} \hat{\mathcal{G}}_{n,m}^{J,2} \frac{a^{2n+m} H(a; s_i)}{(s_1')^{2n+m} H(s_1', s_i)}\right] H(s_1', s_i)$$

$$+ \int_{M^2}^\infty \frac{ds_1}{2\pi s_1} \sum_{J=3,5,7,\cdots} (2J+1) a_J^{(3)}(s_1) \left[f_J^{(3)}(\xi) - \sum_{\substack{n<m \\ m\geq 2}} c_{n,m} \hat{\mathcal{G}}_{n,m}^{J,3} \frac{a^{2n+m} H(a; s_i)}{(s_1')^{2n+m} H(s_1', s_i)}\right] H(s_1', s_i) \geq 0,$$

(5.17)

where $\hat{\mathcal{G}}_{n,m}^{J,i}$ has been defined in (5.11). Note that this is similar to the equation we had for the scalar case (3.37) and therefore the analysis is also similar. The algorithm is very similar with the only difference is that the $\xi_{min}$ is determined by the maximum lower bound obtained by considering the positivity of three different classes of inequalities- the coefficients of $a_J^{(i)}$ for $i = 1, 2, 3$. This exercise, outlined in detail in subsubsection 3.5.3, leads us to $\xi_{min}^\gamma = 0.723$ for the photon EFT when we consider all locality constraints up to $2n + m \leq 12$ and $J_{max} \leq 20$. Using the relation $\frac{\xi_{min}^2 - 1}{\xi_{min}^2 + 3} M^2 < a < M^2$ and (5.16), we obtain,

$$\textbf{Photon}: -0.1355 M^2 < a^\gamma < \frac{2M^2}{3} .$$

(5.18)

Similar to the scalar case, for the photon also we discover the phenomenon of *Low Spin Dominance* (LSD). Consider the set of equations (5.17) without the locality constraints but with a maximal spin cut-off $J = J_c$. If we assume that the absorptive part is unaffected by the contributions from partial waves after $J > J_c$, the positivity of this finite sum of partial waves leads us to an independent derivation of $\xi_{min}^\gamma$. It suffices to choose the largest root $\xi^\gamma(J) \leq 1$ of the set of polynomials $\{d_{0,0}^J, d_{2,2}^J \pm d_{2,-2}^J\}$ for a fixed $J \leq J_c$ to ensure the positivity of the corresponding term in (5.2). We observe the following table.

| $J_c$ | Photon |
|---|---|
| 2 | $-0.2M^2 < a^\gamma < \frac{2M^2}{3}$ |
| 3 | $-0.143M^2 < a^\gamma < \frac{2M^2}{3}$ |
| 4 | $-0.069M^2 < a^\gamma < \frac{2M^2}{3}$ |

We can see that the argument with Locality constraints combined with the above clearly indicates spin-3 dominance for the photon case. Therefore for this range of $a$, we can impose the Bieberbach Rogosinski bounds of subsection 3.6 on the Wilson coefficients $\mathcal{W}_{n-m,m}^{(x_1)}$ -these constraints are called $TR_U^\gamma$. In the next section we present the bounds obtained from $PB_C^\gamma$, $TR_U^\gamma$ and the corresponding range of $a$ (5.18).

## 5.3   Bounds

We now apply our formalism to bound Wilson coefficients in the Euler-Heisenberg type EFT for the photon. Recall that the low energy EFT expansion has the form,

$$\mathcal{L} = \frac{-1}{4} F_{\mu\nu} F^{\mu\nu} + a_1 \left( F_{\mu\nu} F^{\mu\nu} \right)^2 + a_2 (F_{\mu\nu} \tilde{F}^{\mu\nu})^2 + \cdots \tag{5.19}$$

For such an EFT, we have the following crossing symmetric S-matrices (see appendix F) ,

$$
\begin{aligned}
F_1(s_1, s_2, s_3) &= 2f_2 x - f_3 y + 4f_4 x^2 - 2f_5 xy + f_{6,1} y^2 + 8f_{6,2} x^3 + \cdots \\
F_2(s_1, s_2, s_3) &= 2g_2 x - 3g_3 y + 2(g_{41} + 2g_{42}) x^2 + (-5g_{5,1} - 3g_{5,2}) xy + 3 \left( g_{6,1} - g_{6,2} + g_{6,3} \right) y^2 + 2g_{6,1} x^3 + \cdots ,
\end{aligned}
\tag{5.20}
$$

where the Wilson coefficients can be related to the EFT couplings such as $a_1 = \frac{f_2+g_2}{16}$, $a_2 = \frac{f_2-g_2}{16}$ etc. We begin by listing out the $PB_c^\gamma$ and $TR_U^\gamma$ conditions for $n = 3$ (see (3.54), (5.14) and (5.18)). The $PB_c^\gamma$ conditions are,

$$\frac{9w_{20}^{(x_1)}}{4M^4} + \frac{3w_{11}^{(x_1)}}{2M^2} + w_{02}^{(x_1)} \geq 0, \ \frac{5w_{20}^{(x_1)}}{2M^2} + w_{11}^{(x_1)} \geq 0, \ 0 \leq w_{20}^{(x_1)} \leq \frac{1}{M^4},$$

$$8w_{03}^{(x_1)} + 3(4w_{12}^{(x_1)} + 6w_{21}^{(x_1)} + 9w_{30}^{(x_1)}) \geq 0, \ 4w_{12}^{(x_1)} + 14w_{21}^{(x_1)} + 33w_{30}^{(x_1)} \geq 0,$$

$$2w_{21}^{(x_1)} + 7w_{30}^{(x_1)} \geq 0, \ 0 \leq w_{30}^{(x_1)} \leq w_{20}^{(x_1)} \tag{5.21}$$

while the $TR_U^\gamma$ conditions are,

$$-2 \leq \frac{a(2w_{01}^{(x_1)} - 27a(a(aw_{02}^{(x_1)} + w_{11}^{(x_1)}) + w_{20}^{(x_1)})) + 2w_{10}^{(x_1)}}{aw_{01}^{(x_1)} + 1} \leq 2,$$

$$-1 \leq \frac{3(a(9a(a(a(27a(a(w_{03}^{(x_1)} a + w_{12}^{(x_1)}) + w_{21}^{(x_1)}) - 4w_{02}^{(x_1)} + 27w_{30}^{(x_1)} - 4w_{11}^{(x_1)}) - 4w_{20}^{(x_1)}) + w_{01}^{(x_1)}) + 1)}{aw_{01}^{(x_1)} + 1} \leq 3,$$

$$\tag{5.22}$$

where as before we have used the notation $\frac{\mathcal{W}_{p,q}^{(x_1)}}{\mathcal{W}_{1,0}^{(x_1)}} = w_{pq}^{(x_1)}$ and the range of $a$ has been specified in (5.18). The coefficients $w_{ij}^{(x_1)}$ are related to the EFT expansion as follows ,

$$
\begin{aligned}
w_{01}^{(x_1)} &= \frac{-3g_3 - x_1 f_3}{2g_2 + 2x_1 f_2}, & w_{02}^{(x_1)} &= \frac{3(g_{6,1} - g_{6,2} + g_{6,3}) + x_1 f_{6,1}}{2g_2 + 2x_1 f_2} \\
w_{20}^{(x_1)} &= \frac{2(g_{4,1} + 2g_{4,2}) + x_1 4f_4}{2g_2 + 2x_1 f_2}, & w_{11}^{(x_1)} &= \frac{(-5g_{5,1} - 3g_{5,2}) - x_1 2f_5}{2g_2 + 2x_1 f_2},
\end{aligned}
\tag{5.23}
$$

where, $\mathcal{W}_{1,0}^{(x_1)} = 2g_2 + 2x_1 f_2$. Before solving these constraints and getting bounds, we want to point some salient features of our inequalities. The positivity of $\mathcal{W}_{1,0}$ (5.13) gives us:

$$g_2 + x_1 f_2 \geq 0 , \tag{5.24}$$

In particular this translates to $g_2 \pm f_2 \geq 0$ in other words $a_1, a_2 \geq 0$. After expanding $F_2 + x_1 F_1$ in $\tilde{z}, a$ we can use relation (3.53) which translates to the following:

$$-27a^2(-2g_2 - 2x_1 f_2 + 3ag_3 + ax_1 f_3) < 0\,.$$

Firstly we note that if $f_2 = \pm g_2$ then, since the above relation has to hold for all $x_1 \in [-1, 1]$ and all $-\frac{5M^2}{37} < a < \frac{2M^2}{3}$, we get $f_3 = \pm 3g_3$, the reasoning is as follows. Suppose $f_2 = \pm g_2$ then by looking at $x_1 = \mp 1$ we get

$$a(3g_3 \mp f_3) > 0, \ \forall -\frac{5M^2}{37} < a < \frac{2M^2}{3}\,.$$

which gives us the result. In particular for the $f_2 = g_2$ case we note that if we truncate to 6-derivatives there is no difference between the massless scalar case and this one since $F_1 = F_2$. This gives us

$$\frac{-3.44}{M^2} < \frac{f_3}{f_2} < \frac{1}{M^2} \tag{5.25}$$

Secondly if $g_2 + x_1 f_2 \neq 0$ then from (5.25) we have

$$\frac{-4.902}{M^2} < \frac{g_3 + x_1 \frac{f_3}{3}}{g_2 + x_1 f_2} < \frac{1}{M^2} \tag{5.26}$$

These can be compared with the results in table 1 in [22] and we can see that there is decent agreement. We can in fact use the above relations to get region plots as shown in the figure below. We have benchmarked where different theories lie in this allowed space of EFT's. These regions can also be compared with the ones in figure 1 in [22] and we note that our method gives a rectangular region for the left figure whereas the right figure is identical.

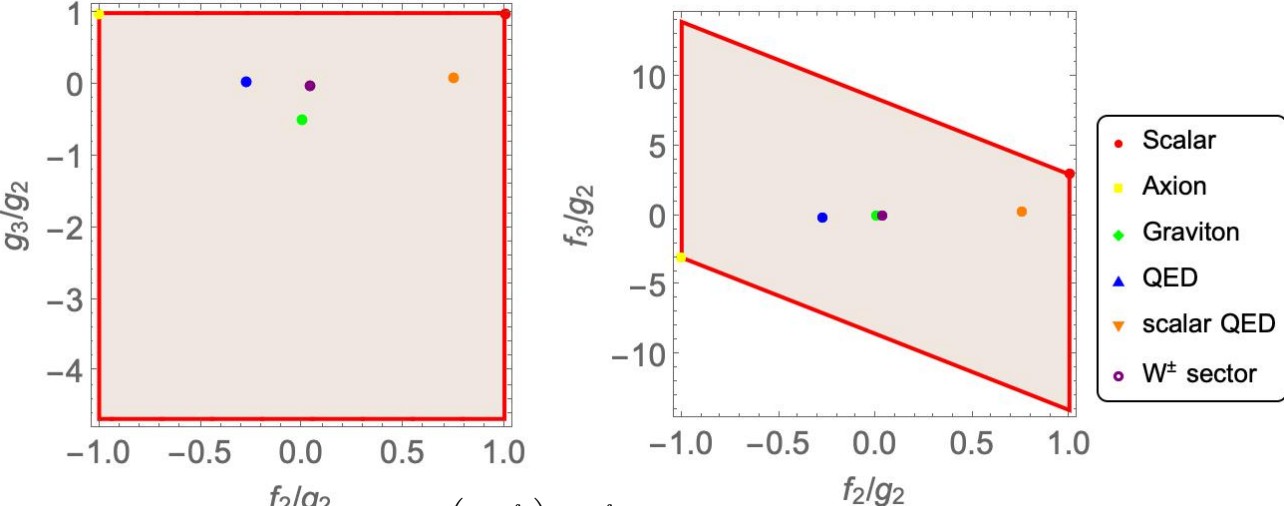

**Figure 2:** The allowed regions in the $\left(\frac{g_3}{g_2}, \frac{f_3}{f_2}\right)$ vs $\frac{f_2}{g_2}$ space with scalar,axion,graviton,QED,scalar QED, $W^\pm$ sector benchmarked.

Furthermore, whenever we have $f_2 = kg_2$ with $k \in [0, 1]$ we can see the space of allowed theories as in this case by choosing a suitable $x_1$ one can make $g_2 + x_1 f_2 = 0$. The plot is shown below.

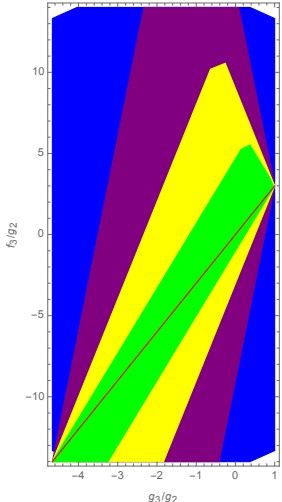

**Figure 3:** The space of allowed regions in the $\left(\frac{f_3}{f_2}, \frac{g_3}{f_2}\right)$ for $f_2 = kg_2$ with $k = 0, 0.25.0.5, 1$ corresponding to blue, purple,yellow,green and red respectively.

By working out the $n = 3$ $PB_C^\gamma$ and $TR_U^\gamma$ conditions explicitly we obtain the following values for $w_{pq}^{(x_1)}$ listed in table 3. A comparative plot for the the first few higher derivative coefficients is given in figure 4. As before in the Wilson coefficients $w_{pq}^{x_1}$, the region $f_2 = \pm g_2$ is special and must be treated with caution. From (5.21), it immediately follows that consistency of the equations for all $x_1 \in [-1, 1]$, enforces the relations of the form $f_i = k \sum_j g_{i,j}$ for $i, J > 2$ whenever $f_2 = \pm kg_2$.

| $w_{p,q}^{(x_1)} = \frac{\mathcal{W}_{p,q}^{(x_1)}}{\mathcal{W}_{1,0}^{(x_1)}}$ | $(TR_U + PB_C)^{min}$ | $(TR_U + PB_C)^{max}$ |
|---|---|---|
| $w_{01}^{(x_1)}$ | $-1.5$ | $7.353$ |
| $w_{20}^{(x_1)}$ | $0$ | $1$ |
| $w_{02}^{(x_1)}$ | $-11.029$ | $4.368$ |
| $w_{11}^{(x_1)}$ | $-2.5$ | $6.353$ |
| $w_{03}^{(x_1)}$ | $-18.479$ | $64.601$ |
| $w_{12}^{(x_1)}$ | $-84.255$ | $15.980$ |
| $w_{21}^{(x_1)}$ | $-3.5$ | $28.1121$ |
| $w_{30}^{(x_1)}$ | $0$ | $1$ |

**Table 3:** A list of bounds obtained using our results $TR_U$ up to $n = 3$ in the normalisation $M^2 = 1$.

In the above table $w_{20}^{(x_1)}$ corresponds to $\frac{g_{4,1}+2g_{4,2}}{g_2}$ and its range is exactly the one obtained in [22], we also have bounds on 10 derivative terms which were not given in [22].

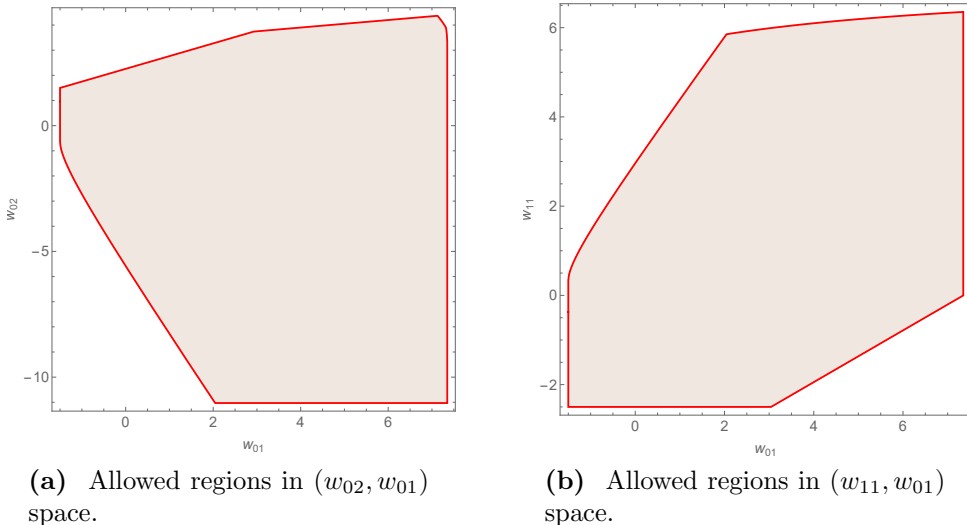

**(a)** Allowed regions in $(w_{02}, w_{01})$ space.

**(b)** Allowed regions in $(w_{11}, w_{01})$ space.

**Figure 4:** Plots for ($w_{11}$ and $w_{02}$) vs $w_{01}$ for $TR_U^\gamma$ upto $n = 3$ .

## 6  Graviton bounds

In this section we will be considering parity preserving graviton amplitudes. We would like to consider the same combination of amplitudes as in the photon case since unitarity guarantees the positivity of these combinations. However, the low energy expansion of $F_2^h(s_1, s_2, s_3) = \sum_{i=1,3,4} T_i(s_1, s_2, s_3)$ starts only at 8-derivatives (the first regular term is the one from $R^4$) which translates to the low energy expansion in $\tilde{z}$ starting from $\tilde{z}^2$ order. Such a function cannot be typically real as can be seen using the following simple argument. Suppose we have a typically-real function $f(z)$ which has a Taylor expansion $f(z) = z^2 + a_3 z^3 + \cdots$ around the origin. In a small neighbourhood of $z = 0$ the leading term is the dominant one and we have $\Im f(z)\Im z > 0 \implies r^2 \sin 2\theta \sin\theta > 0$ for all $z = re^{I\theta}$ and $\theta \in (0, \pi) \cup (\pi, 2\pi)$, this however is not possible as $\sin 2\theta$ changes sign in the upper/lower half plane but $\sin\theta$ does not. Thus our hypothesis that $f(z)$ is typically real is incorrect.

Thus our methods will not directly apply to these combinations. For our purposes we will considering the modified combination:

$$
\begin{aligned}
M^h(s_1, s_2, s_3) &= \tilde{F}_2^h(s_1, s_2, s_3) \\
&= \left( \frac{T_1^h(s_1, s_2, s_3)}{s_1^2} + \frac{T_3^h(s_1, s_2, s_3)}{s_3^2} + \frac{T_4^h(s_1, s_2, s_3)}{s_2^2} \right)
\end{aligned}
\tag{6.1}
$$

As can be readily checked the above combination $\tilde{F}_2^h(s_1, s_2, s_3)$ does not have any additional low energy spurious poles, is fully crossing symmetric and obeys the same $o(s^2)$ Regge growth we demand for $F_i^h(s_1, s_2, s_3)$. Thus $\tilde{F}_2^h(s_1, s_2, s_3)$ also satisfies the crossing symmetric dispersion (3.8) [17]. Furthermore it has the low energy expansion given by

$$
\tilde{F}_2^h(s_1, s_2, s_3) = \mathcal{W}_{1,0}^{(f)} x + \mathcal{W}_{0,1}^{(f)} y + \mathcal{W}_{1,1}^{(f)} xy + \mathcal{W}_{2,0}^{(f)} x^2 + \cdots .
\tag{6.2}
$$

We can also consider $F_1^h(s_1, s_2, s_3)$ which[18] has an expansion

$$
F_1^h(s_1, s_2, s_3) = \mathcal{W}_{0,1}^{(g)} y + \mathcal{W}_{1,1}^{(g)} xy + \mathcal{W}_{2,0}^{(g)} x^2 + \cdots .
\tag{6.3}
$$

---

[17] We have verified that this is indeed true for all the 4-graviton string amplitudes for details see B.

[18] As for the photon case we could have considered $\tilde{F}_2^h(s_1, s_2, s_3) + x_1 F_1^h(s_1, s_2, s_3)$ for $x_1 \in [-1, 1]$ however this leads to a spectral coefficient $\frac{\rho_1}{s^{1'2}} + \rho_2$ which doesn't seem to have a fixed sign from unitarity alone $\rho_1 \geq 0, \rho_1 \pm \rho_2 \geq 0$. We shall use a different method to bound $F_1^h(s_1, s_2, s_3)$.

We shall not explore this case in the current work. When we write the above expansions we have a low-energy gravitational EFT in mind

$$\mathcal{L} = \frac{-2}{\kappa^2}\sqrt{-g}R + 8\frac{\beta_{R^3}}{\kappa^3}R^3 + 2\frac{\beta_{R^4}}{\kappa^4}C^2 + \frac{2\tilde{\beta}_{R^4}}{\kappa^4}\tilde{C}^2 + \cdots, \tag{6.4}$$

where $R$ is the Ricci scalar, $\kappa^2 = 32\pi G$ and $C = R^{\mu\nu\kappa\lambda}R_{\mu\nu\kappa\lambda}$, $\tilde{C} = \frac{1}{2}R^{\mu\nu\alpha\beta}\epsilon_{\alpha\beta}^{\gamma\delta}R_{\gamma\delta\mu\nu}$ and the metric $g_{\mu\nu} = \eta_{\mu\nu} + h_{\mu\nu}$ is given in-terms of the gravitational field $h_{\mu\nu}$. We subtract out the poles corresponding to the $R$ and $R^3$ terms and look at the low energy expansion of the rest of the amplitude. The Wilson coefficients of the low-energy expansion of the amplitudes are related to the parameters in the gravitational EFT Lagrangian such as

$$\mathcal{W}_{1,0}^{(f)} = \frac{\beta_{R^4} + \tilde{\beta}_{R^4}}{\kappa^4}. \tag{6.5}$$

## 6.1  Wilson coefficients and Locality constraints: $PB_C^h$

The local low energy expansion of the amplitude (6.1) can be written as

$$\tilde{F}_2^h(s_1, s_2) = \sum_{p,q=0}^{\infty}\mathcal{W}_{p,q}^{(f)}x^p y^q = \sum_{p,q=0}^{\infty}\mathcal{W}_{p,q}^{(f)}x^{p+q}a^q \tag{6.6}$$

where we have used $a = y/x$ and $\mathcal{W}_{0,0}^{(h)} = 0$. We can solve for the $\mathcal{W}_{p,q}^{(h)}$ by expanding around $a = 0$ and comparing powers of $x, a$. We obtain,

$$\begin{aligned}
\mathcal{W}_{n-m,m}^{(f)} &= \int_{M^2}^{\infty}\frac{ds_1}{2\pi s_1^{2n+m+1}}\sum_{J=0,2,4,\cdots}(2J+1)\tilde{a}_J^{(1)}(s_1)\mathcal{K}_{n,m}^{J,1}, \\
&+ \int_{M^2}^{\infty}\frac{ds_1}{2\pi s_1^{2n+m+1}}\sum_{J=4,6,\cdots}(2J+1)\tilde{a}_J^{(2)}(s_1)\hat{\mathcal{K}}_{n,m}^{J,2}, \\
&+ \int_{M^2}^{\infty}\frac{ds_1}{2\pi s_1^{2n+m+1}}\sum_{J=5,7,\cdots}(2J+1)\tilde{a}_J^{(3)}(s_1)\hat{\mathcal{K}}_{n,m}^{J,3}, \\
\hat{\mathcal{K}}_{n,m}^{J,i} &= 2\sum_{j=0}^{m}\frac{(-1)^{1-j+m}q_J^{(j,i)}(1)(4)^j(3j-m-2n)\Gamma(n-j)}{j!(m-j)!\Gamma(n-m+1)}.
\end{aligned} \tag{6.7}$$

where $\tilde{a}_J^{(1)} = \frac{\rho_J^{1,h}}{s_1^2}$, $\tilde{a}_J^{(2)} = \tilde{a}_J^{(3)} = \frac{\rho_J^{3,h}}{s_1^2}$ and $q_J^{(j,i)}(1) = \frac{\partial^j f^{(i)}(\sqrt{\xi})}{\partial\xi^j}\Big|_{\xi=\xi_0=1}$ with $f^{(1)} = d_{0,0}^J$, $f^{(2)} = \frac{d_{4,4}^J(\cos^{-1}(\sqrt{\xi}))}{(1+\sqrt{\xi})^2} + \frac{d_{4,-4}^J(\cos^{-1}(\sqrt{\xi}))}{(1-\sqrt{\xi})^2}$, $f^{(3)} = \frac{d_{4,4}^J(\cos^{-1}(\sqrt{\xi}))}{(1+\sqrt{\xi})^2} - \frac{d_{4,-4}^J(\cos^{-1}(\sqrt{\xi}))}{(1-\sqrt{\xi})^2}$.

*A key difference between the scalar/photon case and the graviton case we are considering now is that the combinations $f^{(i)}$ are no longer positive even for $\xi > 1$.*

However for $\xi = 1$ we can check that $f^{(i)} = 1$ and since the spectral functions $\tilde{a}_J^{(i)} \geq 0$ by unitarity namely $a_J^{(i)} \geq 0$ so this in particular implies

$$\begin{aligned}
\mathcal{W}_{n,0}^{(h)} &= \int_{M^2}^{\infty}\frac{ds_1}{2\pi s_1^{2n+m+1}}\sum_{J=0,2,4,\cdots}(2J+1)\tilde{a}_J^{(1)}(s_1), \\
&+ \int_{M^2}^{\infty}\frac{ds_1}{2\pi s_1^{2n+m+1}}\sum_{J=4,6,\cdots}(2J+1)\tilde{a}_J^{(2)}(s_1), \\
&+ \int_{M^2}^{\infty}\frac{ds_1}{2\pi s_1^{2n+m+1}}\sum_{J=5,7,\cdots}(2J+1)\tilde{a}_J^{(3)}(s_1) \geq 0.
\end{aligned} \tag{6.8}$$

We can see straightforwardly that the above implies

$$0 \leq \mathcal{W}_{n,0}^{(f)} \leq \frac{1}{M^4} \mathcal{W}_{n-1,0}^{(f)}.$$

As alluded to before, the non-positivity of $f^{(i)}$ in (6.7) implies the sign of any term in $J$ expansion is no longer controlled by $\mathcal{K}_{n,m}^{J,i}(s_1)$ alone. So this makes obtaining a closed form for $PB_C^h$ much harder in this case. We can however do this case by case. For $n = 2$ these read:

$$\frac{9w_{20}^{(f)}}{4M^4} + \frac{3w_{11}^{(f)}}{2M^2} + w_{02}^{(f)} \geq 0, \; \frac{5w_{20}^{(f)}}{2M^2} + w_{11}^{(f)} \geq 0, \; 0 \leq w_{20}^{(f)} \leq \frac{1}{M^4},$$

$$(6.9)$$

where $w_{p,q}^{(f)} = \frac{\mathcal{W}_{p,q}^{(f)}}{\mathcal{W}_{1,0}^{(f)}}$. As before the locality constraints for this case are therefore given by

$$\mathcal{W}_{n-m,m}^{(f)} = 0 \quad \forall n < m.$$

$$(6.10)$$

## 6.2 Typically-Realness and Low spin dominance: $TR_U^h$

In this section we try to get $a$ range of $a$ using positivity of the amplitude coupled with locality constraints and typically-realness of the amplitude. The analysis in this case has key differences due to the non-positivity of the $f^{(i)}(\sqrt{\xi})$ even for $\xi > 1$. We know the typically-realness of the amplitude followed from two crucial ingredients namely the regularity of the kernel inside the unit disk and the positivity of the absorptive part. The former remains unchanged the latter however crucially needs the locality constraints to justify now, since $\xi > 1$ is no longer sufficient to guarantee positivity.

$$\left( a \in \left[ -\frac{M^2}{3}, 0 \right) \cup \left( 0, \frac{2M^2}{3} \right] \right) \quad \cap \quad \left( a \; \forall \; A(s_1, ; s_2^{(+)}(s_1, a) \geq 0 \right)_{LSD}.$$

$$(6.11)$$

We can proceed with the LSD analysis as before by including the locality constraints $N_c^h = -\sum_{\substack{n<m \\ m\geq 2}} c_{n,m} \mathcal{W}_{n-m,m}^{(f)} a^{2n+m-3} y$ to it with arbitrary weights $c_{n,m}$'s (6.10).

$$\mathcal{M}(s_i, a) + N_c =$$

$$\int_{M^2}^{\infty} \frac{ds_1}{2\pi s_1} \sum_{J=0,2,4,\cdots} (2J+1) \tilde{a}_J^{(1)}(s_1) \left[ f_J^{(1)}(\xi) - \sum_{\substack{n<m \\ m\geq 2}} c_{n,m} \hat{\mathcal{K}}_{n,m}^{J,1} \frac{a^{2n+m} H(a;s_i)}{(s_1')^{2n+m} H(s_1',s_i)} \right] H(s_1',s_i)$$

$$+ \int_{M^2}^{\infty} \frac{ds_1}{2\pi s_1} \sum_{J=4,6,\cdots} (2J+1) \tilde{a}_J^{(2)}(s_1) \left[ f_J^{(2)}(\xi) - \sum_{\substack{n<m \\ m\geq 2}} c_{n,m} \hat{\mathcal{K}}_{n,m}^{J,2} \frac{a^{2n+m} H(a;s_i)}{(s_1')^{2n+m} H(s_1',s_i)} \right] H(s_1',s_i)$$

$$+ \int_{M^2}^{\infty} \frac{ds_1}{2\pi s_1} \sum_{J=5,7,\cdots} (2J+1) \tilde{a}_J^{(3)}(s_1) \left[ f_J^{(3)}(\xi) - \sum_{\substack{n<m \\ m\geq 2}} c_{n,m} \hat{\mathcal{K}}_{n,m}^{J,3} \frac{a^{2n+m} H(a;s_i)}{(s_1')^{2n+m} H(s_1',s_i)} \right] H(s_1',s_i) \geq 0,$$

$$(6.12)$$

where $\hat{\mathcal{K}}_{n,m}^{J,i}$ has been defined in (6.7). As before $\xi_{min}$ is determined by the maximum lower bound obtained by considering the positivity of three different classes of inequalities namely corresponding to the coefficients of $\tilde{a}_J^{(i)}$ for $i = 1, 2, 3$. We can also determine $\xi_{max}$ now which is determined by the minimum upper bound obtained by considering the positivity of the same three classes of inequalities. This exercise leads us to $\xi_{min}^h = 0.593$ and $\xi_{max}^h = 3$ for the graviton EFT when we consider all locality

constraints up to $2n + m \leq 12$ and $J_{max} \leq 20$. Using the relation $\frac{\xi_{min}^2 - 1}{\xi_{min}^2 + 3} M^2 < a < \frac{\xi_{max}^2 - 1}{\xi_{max}^2 + 3} M^2$ and (6.11), we obtain,

$$\textbf{Graviton} : - 0.1933 M^2 < a^h < \frac{2M^2}{3} \,,$$

(6.13)

We would now like to show that this is indicative of Spin-2 dominance for the graviton case. Since in the set of polynomials $\{d_{0,0}^J, \frac{d_{4,4}^J(cos^{-1}x)}{(1+x)^2} \pm \frac{d_{4,-4}^J(cos^{-1}x)}{(1-x)^2}\}$ the latter two elements do not have straightforward positivity properties for general $J$. The identification of the critical spin $J_c$ is more complicated and needs more detailed consideration in this case. A key difference between the scalar/photon cases and the graviton case we are looking at now is that both the upper and lower bound of $a$ can change. Let us recall how that happens- the condition $TR_U$ tells us that the allowed range of $\xi \in [0,3]$. The overlap of this region with the positive part of the absorptive part gave us the required range of $\xi$ to be used in our analysis. In the analogous exercise for the photons and scalars, we had truncated the partial wave sum to a finite cut-off in spin and so then the range of $\xi$ was determined by what range for which these polynomials were positive. We had determined the lower range of $\xi$ to be given by the largest root of the Wigner-$d$ combinations that appear with respective spectral coefficients- this was usually such that $\xi_{min} < 1$. The upper range of $\xi$ was automatically determined by the $TR_U$ conditions since the relevant Wigner-$d$ matrices were manifestly positive for $\xi > 1$— in other words there were no restrictions on the upper limit of $\xi$ from the Wigner-$d$ polynomials. The story for gravitons remains the same for the lower bound for $\xi$, but we note the following changes for the upper bound.

To illustrate this point, notice that the Wigner-$d$ combination $f^{(3)} = \frac{d_{4,4}^J(\cos^{-1}(\sqrt{\xi}))}{(1+\sqrt{\xi})^2} - \frac{d_{4,-4}^J(\cos^{-1}(\sqrt{\xi}))}{(1-\sqrt{\xi})^2}$ is not always positive for $\xi > 1$ for $J \geq 9$. Therefore if we assume $J_c = 9$, the upper limit for $\xi$ (and hence $a$) also changes along with the lower limit. As an example we present shortening of the positive regions for $f^{(3)}$ for $J = 9, 11$ in figure 5. We present the allowed range of $a$ as a function of $J_c$ in the form of a table below.

| $J_c$ | Graviton |
|---|---|
| 2 | $-0.2 M^2 < a^h < \frac{2M^2}{3}$ |
| 4 | $-0.069 M^2 < a^h < \frac{2M^2}{3}$ |
| 6 | $-0.034 M^2 < a^h < \frac{2M^2}{3}$ |
| 9 | $-0.014 M^2 < a^h < \frac{19641 M^2}{140000}$ |

We can see the spin-2 dominance clearly for the graviton case from the above table. We have also illustrated this for the case of the type-II string amplitude in appendix(G).Note that the locality constraints play an important role in maintaining the positivity of the amplitude despite the Wigner-$d$ combination $f^{(3)}$ not having nice positivity properties. This is not a surprise since the locality constraints encode the details of the theory and put constraints on allowed spectral densities that appear in each sector. It would be interesting to explore the detailed implications of locality constraints in future.

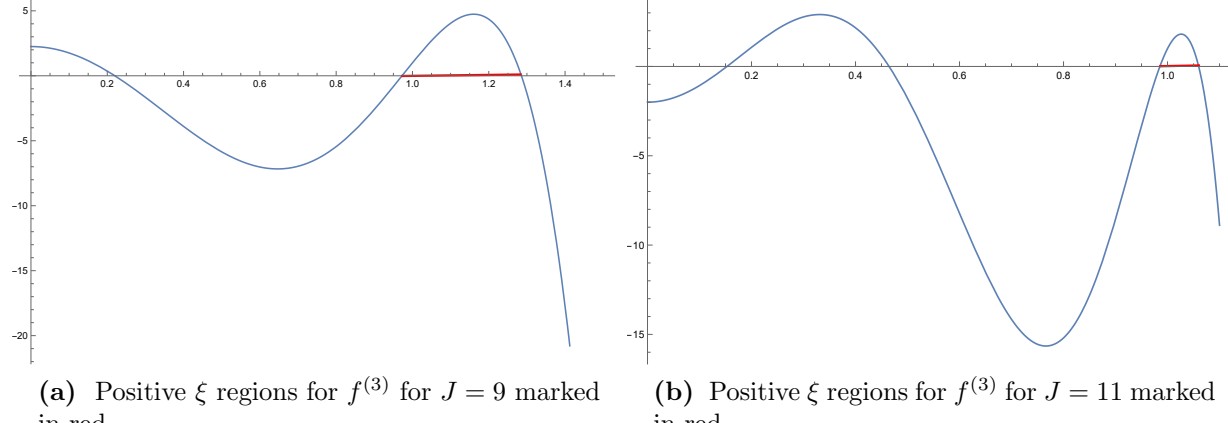

**(a)** Positive $\xi$ regions for $f^{(3)}$ for $J = 9$ marked in red.

**(b)** Positive $\xi$ regions for $f^{(3)}$ for $J = 11$ marked in red.

**Figure 5:** A comparative plot of changing regions of positivity in $\xi$ with spin for $f^{(3)}$.

Therefore for this range of $a$ in (6.13), we can impose the Bieberbach-Rogosinski bounds of subsection 3.6 on the Wilson coefficients $\mathcal{W}^{(f)}_{n-m,m}$ —these constraints are called $TR^f_U$.

## 6.3 Bounds

In this section we put the bounds on the low energy EFT expansion which is parametrized by,

$$\tilde{F}^h_2 = 2xf_{0,0} + 3yf_{1,0} + 2x^2f_{2,0} + xy\left(2f_{3,1} - f_{3,0}\right) + y^2\left(-3f_{4,0} - 3f_{4,1} + 9f_{4,2}\right) + 2x^3f_{4,0} + \cdots \quad (6.14)$$

where in terms of parametrization of [7],

$$T^h_3(s_1, s_2, s_3) = s_3^4\left(\sum_{i=0}^{\infty} f_{2i,i}s_2^i s_1^i + \sum_{i=1}^{\infty}\sum_{j=0}^{\lfloor \frac{i}{2} \rfloor} f_{i,j}(s_2^{i-j}s_1^j + s_1^{i-j}s_2^j)\right). \quad (6.15)$$

We have explicitly,

$$w^{(f)}_{1,0} = 2f_{0,0}, \quad w^{(f)}_{0,1} = 3f_{1,0}, \quad w^{(f)}_{2,0} = 2f_{2,0}, \quad w^{(f)}_{1,1} = (2f_{3,1} - f_{3,0})$$
$$w^{(f)}_{0,2} = (-3f_{4,0} - 3f_{4,1} + 9f_{4,2}), \quad w^{(f)}_{3,0} = 2f_{4,0}. \quad (6.16)$$

We demonstrated in the previous subsection that due to positivity and typical realness of the amplitudes, we can put two sided bounds on Wilson coefficients. Using (3.53) we have,

$$-1.5 \leq w^f_{01} \leq 5.17331 \quad (6.17)$$

where $w^f_{01} = \frac{w^{(f)}_{0,1}}{w^{(f)}_{1,0}}$, which implies $-1 \leq \frac{f_{1,0}}{f_{0,0}} \leq 3.44$.

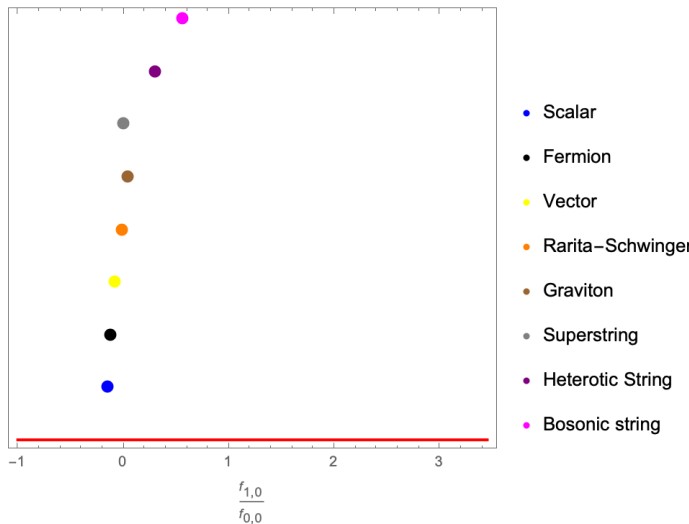

**Figure 6:** A line plot to show the allowed range of $f_{1,0}$ vs $f_{0,0}$ with scalar, fermion, photon, gravitino, graviton, super string, Heterotic string and bosonic string sector benchmarked.

The above figure is the crossing symmetric analogue of figure 8 in [7]. For $n = 2$ $TR_U^h$ and $PB_C^h$, we obtain table 4 (in units of $M^2 = 1$),

| $w_{pq}^f = \frac{w_{p,q}^{(f)}}{w_{1,0}^{(f)}}$ | $(TR_U + PB_C)^{min}$ | $(TR_U + PB_C)^{max}$ |
|:---:|:---:|:---:|
| $w_{01}^f$ | $-1.5$ | $5.1733$ |
| $w_{02}^f$ | $-7.7600$ | $3.8273$ |
| $w_{20}^f$ | $0$ | $1$ |
| $w_{11}^f$ | $-2.5$ | $4.1734$ |

**Table 4:** A list of graviton bounds obtained using our results $TR_U$ up to $n = 2$ in the normalisation $M^2 = 1$.

We note that terms such as $f_{2,1}$ or $f_{1,1}$ vanish when we consider fully crossing symmetric combinations as these are proportional to $s_1 + s_2 + s_3 = 0$ thus we will not be able to bound these using the current combinations[19] we are looking at. However, using our method, we can bound coefficients like $\frac{f_{1,0}}{f_{0,0}}$ for which no non-trivial bounds were found using the fixed-$t$ dispersion relation, to the best of our knowledge. The region carved out by the Wilson coefficients with their respective data points for various theories is given. The data has been obtained from [7].

---

[19]However by looking at $F_4(s_1, s_2, s_3)$ and $F_5(s_1, s_2, s_3)$ these terms do appear so we can bound them in principle. We do not attempt to do this in our current work.

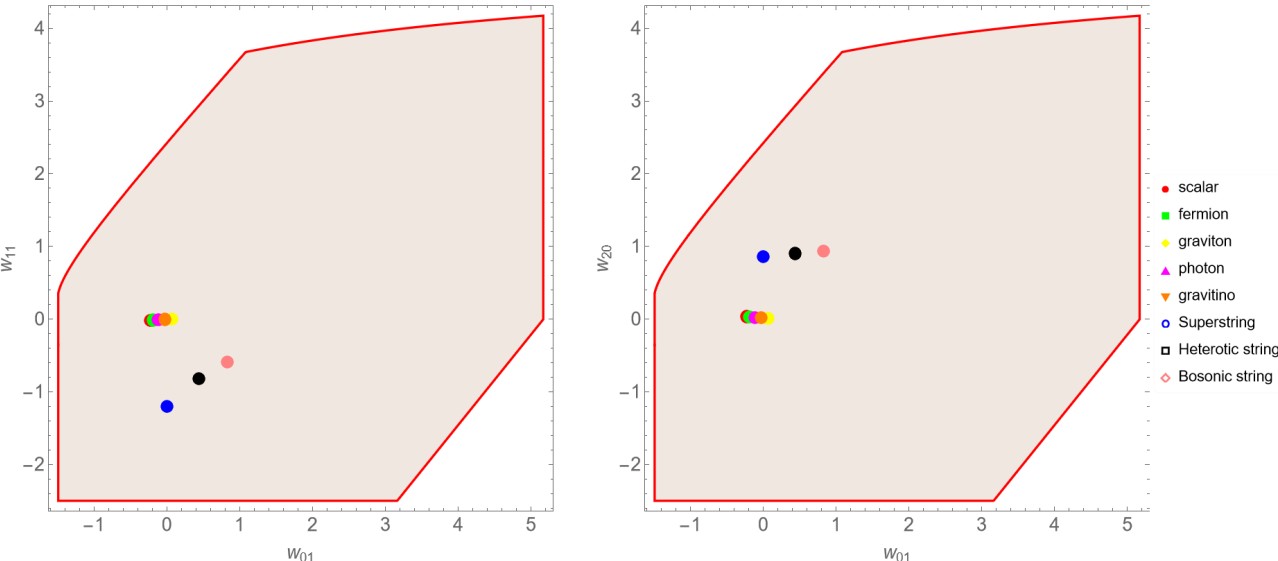

**Figure 7:** The allowed regions in the $(w_{11}, w_{02})$ vs $w_{01}$ space with scalar, fermion, photon, gravitino, graviton, superstring, Heterotic string and bosonic string sector benchmarked.

# 7 Discussion

In this paper, we set up a crossing symmetric dispersion relation for external particles carrying spin. Given a basis of amplitudes, which transform under crossing, we give a general prescription to construct crossing symmetric amplitudes relevant for CSDR from them. We demonstrated this construction explicitly for massless photons, gravitons and massive Majorana fermion helicity amplitudes in $d = 4$. We then use the CSDR for certain photons and graviton crossing symmetric amplitudes and put bounds on low energy Wilson coefficients. Our analysis suggested that the positivity of the absorptive part is dominated by partial waves of low-lying spins—we found indications of spin-3 LSD for photons and spin-2 for gravitons (see [2, 7]). Using the typically-realness property of the amplitude, the Wilson coefficients satisfied the Bieberbach-Rogosinski ($BR$) bounds. We supplemented the $BR$ bounds using certain additional positivity conditions to get tighter bounds in some cases. The photon bounds are in good agreement with existing results in literature. We dealt with the graviton amplitude separately since the low energy EFT expansion starts from the eighth order in derivatives for the crossing symmetric amplitude we consider. In order for the low energy expansion to be typically real, we considered a modified amplitude which then had the requisite properties. Similar to the photon case, we wrote down the locality constraints in closed form and analysed certain bounds. One would like to tackle several problems, some of which we outline below.

- Compared to the fixed-$t$ dispersion relation, the non-linear unitarity constraints arising from the crossing symmetric dispersion relation is mathematically different. In the analysis of the recently resurrected (numerical) S-matrix bootstrap, e.g., [48, 51], the starting point is a crossing symmetric basis that captures some of the known analytic properties of the amplitude. The crossing symmetric dispersion relation gives a systematic crossing symmetric starting point where the parametrisation of the amplitude now is in terms of the absorptive partial wave amplitudes. We envisage some simplification arising from this, since instead of a two variable parametrisation, one now can focus on a one variable one. It will be very important to examine this systematically in the near future.

- Since our approach enables us to write down the locality/null constraints in a closed form, it will be interesting to attempt a systematic derivation of the stronger version of the low spin dominance conjectured in [7].

- We have not attempted to use the non-linear constraints arising from Toeplitz determinants [25]. This should further constrain the space of EFTs.

- We hope for a consolidated treatment of graviton positivity conditions $PB_C^h$ in the future. The main reason for the failure of scalar $PB_C$ ansatz stems from the fact that the $p^{j,i}$s are not explicitly positive for all spins.

- In this work we considered the most natural combinations of helicity amplitudes which are simple and suffice to illustrate our method. There are other combinations which we can consider. We list below a couple of them:

$\mathbf{F_4}$&$\mathbf{F_5}$ : We can consider $F_3^\alpha(s_1, s_2, s_3)$, $F_4^\alpha(s_1, s_2, s_3)$ and $F_5^\alpha(s_1, s_2, s_3)$. In particular note that for the photon case we have not been able to put constraints on $g_{4,1}$ and $g_{4,2}$ separately. This is an artifact of the construction of $F_1^\gamma$ and $F_2^\gamma$, where the coefficients $g_{4,1}$ and $g_{4,2}$ appear only in the combination $g_{4,1} + 2g_{4,2}$. However in the low energy expansion of $F_4^\alpha$ and $F_5^\alpha$ the coefficients $g_{4,1}$ and $g_{4,2}$ do appear separately see appendix (F).Thus considering these combinations will help us bound these coefficients. For the photon a preliminary analysis assuming spin-3 dominance shows that both $F_4^\alpha(s_1, s_2, s_3)$ and $F_5^\alpha(s_1, s_2, s_3)$ have suitable ranges of $a$ for which their absorptive parts are positive namely

$$\mathbf{F_4} : -\frac{M^2}{5} < a < \frac{M^2}{2} \,,$$

$$\mathbf{F_5} : -\frac{M^2}{5} < a < 0 \,.$$

$\mathbf{F_1^h}$ : For gravitons we can consider the combination

$$F_1^h(s_1, s_2, s_3) + F_2^h(s_1, s_2, s_3) + \tilde{F}_2^h(s_1, s_2, s_3)$$

to bound the Wilson coefficients appearing in $F_1^h(s_1, s_2, s_3)$ (see (6.3)).

**Parity violating amplitudes:** As spelt out in the appendix C.14 (see below (C.23)), the spectral functions for the parity violating amplitudes do not seem to obey definite positivity conditions and some non-linear constraints of the kind dealt in [7] might be useful.

We leave a more careful analysis and GFT bounds from these combinations for future forays.

- We considered helicity amplitudes for spinning particles in our analysis. There are other formulations for handling spinning amplitudes as well. One such is transversity amplitudes [52]. In transversity formalism, the spin is quantised normal to the plane of scattering. In this formalism, the crossing equations are diagonalised. This, however, comes at the price that the unitarity is now straightforward. However, one can still work the unitarity exploiting the relation between the transversity amplitude and the helicity amplitude, the former being a linear combination of the latter. The unitarity consideration, along with fixed transfer dispersion relations, was employed to obtain positivity bounds for transversity amplitudes for EFTs in [53]. However, it is not clear how to translate these positivity bounds to constraints on EFT parameters like Wilson coefficients. Therefore, it is worth investigating how these positivity bounds can be used to constrain the EFT parametric space. Further, it will be interesting to consider applying the crossing-symmetric dispersive techniques to these transversity amplitudes.

- It should be possible to extend our analysis to Mellin amplitudes for CFTs building on [30]. This would be relevant for studying EFT bounds in AdS space.

- An important assumption of our work is that we are only analysing low energy effective field theories at the tree level. This is justifiable for EFTs having weakly coupled UV completion. Even in this situation, it will be interesting to know how these bounds get modified, including massless loops [54]. This is beyond the scope of our present framework since we expand our low

energy effective amplitude around $s, t = 0$. In crossing symmetric dispersion relation, it is not natural to expand in this forward limit. So our set-up might be better suited to address this issue, and we leave this exciting possibility for future exploration.

# Acknowledgements

We thank Ahmadullah Zahed and Debapriyo Chowdhury for discussions. SDC is supported by a Kadanoff fellowship at the University of Chicago, and in part by NSF Grant No. PHY2014195. KG is supported by ANR Tremplin-ERC project FunBooTS. KG thanks ICTS, Bangalore for hospitality during the intial stage of this work. AS acknowledges partial support from a SPARC grant P315 from MHRD, Govt of India.

# A    Representation theory of $S_3$: A crash course

In this appendix, we present a short self contained review of $S_3$ representations following [39]. We can represent the three irreps of $S_3$ by the following young diagrams.

$$\mathbf{1_S} = \square\square\square \qquad \mathbf{1_A} = \begin{array}{c}\square\\\square\\\square\end{array} \qquad \mathbf{2_M} = \square\square. \tag{A.1}$$

where $\mathbf{1_S}$ is the one dimensional totally symmetric representation, $\mathbf{1_A}$ is the one dimensional totally anti-symmetric representation and $\mathbf{2_M}$ is the mixed symmetry two dimensional representation. Given an representation of $S_3$, we can easily decompose it to the irreducible sub spaces of $\mathbf{1_S}$, $\mathbf{1_A}$ and $\mathbf{2_M}$ representations using the respective projectors. Denoting the generators for $S_3$ by $P_{12}$ and $P_{23}$ (where $P_{ij}$ denotes interchange of particles in $i$ and $j$ th position in a set $(123)$ ), the projectors for the totally symmetric and anti-symmetric subspaces are given by

$$P_{\mathbf{1_S}} = \frac{(1 + P_{12} + P_{23} + P_{13} + P_{23}P_{12} + P_{12}P_{23})}{6}$$
$$P_{\mathbf{1_A}} = \frac{(1 - P_{12} - P_{23} - P_{13} + P_{23}P_{12} + P_{12}P_{23})}{6} \tag{A.2}$$

where $P_{13} = P_{23}P_{12}P_{23}$. The formulae (A.2) make it clear that complete symmetrization and anti symmetrization lead to projection onto the $\mathbf{1_S}$ and $\mathbf{1_A}$ subspace, respectively, while the part that transforms in the $\mathbf{2_M}$ representation is annihilated by both the symmetric and anti-symmetric projectors. The group theory for the action of $S_3$ on the Mandelstam invariants is given by the left action of $S_3$ on itself. The $\mathbf{6}_{\text{left}}$ generated by the left action of $S_3$ onto itself can be decomposed as.

$$\mathbf{6}_{\text{left}} = \mathbf{1_S} + 2.\mathbf{2_M} + \mathbf{1_A}. \tag{A.3}$$

Note the appearance of two $\mathbf{2_M}$ subspaces, which differ from one another because they have different $\mathbb{Z}_2$ charges. The explicit projectors for these two (two-dimensional) sub-spaces can be constructed as follows. The projectors for the two-dimensional subspace of positive $\mathbb{Z}_2$ charge are

$$P_{\mathbf{2_{M+}}}^{(1)} = \frac{1 + P_{23}}{2} - \frac{(1 + P_{12} + P_{23} + P_{13} + P_{23}P_{12} + P_{12}P_{23})}{6}$$
$$P_{\mathbf{2_{M+}}}^{(2)} = \frac{P_{23}P_{12} + P_{13}}{2} - \frac{(1 + P_{12} + P_{23} + P_{13} + P_{23}P_{12} + P_{12}P_{23})}{6} \tag{A.4}$$

Note that the above two projectors are respectively symmetric under the action of $\mathbb{Z}_2$ generator $P_{23}$ and $P_{13}$ and hence having a positive $\mathbb{Z}_2$ charge. We note that the projector $P_{\mathbf{2_{M+}}}^{(1)}$ projects to a

subspace which is symmetric under $P_{23}$ while $P_{\mathbf{2}_{\mathrm{M}+}}^{(2)}$ projects to a subspace which is symmetric under $P_{12}$. The projectors for the two dimensional subspace for the negative $\mathbb{Z}_2$ charge (anti-symmetric under $P_{23}$ and $P_{13}$ respectively) are

$$
\begin{aligned}
P_{\mathbf{2}_{\mathrm{M}-}}^{(1)} &= \frac{1 - P_{23}}{2} - \frac{(1 - P_{12} - P_{23} - P_{13} + P_{23}P_{12} + P_{12}P_{23})}{6} \\
P_{\mathbf{2}_{\mathrm{M}-}}^{(2)} &= \frac{P_{23}P_{12} - P_{13}}{2} - \frac{(1 - P_{12} - P_{23} - P_{13} + P_{23}P_{12} + P_{12}P_{23})}{6}
\end{aligned}
\tag{A.5}
$$

To explicitly see the formalism in action, consider an arbitrary function of the Mandelstam invariants $F(s,t,u)$. The various irreducible subspaces are given by[20],

$$
\begin{aligned}
f_{\mathrm{Sym}}(s,t,u) &= \frac{1}{6}\left(F(s,t,u) + F(t,s,u) + F(s,u,t) + F(u,t,s) + F(t,u,s) + F(u,s,t)\right) \\
f_{\mathrm{Anti-sym}}(s,t,u) &= \frac{1}{6}\left(F(s,t,u) - F(t,s,u) - F(s,u,t) - F(u,t,s) + F(t,u,s) + F(u,s,t)\right) \\
f_{\mathrm{Mixed+}}(s,t,u) &= \frac{1}{6}\left(2F(s,t,u) - F(t,s,u) - F(u,s,t) + 2F(s,u,t) - F(u,t,s) - F(t,u,s)\right) \\
f_{\mathrm{Mixed-}}(s,t,u) &= \frac{1}{6}\left(2F(s,t,u) - F(t,s,u) - F(u,s,t) - 2F(s,u,t) + F(u,t,s) + F(t,u,s)\right)
\end{aligned}
\tag{A.6}
$$

We can easily write down examples of such functions built out of polynomials of mandelstam invariants [39, 45, 55].

$$
\begin{aligned}
f_{\mathrm{Sym}}(s,t,u) &= (s^2 + t^2 + u^2)^m (stu)^n \\
f_{\mathrm{Anti-sym}}(s,t,u) &= (s^2 t - t^2 s - s^2 u + su^2 - u^2 t + t^2 u) f_{\mathrm{Sym}}(s,t,u) \\
f_{\mathrm{Mixed+}}(s,t,u) &= \{(2s - t - u)f_{\mathrm{Sym}}(s,t,u),\ (2s^2 - t^2 - u^2)f_{\mathrm{Sym}}(s,t,u)\} \\
f_{\mathrm{Mixed+}}(s,t,u) &= \{(s - u)f_{\mathrm{Sym}}(s,t,u),\ (s^2 - u^2)f_{\mathrm{Sym}}(s,t,u)\}
\end{aligned}
\tag{A.7}
$$

# B  Massless amplitudes: Examples

We expect that the combinations $F_I$ also obey (3.8) since they satisfy all the necessary conditions. We can do some sanity checks by considering a couple of examples. In particular we look at $F_4$, $F_5$ since $F_I$ for $I = 1, 2, 3$ the dispersion relation (3.8) is identical to the scalar case considered in [29].

We first consider the Photon amplitude in superstring theory with a kinematic pre-factor being stripped off for appropriate Regge growth namely:

$$
\begin{aligned}
T_1(s_1, s_2, s_3) &= \frac{\Gamma\left[\frac{-s_2}{2}\right]\Gamma\left[\frac{-s_3}{2}\right]}{\Gamma\left[1 + \frac{s_1}{2}\right]}, \\
T_3(s_1, s_2, s_3) &= \frac{\Gamma\left[\frac{-s_1}{2}\right]\Gamma\left[\frac{-s_2}{2}\right]}{\Gamma\left[1 + \frac{s_3}{2}\right]}, \\
T_4(s_1, s_2, s_3) &= \frac{\Gamma\left[\frac{-s_1}{2}\right]\Gamma\left[\frac{-s_3}{2}\right]}{\Gamma\left[1 + \frac{s_2}{2}\right]}.
\end{aligned}
\tag{B.1}
$$

We can construct (2.23),(2.24) from the above, we need to subtract out the massless poles and this is done by multipliying $F_4$ as defined in (2.23) by an $s_1 s_2 s_3$ factor. We can then check if (3.8) is satisfied

---

[20]We note that $f_{\mathrm{Mixed+}}(s,t,u) + f_{\mathrm{Mixed+}}(t,u,s) + f_{\mathrm{Mixed+}}(u,s,t) = f_{\mathrm{Mixed-}}(s,t,u) + f_{\mathrm{Mixed-}}(t,u,s) + f_{\mathrm{Mixed-}}(u,s,t) = 0$ denoting that they form two dimensional subspaces.

by comparing the exact answer with the result obtained from (3.8) by computing the absorbtive part. Since (B.1) has infinitely many poles at $s_1' = k$ with $k = 2, 4, \cdots$, and each pole $p$ contributes a $-\pi\delta(s_1' - p)$ factor in the absorbtive part thus (3.8) reduces to an infinite sum over all the poles $k$, $k \in 2\mathbb{Z}_+$ which we call $G_I$. We can then compare the results by truncating this sum to some $k_{max}$ (say $k_{max} = 100$) and the results are shown in first row of the plots below.

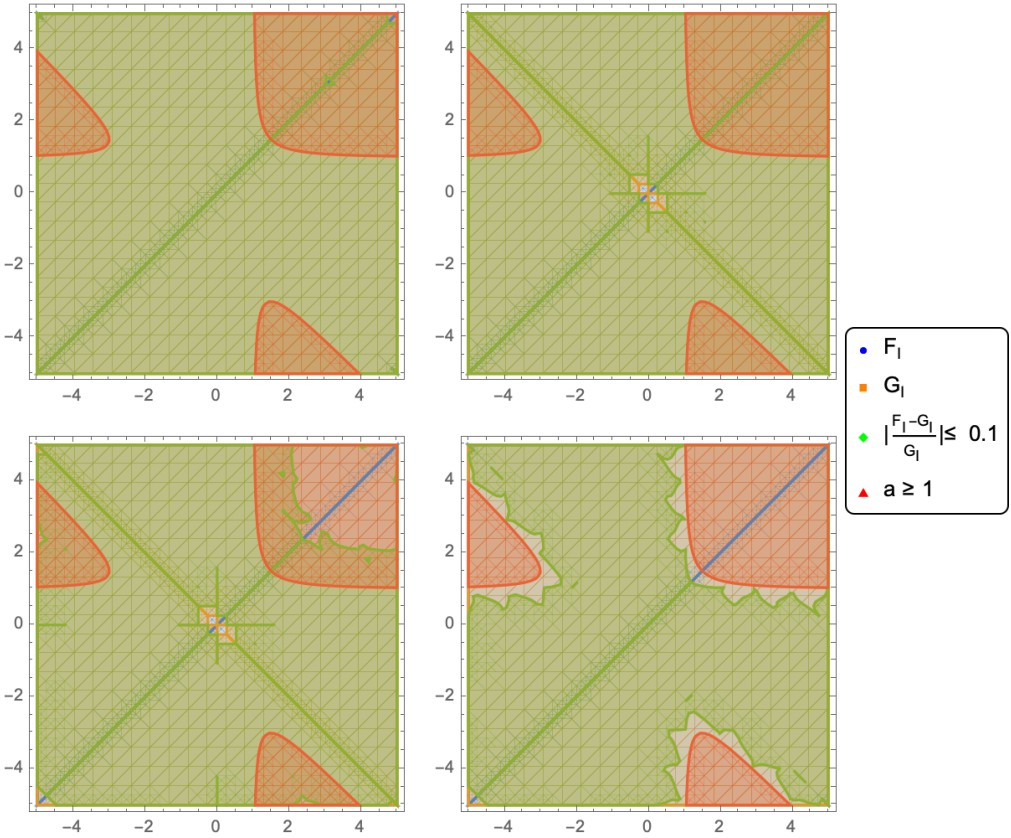

**Figure 8:** A comparison of the crossing symmetric dispersion for the Photon and Graviton cases which are shown in the first and second rows respectively. We have indicated the regions where $F_4, F_5$ differ from their dispersive analogues $G_4, G_5$ by less than 10% in green.

We can also consider the Graviton amplitude from superstring theory (again with appropriate kinematic pre-factors stripped off )

$$T_1(s_1, s_2, s_3) = \frac{\Gamma[-s_1]\,\Gamma[-s_2]\,\Gamma[-s_3]}{\Gamma[1+s_1]\,\Gamma[1+s_2]\,\Gamma[1+s_3]}\left(1 - \frac{s_2 s_3}{s_1+1}\right),$$

$$T_3(s_1, s_2, s_3) = \frac{\Gamma[-s_1]\,\Gamma[-s_2]\,\Gamma[-s_3]}{\Gamma[1+s_1]\,\Gamma[1+s_2]\,\Gamma[1+s_3]}\left(1 - \frac{s_1 s_2}{s_3+1}\right),$$

$$T_4(s_1, s_2, s_3) = \frac{\Gamma[-s_1]\,\Gamma[-s_2]\,\Gamma[-s_3]}{\Gamma[1+s_1]\,\Gamma[1+s_2]\,\Gamma[1+s_3]}\left(1 - \frac{s_1 s_3}{s_2+1}\right). \tag{B.2}$$

To remove the massless poles, we now need to multiply $F_5$ as defined in (2.24) by the factor $s_1 s_2 s_3$, and we can follow the same procedure as the photon case with the only change being that we now have poles at $s_1' = k$ with $k \in \mathbb{Z}_+$.

The results are shown in the second row of the figure above. We note that since (3.8) was written down assuming $o(s_1^2)$ behaviour for large $|s_1|$ and fixed $s_2$, examining the growth of $F_I$ restricts $s_2$ to a region where we should trust the results, e.g., $F_4$ in the photon case has a growth $s_1^{s_2}$ for large $s_1$ which implies we can strictly expect an agreement for only $s_2 < 2$, though we have considered a bigger region in the figure we see that there is an excellent agreement between the dispersion relation and the exact answer.

We have also verified that the crossing symmetric combination in eqn.(6.1) namely

$$\tilde{F}_2^h(s_1, s_2, s_3) = \left( \frac{T_1^h(s_1, s_2, s_3)}{s_1^2} + \frac{T_3^h(s_1, s_2, s_3)}{s_3^2} + \frac{T_4^h(s_1, s_2, s_3)}{s_2^2} \right)$$

for tree-level 4-graviton scattering amplitudes in superstring, Heterotic string and bosonic string theories all obey that the crossing symmetric dispersion relation after subtracting out the massless poles.

# C  Unitarity constraints

In this section we review the unitarity constraints on partial wave amplitudes following [37, 41]. Unitarity constraints can be summarized as positivity of norm of a state $\langle \psi | \psi \rangle \geq 0$. If we have multiple states (say of number $N$), this translates to *positive semi-definiteness* of a $N \times N$ hermitian matrix. In order to see the relation of this statement in context of S-matrices, consider the incoming and outgoing particles as decomposed into irreps of the poincare group. To be more precise (eq (2.21) of [41]),

$$|\kappa_1, \kappa_2\rangle = \int \frac{d^4p}{(2\pi)^4} \theta(p^0)\theta(-p^2) \sum_{i,j} \sum_{J,\lambda} |c, \vec{p}; J, \lambda; \lambda_i, \lambda_j\rangle\langle c, \vec{p}; J, \lambda; \lambda_i, \lambda_j | \kappa_1, \kappa_2\rangle. \tag{C.1}$$

$|\kappa_1, \kappa_2\rangle$ is generic $2-$particle momentum state

$$|\kappa_1, \kappa_2\rangle := |m_1, \vec{p}_1; j_1, \lambda_1\rangle \otimes |m_2, \vec{p}_2; j_2, \lambda_2\rangle, \tag{C.2}$$

where $\vec{p}_i, m_i$ are corresponding $3-$momentum, mass respectively, $p_i^\mu p_{i\mu} = -m_i^2$. $j_i, \lambda_i$ are spin and helicity respectively. For massive particles helicity takes $2j_i + 1$ values, $\lambda_i \in \{-j_i, -j_i + 1, \ldots, j - 1, j\}$, $m_i \neq 0$, while for massless particles it takes two values, $\lambda_i = \pm j_i$, $m_i = 0$. The Poincare $2-$particle irreps $\{|c, \vec{p}; J, \lambda; \lambda_i, \lambda_j\rangle\}$ are states of definite total momenta and total angular momenta, $\vec{p}$ being the total $3-$momentum and $J$ being the total angular momentum with $\lambda$ corresponding $3-$component taking $2J + 1$ values, $\lambda \in \{-J, -J + 1, \ldots, J - 1, J\}$. In particular,

$$\langle c, \vec{p}; J, \lambda; \lambda_i, \lambda_j \mid \kappa_1, \kappa_2\rangle \propto (2\pi)^4 \delta^4(p^\mu - p_1^\mu - p_2^\mu). \tag{C.3}$$

Further, these states are normalized by

$$\langle c', \vec{p}'; J', \lambda'; \lambda_1', \lambda_2' \mid c, \vec{p}; J, \lambda; \lambda_1, \lambda_2\rangle = (2\pi)^4 \delta^4(p'^\mu - p^\mu) \, \delta_{l'l} \delta_{\lambda'\lambda} \delta_{\lambda_1'\lambda_1} \delta_{\lambda_2'\lambda_2}. \tag{C.4}$$

For our purpose, we will work in CoM frame. Thus the states of interest to us are $\{|c, \vec{0}; J, \lambda; \lambda_1, \lambda_2\rangle\}$. Under the action of parity operator $\mathcal{P}$, these states transform as (C.5)

$$\mathcal{P} |c, \vec{0}; J, \lambda; \lambda_1, \lambda_2\rangle = \eta_1 \eta_2 (-1)^{J - j_1 + j_2} |c, \vec{0}; J, \lambda; -\lambda_1, -\lambda_2\rangle. \tag{C.5}$$

Here $\eta_1$ and $\eta_2$ are pure phases, also called *intrinsic parity* associated with the particle, obey the constraint $\eta_i^2 = \pm 1$ (with the negative sign only possible for fermions).

For identical particles we need to take care of the exchange symmetry. This prompts us to define the following states which we will use in our subsequent analysis

$$|c, \vec{0}; J, \lambda; \lambda_1, \lambda_2\rangle_{\text{id}} = \frac{1}{2} \left[ |c, \vec{0}; J, \lambda; \lambda_1, \lambda_2\rangle + (-1)^{J + \lambda_1 - \lambda_2} |c, \vec{0}; J, \lambda; \lambda_1, \lambda_2\rangle \right] \tag{C.6}$$

We also note the relation between the 2-particle reducible state $|\kappa_1, \kappa_2\rangle_{\text{COM}}$ in the COM frame and $|c, \vec{0}; J, \lambda; \lambda_1, \lambda_2\rangle_{\text{id}}$. This is essential in determining the range of $J$ for the irreducible 2-particle reps.

Following [41] we can define the 2-particle reducible state in COM frame as product of eigenstates of the $J_z$ operator.

$$|\kappa_1, \kappa_2\rangle_{\text{COM}} \equiv |(\vec{p}, \theta, \phi)\,; \lambda_1, \lambda_2\rangle \equiv |m_1, \vec{p}; j_1, \lambda_1\rangle \otimes |m_2, -\vec{p}; j_2, \lambda_2\rangle \tag{C.7}$$

where $J_z = L_z + j_z^1 + j_z^2$ ($L_z$ is the orbital angulam momentum, $j_z^i$ are the intrinsic spins). In the COM frame, therefore, (C.1) can be expressed as follows (see (C.18) of [41]),

$$|(\vec{p}, \theta, \phi)\,; \lambda_1, \lambda_2\rangle_{\text{id}} = \sqrt{2} \sum_J \sum_{\lambda=-J}^{J} C_J(\vec{p}) e^{i\phi(\lambda_1+\lambda_2-\lambda)} d_{\lambda\lambda_{12}}^{(l)}(\theta) |c, 0; J, \lambda; \lambda_1, \lambda_2\rangle_{\text{id}} \tag{C.8}$$

The sum over $J$ is not unbounded and can be fixed as follows. Let us consider the case where the $\vec{p}$ is aligned along the $z$-axis (i.e $\theta = \phi = 0$). The LHS is an eigenstate of $J_z = \lambda_1 - \lambda_2$, since the orbital angular momentum is zero and the projection of intrinsic spin onto the direction of momenta now becomes the helicity itself. The RHS sum over $\lambda$ therefore must be therefore over only those states for which $\lambda = \lambda_1 - \lambda_2$, and hence $J \geq |\lambda_1 - \lambda_2|$.

$$|(\vec{p}, 0, 0)\,; \lambda_1, \lambda_2\rangle_{\text{id}} = \sqrt{2} \sum_{J=|\lambda_1-\lambda_2|}^{\infty} C_J(\vec{p}) |c, 0; J, \lambda = |\lambda_1 - \lambda_2|; \lambda_1, \lambda_2\rangle_{\text{id}} \tag{C.9}$$

We can now apply the rotation matrix on both sides to bring it to the form (C.8). Note that the rotation matrix does not change the Casimir $J^2$ (and hence the $J$ sum).

$$|(\vec{p}, \theta, \phi)\,; \lambda_1, \lambda_2\rangle_{\text{id}} = \sqrt{2} \sum_{J=|\lambda_1-\lambda_2|}^{\infty} \sum_{\lambda=-J}^{J} C_J(\vec{p}) e^{i\phi(\lambda_1+\lambda_2-\lambda)} d_{\lambda\lambda_{12}}^{(l)}(\theta) |c, 0; J, \lambda; \lambda_1, \lambda_2\rangle_{\text{id}} \tag{C.10}$$

To summarise, the additional symmetry due to the identical nature of 2 particle states decides even or odd spins (or both) appear in the partial wave expansion. The cut-off for the spin is decided by the helicities of the constituent states.

Corresponding to the incoming and the outgoing states, therefore, we can write down a basis of irreducible 2 particle states for each spin $l$ that appears in the decomposition (C.1). Generically they are denoted as,

$$\begin{aligned}
|1\rangle_{\text{in}} &= |c, \vec{p}; J, \lambda; j_1, j_2\rangle_{\text{in}}, & |1\rangle_{\text{out}} &= |c, \vec{p}; J, \lambda; j_3, j_4\rangle_{\text{out}} \\
|2\rangle_{\text{in}} &= |c, \vec{p}; J, \lambda; j_1 - 1, j_2\rangle_{\text{in}}, & |2\rangle_{\text{out}} &= |c, \vec{p}; J, \lambda; j_3 - 1, j_4\rangle_{\text{out}} \\
&\qquad\qquad . \\
&\qquad\qquad . \\
|N_{\text{in}}\rangle_{\text{in}} &= |c, \vec{p}; J, \lambda; -j_1, -j_2\rangle_{\text{in}}, & |N_{\text{out}}\rangle_{\text{out}} &= |c, \vec{p}; J, \lambda; -j_3, -j_4\rangle_{\text{out}}
\end{aligned} \tag{C.11}$$

Imposing the positivity constraints of hermitian matrix, therefore, translates to positivity of the following matrix (eq 2.118 of [41])

$$\mathcal{H}_J(s) \times (2\pi)^4 \delta^4 \delta_{J'J} \delta_{\lambda'\lambda} = \begin{pmatrix} {}_{\text{in}}\langle a'|b\rangle_{\text{in}} & {}_{\text{in}}\langle a'|b\rangle_{\text{out}} \\ {}_{\text{out}}\langle a'|b\rangle_{\text{in}} & {}_{\text{out}}\langle a'|b\rangle_{\text{out}} \end{pmatrix} \tag{C.12}$$

where $s = -p^2$. Using the normalisation conditions (C.4), and

$$_{\text{out}}\langle c', \vec{p}'; J', \lambda'; \lambda_1', \lambda_2' \,|\, c, \vec{p}; J, \lambda; \lambda_1, \lambda_2 \rangle_{\text{in}} = (2\pi)^4 \delta^4(p'^\mu - p^\mu)\, \delta_{J'J}\delta_{\lambda'\lambda}\, S_J^{\lambda_1', \lambda_2'}(s) \tag{C.13}$$

where $S_{\ell\lambda_1,\lambda_2}^{\lambda_1', \lambda_2'}(s) = (\delta_{\lambda_1'\lambda_1}\delta_{\lambda_2'\lambda_2} + (-1)^{J-(\lambda_1'-\lambda_2')}\delta_{\lambda_2'\lambda_1}\delta_{\lambda_1'\lambda_2}) + iT_{\ell\lambda_1,\lambda_2}^{\lambda_1',\lambda_2'}(s)$ is the partial amplitude with spin $J$, we get,

$$\begin{pmatrix} \delta_{a'b} & S_{\ell a'b}^* \\ S_{\ell a'b} & \delta_{a'b} \end{pmatrix} \succeq 0 \tag{C.14}$$

## C.1 Massless bosons: Photons and gravitons

The two particle irreducible states can be labelled by the helicities ($\lambda_i = \pm m$ with $m = 1, 2$ for photons and gravitons respectively ) as

$$\begin{aligned}
|c, \vec{p}; J, \lambda; \lambda_1, \lambda_2\rangle &\equiv |\lambda_1, \lambda_2\rangle \\
|1\rangle &= \frac{1}{\sqrt{2}} \left( |++\rangle + |--\rangle \right), \qquad J = 0, 2, 4, \cdots \\
|2\rangle &= \sqrt{2} \left( |+-\rangle \right), \qquad J = 2m, 2m+1, 2m+2, 2m+3, \cdots \\
|3\rangle &= \frac{1}{\sqrt{2}} \left( |++\rangle - |--\rangle \right), \qquad J = 0, 2, 4, \cdots
\end{aligned} \tag{C.15}$$

We note that the states $|1\rangle$ and $|3\rangle$ only contain even spin. This is evident from the symmetry of the states ( see (C.6)) and the discussion around (C.10). In the following subsections, we try to impose the unitarity conditions assuming parity invariance and non-invariance respectively. For gravitons $\lambda_i = \pm 2$, so spins will change as $|\lambda_1 - \lambda_2|$.

### C.1.1 Parity invariant theories

We note that the parity of these states: states $|1\rangle$ and $|2\rangle$ are parity even states while $|3\rangle$ is a parity odd state. This is due to the following[21]

$$\mathcal{P}|++\rangle = |--\rangle, \quad \mathcal{P}|--\rangle = |++\rangle, \quad \mathcal{P}|+-\rangle = (-1)^J|-+\rangle = (-1)^J(-1)^{J-2}|+-\rangle = |+-\rangle \tag{C.16}$$

We are now in a position to evaluate the matrix (C.14) for the set of states (C.15). Furthermore, we assume parity invariance which implies that the states of definite parity do not mix. The following conditions are obtained for the parity even sector.

$$\begin{pmatrix} _{\text{in}}\langle 1'|1\rangle_{\text{in}} & _{\text{in}}\langle 1'|1\rangle_{\text{out}} \\ _{\text{out}}\langle 1'|1\rangle_{\text{in}} & _{\text{out}}\langle 1'|1\rangle_{\text{out}} \end{pmatrix} \succeq 0, \qquad J = 0,$$

$$\begin{pmatrix} _{\text{in}}\langle 2'|2\rangle_{\text{in}} & _{\text{in}}\langle 2'|2\rangle_{\text{out}} \\ _{\text{out}}\langle 2'|2\rangle_{\text{in}} & _{\text{out}}\langle 2'|2\rangle_{\text{out}} \end{pmatrix} \succeq 0, \qquad J = 2m+1, 2m+3, \ldots$$

$$\begin{pmatrix} _{\text{in}}\langle 1'|1\rangle_{\text{in}} & _{\text{in}}\langle 1'|2\rangle_{\text{in}} & _{\text{in}}\langle 1'|1\rangle_{\text{out}} & _{\text{in}}\langle 1'|2\rangle_{\text{out}} \\ _{\text{in}}\langle 2'|1\rangle_{\text{in}} & _{\text{in}}\langle 2'|2\rangle_{\text{in}} & _{\text{in}}\langle 2'|1\rangle_{\text{out}} & _{\text{in}}\langle 2'|2\rangle_{\text{out}} \\ _{\text{out}}\langle 1'|1\rangle_{\text{in}} & _{\text{out}}\langle 1'|2\rangle_{\text{in}} & _{\text{out}}\langle 1'|1\rangle_{\text{out}} & _{\text{out}}\langle 1'|2\rangle_{\text{out}} \\ _{\text{out}}\langle 2'|1\rangle_{\text{in}} & _{\text{out}}\langle 2'|2\rangle_{\text{in}} & _{\text{out}}\langle 2'|1\rangle_{\text{out}} & _{\text{out}}\langle 2'|2\rangle_{\text{out}} \end{pmatrix} \succeq 0, \qquad J = 2m, 2m+2, \ldots \tag{C.17}$$

---

[21]We work in the convention $\eta_i = 1$ for photons.

and for the parity odd sector,

$$
\begin{pmatrix} {}_{\mathrm{in}}\langle 3'|3\rangle_{\mathrm{in}} & {}_{\mathrm{in}}\langle 3'|3\rangle_{\mathrm{out}} \\ {}_{\mathrm{out}}\langle 3'|3\rangle_{\mathrm{in}} & {}_{\mathrm{out}}\langle 3'|3\rangle_{\mathrm{out}} \end{pmatrix} \succeq 0, \qquad J = 0, 2, 4, \ldots
$$

(C.18)

Let us work out one of the conditions in detail: 1st matrix of (C.17) gives us the following

$$
\begin{pmatrix} 1 & 1 - i\left(T_{\ell++}^{*\,--} + T_{\ell++}^{*\,++}\right) \\ 1 + i\left(T_{\ell++}^{--} + T_{\ell++}^{++}\right) & 1 \end{pmatrix} \succeq 0, \qquad J = 0,
$$

(C.19)

Noting that the trace is positive trivially, the condition of positivity translates to the determinant of the matrix being positive:

$$
2\,\mathrm{Im}(T_1^{J=0} + T_2^{J=0}) \geq |T_1^{J=0} + T_2^{J=0}|^2 \geq 0
$$

(C.20)

Similarly, 2nd matrix of (C.17) and (C.18) gives us the following

$$
\begin{aligned}
\mathrm{Im}T_3^J &\geq |T_3^J|^2 \geq 0, & J = 2m+1, 2m+3, \ldots. \\
2\mathrm{Im}(T_1^J - T_2^J) &\geq |T_1^J - T_2^J|^2 \geq 0, & J = 0, 2, 4, \ldots.
\end{aligned}
$$

(C.21)

Now let us consider the conditions coming from the third matrix in (C.17). We find that analysing the $2 \times 2$ principal minors is sufficient for our purposes and we obtain,

$$
\begin{aligned}
\mathrm{Im}T_3^J &\geq |T_3^J|^2 \geq 0, & J = 2m, 2m+2, \ldots, \\
2\mathrm{Im}(T_1^J + T_2^J) &\geq |T_1^J + T_2^J|^2 \geq 0, & J = 2, 4, \ldots, \\
1 &\geq 4|T_5^J|^2 \geq 0, & J = 2m, 2m+2, \ldots
\end{aligned}
$$

(C.22)

### C.1.2 Parity violating theories

For this case, the assumption that parity even and odd states do not mix no longer holds true. This leads to the modification of the unitarity equations,

$$
\begin{pmatrix} {}_{\mathrm{in}}\langle 1'|1\rangle_{\mathrm{in}} & {}_{\mathrm{in}}\langle 1'|3\rangle_{\mathrm{in}} & {}_{\mathrm{in}}\langle 1'|1\rangle_{\mathrm{out}} & {}_{\mathrm{in}}\langle 1'|3\rangle_{\mathrm{out}} \\ {}_{\mathrm{in}}\langle 3'|1\rangle_{\mathrm{in}} & {}_{\mathrm{in}}\langle 3'|3\rangle_{\mathrm{in}} & {}_{\mathrm{in}}\langle 3'|1\rangle_{\mathrm{out}} & {}_{\mathrm{in}}\langle 3'|3\rangle_{\mathrm{out}} \\ {}_{\mathrm{out}}\langle 1'|1\rangle_{\mathrm{in}} & {}_{\mathrm{out}}\langle 1'|3\rangle_{\mathrm{in}} & {}_{\mathrm{out}}\langle 1'|1\rangle_{\mathrm{out}} & {}_{\mathrm{out}}\langle 1'|3\rangle_{\mathrm{out}} \\ {}_{\mathrm{out}}\langle 3'|1\rangle_{\mathrm{in}} & {}_{\mathrm{out}}\langle 3'|3\rangle_{\mathrm{in}} & {}_{\mathrm{out}}\langle 3'|1\rangle_{\mathrm{out}} & {}_{\mathrm{out}}\langle 3'|3\rangle_{\mathrm{out}} \end{pmatrix} \succeq 0, \qquad J = 0
$$

$$
\begin{pmatrix} {}_{\mathrm{in}}\langle 2'|2\rangle_{\mathrm{in}} & {}_{\mathrm{in}}\langle 2'|2\rangle_{\mathrm{out}} \\ {}_{\mathrm{out}}\langle 2'|2\rangle_{\mathrm{in}} & {}_{\mathrm{out}}\langle 2'|2\rangle_{\mathrm{out}} \end{pmatrix} \succeq 0, \qquad J = 2m+1, 2m+3, \ldots
$$

$$
\begin{pmatrix} {}_{\mathrm{in}}\langle 1'|1\rangle_{\mathrm{in}} & {}_{\mathrm{in}}\langle 1'|2\rangle_{\mathrm{in}} & {}_{\mathrm{in}}\langle 1'|3\rangle_{\mathrm{in}} & {}_{\mathrm{in}}\langle 1'|1\rangle_{\mathrm{out}} & {}_{\mathrm{in}}\langle 1'|2\rangle_{\mathrm{out}} & {}_{\mathrm{in}}\langle 1'|3\rangle_{\mathrm{out}} \\ {}_{\mathrm{in}}\langle 2'|1\rangle_{\mathrm{in}} & {}_{\mathrm{in}}\langle 2'|2\rangle_{\mathrm{in}} & {}_{\mathrm{in}}\langle 2'|3\rangle_{\mathrm{in}} & {}_{\mathrm{in}}\langle 2'|1\rangle_{\mathrm{out}} & {}_{\mathrm{in}}\langle 2'|2\rangle_{\mathrm{out}} & {}_{\mathrm{in}}\langle 2'|3\rangle_{\mathrm{out}} \\ {}_{\mathrm{in}}\langle 3'|1\rangle_{\mathrm{in}} & {}_{\mathrm{in}}\langle 3'|2\rangle_{\mathrm{in}} & {}_{\mathrm{in}}\langle 3'|3\rangle_{\mathrm{in}} & {}_{\mathrm{in}}\langle 3'|1\rangle_{\mathrm{out}} & {}_{\mathrm{in}}\langle 3'|2\rangle_{\mathrm{out}} & {}_{\mathrm{in}}\langle 3'|3\rangle_{\mathrm{out}} \\ {}_{\mathrm{out}}\langle 1'|1\rangle_{\mathrm{in}} & {}_{\mathrm{out}}\langle 1'|2\rangle_{\mathrm{in}} & {}_{\mathrm{out}}\langle 1'|3\rangle_{\mathrm{in}} & {}_{\mathrm{out}}\langle 1'|1\rangle_{\mathrm{out}} & {}_{\mathrm{out}}\langle 1'|2\rangle_{\mathrm{out}} & {}_{\mathrm{out}}\langle 1'|3\rangle_{\mathrm{out}} \\ {}_{\mathrm{out}}\langle 2'|1\rangle_{\mathrm{in}} & {}_{\mathrm{out}}\langle 2'|2\rangle_{\mathrm{in}} & {}_{\mathrm{out}}\langle 2'|3\rangle_{\mathrm{in}} & {}_{\mathrm{out}}\langle 2'|1\rangle_{\mathrm{out}} & {}_{\mathrm{out}}\langle 2'|2\rangle_{\mathrm{out}} & {}_{\mathrm{out}}\langle 2'|3\rangle_{\mathrm{out}} \\ {}_{\mathrm{out}}\langle 3'|1\rangle_{\mathrm{in}} & {}_{\mathrm{out}}\langle 3'|2\rangle_{\mathrm{in}} & {}_{\mathrm{out}}\langle 3'|3\rangle_{\mathrm{in}} & {}_{\mathrm{out}}\langle 3'|1\rangle_{\mathrm{out}} & {}_{\mathrm{out}}\langle 3'|2\rangle_{\mathrm{out}} & {}_{\mathrm{out}}\langle 3'|3\rangle_{\mathrm{out}} \end{pmatrix} \succeq 0, \qquad J = 2, 4, 6 \cdots
$$

(C.23)

The analysis of these matrices is tedious and we find the following constraints

$$\mathrm{Im}T_3^J \geq |T_3^J|^2 \geq \quad 0, \qquad J = 2m, 2m+1\ldots,$$
$$2\mathrm{Im}(T_1^J + \frac{1}{2}(T_2^J + T_2'^J)) \geq |T_1^J + \frac{1}{2}(T_2^J + T_2'^J)|^2 \geq \quad 0, \qquad J = 0, 2, 4, 6, \ldots,$$
$$2\mathrm{Im}(T_1^J - \frac{1}{2}(T_2^J + T_2'^J)) \geq |T_1^J - \frac{1}{2}(T_2^J + T_2'^J)|^2 \geq \quad 0, \qquad J = 0, 2, 4, 6, \ldots,$$
$$|T_2^J - T_2'^J|^2 \leq \quad 4, \qquad J = 0, 1, 2, \ldots$$
$$|T_5^J - T_5'^J| \leq \quad 1, \qquad J = 2m, 2m+1, \ldots$$
$$|T_5^J + T_5'^J| \leq \quad 1 \qquad J = 2m, 2m+1, \ldots$$

$$(\text{C.24})$$

From the last three conditions listed above, it seems that linear unitarity analysis doesn't fix the sign of $\rho_2^J - \rho_2'^J, \rho_5^J \pm \rho_5'^J$ and perhaps a more thorough investigation is required [7]. Hence, in this work, we do not attempt to bound the parity violating amplitudes.

## C.2 Massive Majorana fermions

The unitarity conditions for massive Majorana fermions were spelt out in [41]. Let us quickly review them for completeness. Recalling the fermion amplitudes $\{\Phi_i\}$ defined in (2.36), the corresponding partial amplitudes are denoted as $\{\Phi_i^J\}$. Then, following the similar arguements as in the previous subsection, one arrives at the unitarity conditions as follows:

1.
$$\begin{pmatrix} 1 & 1 - i\left[\Phi_1^{0*}(s) - \Phi_2^{0*}(s)\right] \\ 1 + i\left[\Phi_1^0(s) - \Phi_2^0(s)\right] & 1 \end{pmatrix} \succeq 0, . \qquad (\text{C.25})$$

The positivity of the determinant of the matrix then gives

$$2\,\mathrm{Im}.\left[\Phi_1^0(s) - \Phi_2^0(s)\right] \geq \left|\Phi_1^0(s) - \Phi_2^0(s)\right|^2. \qquad (\text{C.26})$$

2.
$$\begin{pmatrix} 1 & 1 - i\left[\Phi_1^{J*}(s) + \Phi_2^{J*}(s)\right] \\ 1 + i\left[\Phi_1^J(s) + \Phi_2^J(s)\right] & 1 \end{pmatrix} \succeq 0, \qquad J \geq 0 \text{ (even)}, \qquad (\text{C.27})$$

implying
$$2\,\mathrm{Im}.\left[\Phi_1^J(s) + \Phi_2^J(s)\right] \geq \left|\Phi_1^J(s) + \Phi_2^J(s)\right|^2, \qquad J \geq 0 \text{ (even)}. \qquad (\text{C.28})$$

3.
$$\begin{pmatrix} 1 & 1 - 2i\,\Phi_3^{J*} \\ 1 + 2i\,\Phi_3^J & 1 \end{pmatrix} \succeq 0, \qquad J \geq 1 \text{ (odd)}, \qquad (\text{C.29})$$

with straightforward consequence

$$\mathrm{Im}.\,\Phi_3^J(s) \geq |\Phi_3^J(s)|^2 \geq 0 \qquad J \geq 1 \text{ (odd)}. \qquad (\text{C.30})$$

4.
$$\begin{pmatrix} \mathbb{I}_{2\times 2} & \mathbb{S}_{2\times 2}^{J\dagger} \\ \mathbb{S}_{2\times 2}^J & \mathbb{I}_{2\times 2} \end{pmatrix} \succeq 0, \qquad J = 2, 4, 6, \ldots \qquad (\text{C.31})$$

with
$$\mathbb{I}_{2\times 2} := \begin{pmatrix} 1 & 0 \\ 0 & 1 \end{pmatrix}, \qquad \mathbb{S}_{2\times 2}^J(s) := \begin{pmatrix} 1 + i\left[\Phi_1^J(s) - \Phi_2^J(s)\right] & 2i\,\Phi_5^{J*} \\ 2i\,\Phi_5^J(s) & 1 + 2i\,\Phi_3^J(s) \end{pmatrix} \qquad (\text{C.32})$$

We get,

$$\mathrm{Det}[\mathbb{S}_{2\times 2}^J(s)] \geq 0$$

# D Representations of Wigner-$d$ functions

In this appendix, we give some convenient representations of the Wigner-$d$ functions that we used in the main text for computational ease. For the photons we use,

$$
\begin{aligned}
d_{0,0}^{J}\left(\cos^{-1}\sqrt{\xi(s_1',a)}\right) &= {}_2F_1\left(-J,J+1;1;\frac{1}{2}\left(1-\sqrt{\xi}\right)\right), \\
d_{2,2}^{J}\left(\cos^{-1}\sqrt{\xi(s_1',a)}\right) &= \frac{1}{24}\left(6\left(\sqrt{\xi}+1\right)^2 {}_2F_1\left(2-J,J+3;1;\frac{1}{2}\left(1-\sqrt{\xi}\right)\right)\right) \\
d_{2,-2}^{J}\left(\cos^{-1}\sqrt{\xi(s_1',a)}\right) &= \frac{\left(1-\sqrt{\xi}\right)^2\Gamma(J+3)\,{}_2F_1\left(2-J,J+3;5;\frac{1}{2}\left(1-\sqrt{\xi}\right)\right)}{96\Gamma(J-1)} \\
d_{4,4}^{J}\left(\cos^{-1}\sqrt{\xi(s_1',a)}\right) &= \frac{1}{16}(\sqrt{\xi}+1)^4\,{}_2F_1\left(4-J,J+5;1;\frac{1-\sqrt{\xi}}{2}\right) \\
d_{4,-4}^{J}\left(\cos^{-1}\sqrt{\xi(s_1',a)}\right) &= \frac{(1-\sqrt{\xi})^4\Gamma(J+5)\,{}_2F_1\left(4-J,J+5;9;\frac{1-\sqrt{\xi}}{2}\right)}{645120\Gamma(J-3)}
\end{aligned}
\tag{D.1}
$$

# E Massive Majorana fermions: Locality constraints

We would like to use our techniques to constrain the EFTs involving Majorana fermions. However, in this work we will only spell out the locality constraints and leave a careful analysis for future work. Now we will list the locality constraint for the amplitude listed in (2.43). Let us assume a low energy EFT expansion of the form

$$
\Psi_1(s_1,s_2,s_3) = \sum_{n,m}\mathcal{W}_{p,q}^{\psi}x^p y^q
\tag{E.1}
$$

The partial wave decomposition reads,

$$
(\Psi_1(s_1,s_2)) = \alpha_0^{\psi} + \frac{1}{\pi}\int_{M^2}^{\infty}\frac{ds_1}{s_1'}\mathcal{A}^{\psi}\left(s_1';s_2^{(+)}\left(s_1',a\right)\right)H\left(s_1';s_1,s_2,s_3\right),
\tag{E.2}
$$

where $H\left(s_1';s_1,s_2,s_3\right)$ is defined in (3.4) and the partial wave decomposition reads

$$
\begin{aligned}
\mathcal{A}^{\psi}\left(s_1';s_2^{(+)}\left(s_1',a\right)\right) &= \sum_{J=0,2,4,\cdots}16\pi(2J+1)\rho_J^{1,\psi}d_{0,0}^{J}(\theta) + \sum_{J=1,2,3,\cdots}64\pi(2J+1)(-1)^{J+1}\rho_J^{3,\psi}d_{1,-1}^{J}(\theta) \\
&\quad - \sum_{J=0,2,4,\cdots}16\pi(2J+1)\rho_J^{5,\psi}d_{0,1}^{J}(\theta)
\end{aligned}
\tag{E.3}
$$

where $(\cos\theta)^2 = \xi(s_1',a) = \xi_0 + 4\xi_0\left(\frac{a}{s_1'-a}\right)$ and $\xi_0 = \frac{s_1^2}{s_1-M^2}$ while $\rho_J^{i,\psi}$ are the respective spectral functions which appear as coefficients in partial wave expansion of the absorptive parts $\mathcal{A}^{\psi}$. We have also used that $\rho_J^{4,\psi} = (-1)^{J+1}\rho_J^{3,\psi}$[41]. The coefficients $\mathcal{W}_{p,q}^{\psi}$ can be obtained from the amplitude via the inversion formula [29]

$$
\begin{aligned}
\mathcal{W}_{n-m,m}^{\psi} &= \int_{M^2}^{\infty}\frac{ds_1}{2\pi s_1^{2n+m+1}}16\pi\left(\sum_{J=0,2,4\cdots}(2J+1)\rho_J^{1,\psi}(s_1)\ominus_{n,m}^{(1,J)}(s_1) + \sum_{J=1,2,3\cdots}(2J+1)\rho_J^{3,\psi}(s_1)\ominus_{n,m}^{(3,J)}(s_1)\right. \\
&\quad \left. \sum_{J=0,2,4\cdots}(2J+1)\rho_J^{5,\psi}(s_1)\ominus_{n,m}^{(5,J)}(s_1)\right).
\end{aligned}
\tag{E.4}
$$

with

$$\ominus_{n,m}^{(i,J)}(s_1) = 2\sum_{j=0}^{m} \frac{(-1)^{1-j+m} r_J^{(i,j)}(\xi_0)\,(4\xi_0)^j\,(3j-m-2n)\Gamma(n-j)}{j!(m-j)!\Gamma(n-m+1)} \qquad \xi_0 := \frac{s_1^2}{(s_1-2\mu/3)^2} \qquad \text{(E.5)}$$

The functions $\{p_J^{(j)}(\xi_0)\}$ are derivatives of respective Wigner-$d$ functions

$$r_J^{(1,j)}(\xi_0) \;\; := \;\; \left.\frac{\partial^j d_{0,0}^J(\sqrt{z})}{\partial z^j}\right|_{z=\xi_0}, \qquad r_J^{(3,j)}(\xi_0) := (-1)^{J+1} \left.\frac{\partial^j d_{1,-1}^J(\sqrt{z})}{\partial z^j}\right|_{z=\xi_0}$$

$$r_J^{(5,j)}(\xi_0) := \left.\frac{\partial^j d_{0,1}^J(\sqrt{z})}{\partial z^j}\right|_{z=\xi_0} \tag{E.6}$$

The locality constraints then are simply,

$$\mathcal{W}_{n-m,m}^{\psi} = 0, \qquad \forall \;\; n < m\,. \tag{E.7}$$

# F  EFT expansion of the crossing basis elements

In this section, we give the low energy EFT expansion for the crossing basis elements (2.7) for some special cases used in the main text. For the photon case, we have the following:

$$
\begin{aligned}
F_1(s_1,s_2,s_3) &= 2f_2 x - f_3 y + 4f_4 x^2 - 2f_5 xy + f_{6,1} y^2 + 8f_{6,2} x^3 + \cdots \\
F_2(s_1,s_2,s_3) &= 2g_2 x - 3g_3 y + 2(g_{41} + 2g_{42})x^2 + (-5g_{5,1} - 3g_{5,2})xy + 3\left(g_{6,1} - g_{6,2} + g_{6,3}\right)y^2 + 2g_{6,1}x^3 + \cdots\,, \\
F_3(s_1,s_2,s_3) &= 2h_2 x - h_3 y + 4h_4 x^2 - 2h_5 xy + h_{6,1} y^2 + 8h_{6,2} x^3 + \cdots \\
F_4(s_1,s_2,s_3) &= \frac{1}{3}\left(g_2 + (g_{41} + 2g_{42})x + (g_{5,2} - g_{5,1})y + g_{6,1}x^2 + \cdots\right), \\
F_5(s_1,s_2,s_3) &= \frac{1}{3}\left(g_3 x - g_{4,1}y + g_{5,1}x^2 - (2g_{6,1} + g_{6,2})xy\right) + \cdots\,.
\end{aligned}
\tag{F.1}
$$

As alluded to in the main text in the discussion below eq.(2.7) if an amplitude is $t-u$ symmetric then the crossing basis has only 3 elements $\{f(s_1,s_2,s_3), g_1(s_1,s_2,s_3), h_1(s_1,s_2,s_3)\}$ and $F_2, F_4, F_5$ above correspond to these for the $t-u$ symmetric amplitude, $T_1(s_1,s_2,s_3)$, while $F_1$ and $F_3$ correspond to the fully crossing symmetric helicity amplitudes $T_2(s_1,s_2,s_3)$ and $T_5(s_1,s_2,s_3)$ respectively

$$
\begin{aligned}
T_1(s_1,s_2,s_3) &= g_2 s_1^2 + g_3 s_1^3 + s_1^4 g_{4,1} + \left(s_1^2\right)\left(s_1^2 + s_2^2 + s_3^2\right)g_{4,2} + s_1^5 g_{5,1} + g_{5,2}\left(s_2^2 s_3 + s_2 s_3^2\right)(s_2 + s_3)^2 \\
&\quad + s_1^6 g_{6,1} + g_{6,2}\left(\left(s_2^3 s_3 + s_2 s_3^3\right)(s_2 + s_3)^2\right) + g_{6,3}\left(\left(s_2^2 s_3^2\right)(s_2 + s_3)^2\right) \\
T_2(s_1,s_2,s_3) &= f_2(s_1^2 + s_2^2 + s_3^2) + f_3(s_1 s_2 s_3) + f_4(s_1^2 + s_2^2 + s_3^2)^2 + f_5(s_1^2 + s_2^2 + s_3^2)(s_1 s_2 s_3) + f_{6,1}(s_1 s_2 s_3)^2 \\
&\quad f_{6,2}(s_1^2 + s_2^2 + s_3^2)^3 \\
T_5(s_1,s_2,s_3) &= h_2(s_1^2 + s_2^2 + s_3^2) + h_3(s_1 s_2 s_3) + h_4(s_1^2 + s_2^2 + s_3^2)^2 + h_5(s_1^2 + s_2^2 + s_3^2)(s_1 s_2 s_3) + h_{6,1}(s_1 s_2 s_3)^2 \\
&\quad h_{6,2}(s_1^2 + s_2^2 + s_3^2)^3
\end{aligned}
\tag{F.2}
$$

For the parity violating case we have two additional elements corresponding to the crossing symmetric amplitudes $T_{2'}$ and $T_{5'}$. We note that $F_i$ for $i = 1, 2, 3$ are the same ones considered in [22], in this paper additionally we also use $F_4$ and $F_5$.

# G  Low spin dominance and Graviton scattering in String theory

In this appendix, we will show that the range of $a$ that we had obtained from our Locality constraint analysis in subsection 6.2, is satisfied for type II string theory amplitude. For convenience, we write

the explicit amplitude

$$T_1 = \frac{s_1^3 \left( \frac{\Gamma\left(1-\frac{\alpha s_1}{2}\right)\Gamma\left(1-\frac{\alpha s_2}{2}\right)\Gamma\left(1-\frac{\alpha s_3}{2}\right)}{\Gamma\left(\frac{s_1\alpha}{2}+1\right)\Gamma\left(\frac{s_2\alpha}{2}+1\right)\Gamma\left(\frac{s_3\alpha}{2}+1\right)} - 1 \right)}{s_2 s_3}$$

$$T_3 = \frac{s_3^3 \left( \frac{\Gamma\left(1-\frac{\alpha s_1}{2}\right)\Gamma\left(1-\frac{\alpha s_2}{2}\right)\Gamma\left(1-\frac{\alpha s_3}{2}\right)}{\Gamma\left(\frac{s_1\alpha}{2}+1\right)\Gamma\left(\frac{s_2\alpha}{2}+1\right)\Gamma\left(\frac{s_3\alpha}{2}+1\right)} - 1 \right)}{s_1 s_2}$$

$$T_4 = \frac{s_2^3 \left( \frac{\Gamma\left(1-\frac{\alpha s_1}{2}\right)\Gamma\left(1-\frac{\alpha s_2}{2}\right)\Gamma\left(1-\frac{\alpha s_3}{2}\right)}{\Gamma\left(\frac{s_1\alpha}{2}+1\right)\Gamma\left(\frac{s_2\alpha}{2}+1\right)\Gamma\left(\frac{s_3\alpha}{2}+1\right)} - 1 \right)}{s_1 s_3} \tag{G.1}$$

Note that we have subtracted out the massless graviton pole. We obtain the $a_i^J(s_1)$ by considering the string amplitude as an infinite sum over poles at $s_1 = \frac{2(n+1)}{\alpha'}$. We want to verify that the constraints (6.12) are satisfied for our range of a: $-0.1933M^2 \le a \le \frac{2}{3}M^2$. To be precise, we want to verify,

$$\mathcal{M}(s_i, a) =$$
$$\int_{M^2}^{\infty} \frac{ds_1}{2\pi s_1} \sum_{J=0,2,4,\cdots} (2J+1)\tilde{a}_J^{(1)}(s_1)f_J^{(1)}(\xi)H(s_1', s_i) + \int_{M^2}^{\infty} \frac{ds_1}{2\pi s_1} \sum_{J=4,6,\cdots} (2J+1)\tilde{a}_J^{(2)}(s_1)f_J^{(2)}(\xi)H(s_1', s_i)$$
$$+ \int_{M^2}^{\infty} \frac{ds_1}{2\pi s_1} \sum_{J=5,7,\cdots} (2J+1)\tilde{a}_J^{(3)}(s_1)f_J^{(3)}(\xi)H(s_1', s_i) \ge 0,$$
$$= \mathcal{M}^{J\le 2}(s_i, a) + \mathcal{M}^{J>2}(s_i, a) \ge 0 \tag{G.2}$$

for the range of a which is where $\mathcal{M}^{J\le 2}(s_i, a) \ge 0$. This is what we called weak LSD in the main text.

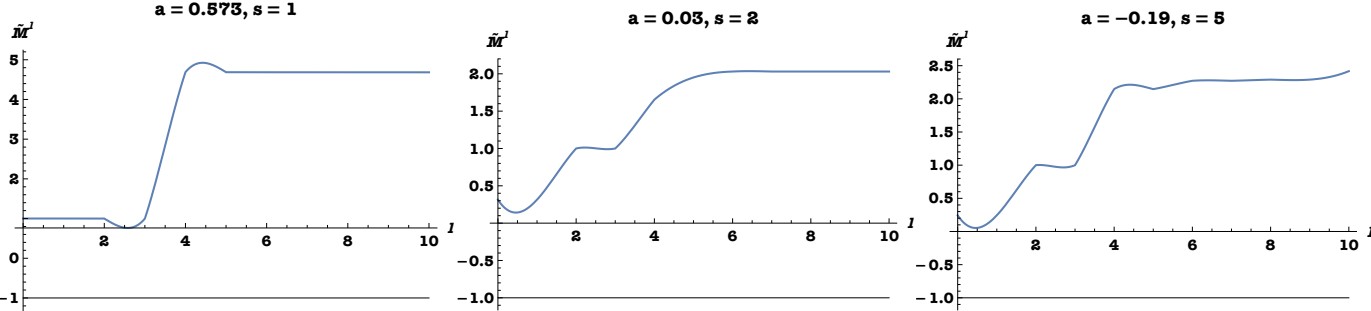

**Figure 9:** $\tilde{\mathcal{M}}^J(s_i, a)$ vs $J$ for various values of $a$.

We consider, where $\tilde{a}_J^{(i)}(s_1) = \frac{a_J^{(i)}(s_1)}{s_1^2}$ and $f_J^{(i)}(\xi)$ relevant for spin 2 have been defined around eq (6.7). Note that we have dropped the null constraints since a physical amplitude satisfies that by default. We have found that the positivity condition is satisfied for our range "a". We present our analysis in the figure 9 above for different values of $a$. The plots show $\tilde{\mathcal{M}}^J(s_i, a) = \frac{\mathcal{M}^J(s_i, a)}{\mathcal{M}^{J\le 2}(s_i, a)}$ as a function of $J$ for various values of $a$. We note that firstly, as advertised, our amplitude is positive for this region of a since $\tilde{\mathcal{M}}^J(s_i, a) > -1$. Secondly, we note that the maximal contribution to the amplitude occurs between spins 2 and 4 which is consistent with our observation that our analysis in subsection 6.2, indicated a spin 2 dominance.

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
