# Peer review of "Crossing Symmetric Spinning S-matrix Bootstrap: EFT bounds"

_SciPost Physics_

## Round 3 · Referee Report · Anonymous (Referee 1) · 2022-5-7

Report

The manuscript uses an explicitly crossing symmetric dispersion relation to derive two-sided positivity bounds on particles with spin, specifically on photons and gravitons, making use of existing results in geometric function theory. This is an important generalization of what the some of the authors have done before for scalars. Particularly, the authors show that low spin dominance is a natural consequence of the formalism, rather than an extra assumption that is needed as an input. The paper is interesting, but the writing, particularly when discussing technical details, is a bit confusing sometimes. I would recommend its publication if the following questions are addressed:

  1. What is the relation between PB_C and the SDPB results in Table 2?

  2. The lower bound on \alpha below Eq 3.32 is 0, but the lower bound is relaxed to − 0.1933 M^2 in Eq 3.37. Should the locality constraints improve the bounds? Also, the table below Eq 3.37 is a bit confusing. In general, the discussion in 3.5.2 and 3.5.3 is a bit confusing to me, please present it more transparently.

  3. Can we use the crossing symmetric dispersion relation and the locality constraints and formulate it as a SDP problem that is solvable by SDPB to obtain the optimal bounds, in a way similar but different to Ref [5]? If possible, how does it compare with Ref [5] in terms of numerically complicity?

  4. Why can the geometric function theory not give the optimal bounds, say, for a scalar? Is it because the a typically real function does not capture all the information of an EFT amplitude?

  5. The obtained bounds for photons and gravitons do not seem to be optimal, apart from using the geometric function theory simplication. At least, there is another group that computed the case of photon (Henriksson et al). How do the results of this paper compare to those?

  6. I have noticed quite a few typos. Please give the paper a thorough read and correct them.

---

## Round 3 · Referee Report · Anonymous (Referee 2) · 2022-5-20

Report

In the manuscript the authors present the use of explicitely crossing symmetric dispersion relations in order to put bounds on theories with particles with spin. The work is based on and extends previous works of the same authors, and their perspective is very relevant in the active and friutful field of constraints on EFTs. However, the manuscript in its current form is not very clear, and would benefit from a real effort in writing the ideas in a comprehensive way. For instance,

  • In page 20, it is not clear to me how to interpret the Table, and how the lower bound on $a$ for different $J_c$ is linked with the low spin dominance. The values do not converge to Eq.3.37?
  • Between Eq 3.32 and 3.33, it is mentioned that each term in the partial wave decomposition 3.9 is set to be positive, and then argued that this might be a too strong condition, and it is relaxed while including the null constraints. What is the phyiscal interpretation of this? It is correct to say that forcing each term to be non-negative implies to have only $\ell=0$ contributions, since $\ell>0$ have a Legendre polynomial that takes positive or negative values depending on $t$? However, allowing them to have any sign, but the sum still positive, together with the null constraints, give bounds on the higher partial waves? In general, it would be nice to back up the numerics and the technical parts with some physical intuition to make it more digestive to the people that want to learn the techniques.
  • In Table 2, a comparison with Ref [5] is performed. Even though below the discrepancy is explained, I'm not sure I understand it. Does it mean that the algoritm bounds a combination of coefficients? Can this be disentangled? Does one understands when this happens, and theferore have a control on whether the bounds are optimal or not? Is there a systematic way to improve them?
  • Is there a simple way to interpret the conditions $TR_U$ in the language of Ref [5]?
  • As a general comment, how much the procedure depends on assuming Eq 3.10, sothat the EFT amplitude is at tree level and therefore there are no IR loops?

In general, the manuscript contains many interesting ideas that can have an impact on the field. However, sometimes the text lacks clarity and would benefit from a precise explanation. Unfortunately I cannot recommend its publication unless the points above, and the presentation of the manuscript, are improved.

---

## Editorial Decision

resubmitted